# Spectral Clustering and Labeling for Crowdsourcing with Inherently Distinct Task Types

**Saptarshi Mandal**[*]                                  *smandal4@illinois.edu*
**Seo Taek Kong**[*]                                     *skong10@illinois.edu*
**Dimitrios Katselis**                                   *katselis@illinois.edu*
**R. Srikant**                                           *rsrikant@illinois.edu*
*Department of Electrical and Computer Engineering*
*University of Illinois Urbana-Champaign*

**Reviewed on OpenReview:** *https://openreview.net/forum?id=jVQjtzcvAc*

## Abstract

The Dawid-Skene model is the most widely assumed model in the analysis of crowdsourcing algorithms that estimate ground-truth labels from noisy worker responses. In this work, we are motivated by crowdsourcing applications where workers have distinct skill sets and their accuracy additionally depends on a task's type. *Focusing on the case where there are two types of tasks, we propose a spectral method to partition tasks into two groups such that a worker has the same reliability for all tasks within a group. Our analysis reveals a separability condition such that task types can be perfectly recovered if the number of workers n scales logarithmically with the number of tasks d.* Numerical experiments show how clustering tasks by type before estimating ground-truth labels enhances the performance of crowdsourcing algorithms in practical applications.

## 1 Introduction

Labeled datasets are required in many machine learning applications to either train classifiers using supervised learning or to evaluate their performance. Crowdsourcing is a popular way to label large datasets by collecting labels from a large number of workers at a low cost. The collected labels are often noisy due to many reasons including the difficulty of some labeling tasks and differing worker skill sets (Bonald & Combes, 2017; Gao et al., 2016)). The crowdsourced labels are then used to infer ground-truth labels by aggregating the responses of the workers. To analyze the quality of the inferred labels, a statistical model for the workers' responses is often assumed.

A widely-studied model for crowdsourcing was first proposed by Dawid & Skene (1979). Their one-coin model assumes that workers have distinct skill sets, and each worker submits responses to a task independently of all other tasks and workers. Formally, each worker $i$ is assumed to submit a response $X_{ij}$ to a task $j$ that correctly reflects the label $y_j$ with an unknown but fixed probability $p_i$. Although the true labels are never observed, it is possible to estimate the unknown accuracy parameters $p = (p_1, \ldots, p_n)$ by assuming that workers respond according to this statistical model. Once the accuracy parameters are estimated, labels can be estimated using the Nitzan-Paroush estimate (Nitzan & Paroush, 1983). Despite the simplicity of this Dawid-Skene model, the optimal error rates of label estimation algorithms have only been understood relatively recently (Berend & Kontorovich, 2014; Gao et al., 2016).

In this paper, we are interested in modeling worker responses when crowdsourced tasks demand different levels of expertise. The considered model is motivated by expert behavior in radiology when labeling the presence of thoracic nodules can be more difficult because of their shape and size, or when they are imaged with different resolutions, resulting in labels that are more reliable for tasks with one type than the other (Shiraishi et al., 2000; He et al., 2016). The contributions of the paper are the following:

---

[*]Equal Contribution

1. We consider a model for crowdsourcing that describes settings when workers label tasks that require different levels of expertise. Hence, different tasks can be associated with different types with this assignment of types being unknown. For this model, we propose a spectral clustering algorithm to cluster the tasks into different types.

2. We analyze the performance of the proposed clustering algorithm and establish sufficient conditions for perfect clustering, focusing on the case of two task types. A key contribution of this paper is proving that the clustering algorithm correctly classifies all tasks with high probability when the number of workers scales logarithmically with the number of tasks which is a natural condition in crowdsourcing applications. To the best of our knowledge, this result is novel for spectral clustering in the context of crowdsourcing models.

   Traditional spectral clustering analyses rely on matrix perturbation results, such as the Davis-Kahan theorem (Yu et al., 2014), which provides bounds on the $l_2$-norm of eigenvector perturbations when noise is added to the signal matrix. However, such bounds are insufficient (see Remark 2) for proving perfect clustering, which requires control over the $l_\infty$-norm of the perturbation, a significantly stronger and more challenging requirement. The Davis-Kahan theorem does not yield meaningful guarantees in this setting.

   Moreover, exact-recovery arguments from stochastic block models (Abbe, 2018)—which rely on matrices with independent Bernoulli edges—do not extend to the dependent, aggregated response matrix that arises in crowdsourcing (see Subsection 2.4 for a detailed discussion), necessitating the $l_\infty$-perturbation analysis we consider here.

   Our main contribution lies in leveraging the specific low-rank signal-plus-perturbation structure of the expected task-similarity matrix. We show that the perturbation remains sufficiently small and adapt techniques from Fan et al. (2018) to establish perfect clustering. While the proof is intricate, we provide a concise outline in the main body of the paper.

3. Clustering, and in particular, identifying the hard tasks may be the end goal in many cases. After the clustering step, one may choose to add more workers to the hard tasks and try to identify experts who are better at the hard tasks. But here, in addition, we also study whether the clusters can be helpful to perform better labeling. For this purpose, we conduct experiments using publicly available datasets. We compared two classes of algorithms: one where we first performed task clustering by type and then applied an algorithm designed for the traditional DS model to label tasks separately for each type and the other where the labeling algorithm is directly applied to the dataset without any clustering. Our experimental results show that clustering followed by labeling outperforms direct labeling in the datasets we considered. We also compared our algorithm with other algorithms which also divide tasks into types. Again, we found that our algorithm outperforms other task type-dependent algorithms.

4. In Section 3.3, we theoretically examine the impact of the clustering step on downstream label estimation in the two-type crowdsourcing model. Specifically, we derive a lower bound on the expected labeling error when applying DS-based weighted majority voting without clustering (i.e., type-agnostic). We then compare this lower bound to the performance of weighted majority voting with clustering, assuming task types are known. Our analysis shows that the latter asymptotically outperforms the lower bound for type-agnostic algorithms.

## 2 Background and Related Work

In this section, we first discuss the model under consideration followed by a discussion on related prior works.

### 2.1 Problem Setting

For any positive integer $m$, denote by $[m]$ the set $\{1, \ldots, m\}$. We use the notation $||\cdot||$ to denote $l_2$-norm and $||\cdot||_\infty$ to denote $l_\infty$-norm in this paper. Let $n \geq 3$ be the number of workers labeling $d$ tasks. Each task $j \in [d]$ is associated with deterministic but unknown ground-truth labels $y_1, y_2, \ldots, y_d \in \{-1, +1\}$ following

Gao et al. (2016). Each worker $i \in [n]$ independently submits a response $X_{ij} \in \{-1, +1\}$ to each task $j$ with $X_{ij}$ being independent across task index $j$. The goal is to estimate the true label $y_j \in \{-1, +1\}$ for every task $j \in [d]$.

Our model is motivated by crowdsourcing scenarios with more than one type of task. For simplicity of exposition in this paper, we are considering there are exactly two types of tasks. More specifically, each task $j$ is associated with a type $k_j \in \{e, h\}$ indicating "easy" and "hard" types, respectively. The task types are also deterministic but unknown, and a task's type $k_j$ determines the accuracy parameter $p_{k_j i} = \mathbb{P}(X_{ij} = y_j)$ as the probability of worker $i$ correctly labeling a task $j$ for all workers $i \in [n]$. Using the accuracy vectors, we can define the reliability vectors $r_e, r_h \in [-1, 1]^n$ as $r_k = 2p_k - 1$, where we denote the $i^{\text{th}}$ element of $p_k$ by $p_{ki}$ for all $k \in \{e, h\}$. Finally, we let the number of tasks of type $k$ be $d_k$; clearly, $d_e + d_h = d$. We assume that $d_k$ is unknown and $r_e \neq r_h$.

This *hard-easy model* is motivated by applications where certain tasks can inherently be more difficult than others. In keeping with the motivation of studying problems with hard and easy tasks, we assume the following:

**Assumption 1**       *1. The reliability vectors satisfy*

    *(a) $\|r_e\|_2 \geq \|r_h\|_2$.*
    *(b) For some universal constant $\rho \in (0, 1/2)$,*

$$\rho \leq \frac{1 \pm r_{ki}}{2} \leq 1 - \rho, \forall i \in [n], \forall k \in \{e, h\}. \tag{1}$$

  *2. There exists $\alpha \in (0, 1)$ such that $d_e = \alpha d$ and $d_h = (1 - \alpha)d$. We assume $\alpha \geq 0.5$, that is, $d_e \geq d_h$.*

  *3. There exists a positive constant $\bar{r}$ such that $\frac{1}{n} \sum_i r_{ki} > \bar{r}$ for all types $k \in \{e, h\}$. Practically speaking, this assumption requires the average reliability of the workers to be positive for each type. Without this assumption, the label vector $y$ is only identifiable up to sign.*

The assumption $d_e \geq d_h$ is practically motivated, as in most crowdsourcing settings, easier tasks tend to be more common than harder ones. Nevertheless, this assumption can be relaxed up to any $\alpha \in (0, 1)$ without affecting the validity of our results.

Our hard-easy model can be considered an extension of the one-coin Dawid-Skene (DS) model to two types of tasks. Henceforth, when we refer to the DS model, we mean the one-coin DS model unless explicitly stated otherwise.

It is worth noting that our model assumes all workers respond to all tasks, as it is motivated by applications where an institution contracts professionals to label a dataset. In this paper, we are not interested in applications that use platforms such as Amazon Mechanical Turk, in which workers independently select a sparse subset of tasks to label.

## 2.2   Related Work: Dawid-Skene Model

Crowdsourcing models differ in the assumed structure for the accuracy matrix $P$, where

$$P_{ij} = \mathbb{P}\left(X_{ij} = y_j\right).$$

In the one-coin DS model, $P$ is a matrix with $d$ identical columns. There is a vast literature on inferring labels from data under this model. These include the original EM algorithm proposed in Dawid & Skene (1979), spectral-EM algorithm in Zhang et al. (2016), message passing algorithm in Karger et al. (2013; 2014b), label estimation from the principal eigenvector of the worker-similarity matrix studied in Dalvi et al. (2013) to name a few. For our experiments, once the tasks are separated by types, we use the following common approach on the DS model to estimate the reliability vector $r_k$ from the responses $X$, denoted as $\hat{r}_k$, and use the Nitzan-Paroush decision rule (Nitzan & Paroush, 1983) to infer the labels for each type $k$:

$$\hat{y}_j^{NP} = \text{sgn}\left(\sum_{i=1}^{n} \log \frac{1 + \hat{r}_{ki}}{1 - \hat{r}_{ki}} X_{ij}\right), \forall j \in [d], k_j = k, \tag{2}$$

where we assign the label as +1 if the argument inside the right-hand side is equal to zero. In the reliability estimation step after the clustering step, we use the Triangular Estimation(TE) algorithm proposed in Bonald & Combes (2017), which we use in our theoretical results. The reason we focus on this algorithm is that it has been compared to other algorithms and shown to perform better in real datasets. Additionally, by comparing the probability of labeling error expression derived from Bonald & Combes (2017) with the lower bounds in Gao et al. (2016), it can be seen that the algorithm is provably asymptotically optimal. We give a brief description of the TE algorithm in Appendix B.

Lu & Zhou (2016) study a sub-Gaussian mixture model and, in the context of crowdsourcing, focus on the standard Dawid–Skene (DS) model where all tasks are assumed to be probabilistically identical. While their framework allows multiple labels per task, it does not incorporate heterogeneity across task types. Their Lloyd-type algorithm is closely related to EM with spectral initialization.

## 2.3 Related Work: More General Models

The DS model has been extended in numerous prior works to account for scenarios where the same worker may exhibit different reliabilities across various tasks. We review these extended models in this subsection.

A rank-1 model studied by Khetan & Oh (2016) assumes that $P$ is an outer product of the accuracy of the workers and a vector parametrizing the easiness of all tasks. A more general model was studied in Shah et al. (2021), where $P$ is assumed to satisfy strong stochastic transitivity (Shah et al., 2016). In the context of crowdsourcing, this assumption implies that workers can be ranked from most to least accurate and that this ranking does not change across tasks. The $P$ that they consider can be associated with a rank as large as $\min(n, d)$. Lastly, the model in Shah & Lee (2018); Kim et al. (2024) assumes an accuracy matrix $P$ that exhibits a low-rank structure with a fixed number of distinct entries. They call it a $k$-type specialization model which is close to a stochastic block model with $k$ communities. The algorithms designed for this model in Shah & Lee (2018); Kim et al. (2024) have a two-step approach. The first step involves clustering workers according to their types. The second step is estimating labels for each task $j$ using a weighted majority vote where significant weight is given to workers that match the type of task $j$ and negligible weight is given to all other workers.

We now compare our model to the above models. As pointed out in Kim et al. (2024), both Khetan & Oh (2016) and Shah et al. (2021) consider the following: if worker A is better than worker B for any task, then this same ordering holds for all other tasks. Such a monotonicity is not assumed in our model. The $k$-type specialization model in Shah & Lee (2018); Kim et al. (2024) is somewhat similar in spirit to our model in the sense it attempts to cluster tasks according to types. However, they also cluster workers according to types and their algorithm uses a simple majority vote or a majority vote with two weights. Such a voting scheme is not optimal when different workers have different reliabilities (Nitzan & Paroush, 1983).

While Ariu et al. (2024) also considers models with multiple task types, their setting and contributions are fundamentally different. They assume all workers are statistically identical, resulting in a rank-1 expected response matrix and making simple majority voting optimal. In contrast, our model captures heterogeneity in worker reliabilities across task types, leading to a rank-2 structure and necessitating the use of weighted majority voting. Moreover, their clustering method requires the number of workers to exceed the number of tasks for reliable recovery, whereas our analysis shows that only $\log(d)$ workers suffice—making our approach significantly more practical in expert-labeling scenarios.

Several well-known probabilistic models also capture heterogeneity in crowdsourcing but pursue a different goal from ours. GLAD Whitehill et al. (2009) represents each worker by a continuous "ability" and each item by a continuous "difficulty" and learns these parameters via Expectation-Maximization, without finite-sample error guarantees. Zhou et al. (2012) assigns a separate latent distribution to every worker–item pair and poses label aggregation as a convex minimax-entropy problem, again with no recovery bounds. The CBCC/EBCC family (Kim & Ghahramani, 2012; Li et al., 2019) introduces latent worker communities with shared confusion matrices, improving aggregation through richer Bayesian priors while still treating tasks as i.i.d. and leaving task types unidentified. The tutorial Drutsa et al. (2019) and the recent paper Ibrahim et al. (2025) review these models from a systems perspective but do not provide new theoretical limits. In contrast, we focus on *discrete latent task types*. We prove that these types can be *perfectly clustered* with only $n = O(\log(d))$

workers under a constant reliability gap. Identifying such hard/easy groups is often an end goal in practice (e.g., triage or active relabeling) and is orthogonal to the per-item difficulty or worker-community structure emphasised by GLAD, Zhou et al. (2012), and BCC variants.

It is also important to note a recent work Han et al. (2025) which considers a different regime from ours: it exploits item features and per-worker continuous embeddings; The method does not posit latent task types; ground truths are not treated as hidden variables in the worker model, and no finite-sample guarantees are reported.

### 2.4   Related Work: Spectral Clustering

Spectral clustering has been widely studied in various contexts. Von Luxburg (2007) provides a comprehensive review of this area. The basic idea behind spectral clustering is to analyze the spectrum of the expected observation matrix and then show that the spectrum of the observed data is close to that of the expected matrix (Von Luxburg, 2007; Ng et al., 2001). The specifics of these steps can differ significantly across applications. To the best of our knowledge, there is limited prior work on spectral clustering specifically for crowdsourcing data. Some works, including Dalvi et al. (2013); Shah et al. (2021); Khetan & Oh (2016) explore the use of spectral methods in crowdsourcing, mainly focusing on analyzing worker-similarity matrices and improving label aggregation. *However, most focus on label aggregation rather than explicitly clustering tasks based on their types. In particular, our result that perfect clustering of tasks by type is possible with $O(\log(d))$ workers when task-type reliabilities are well-separated in norm appears to be novel.*

The condition that a logarithmic number of workers suffices for perfect clustering bears resemblance to sample complexity thresholds in other latent structure models, such as the degree condition for exact recovery in stochastic block models (SBMs) (Abbe, 2018) and the signal-to-noise ratio threshold in sub-Gaussian mixture models (Lu & Zhou, 2016). However, the analogy appears to be superficial. In particular, unlike SBMs where Laplacian matrix entries (Abbe, 2018) are independent Bernoulli random variables, the entries of our task-task similarity matrix are highly dependent, as each aggregates the responses of shared workers taking integer values in $[0, n]$. These dependencies violate key assumptions in SBM-style analyses, which is why classical guarantees—e.g., those based on Fiedler vectors—do not directly apply in our setting.

To frame our setting in mixture-model terms we can treat the four task groups—(easy, label +1), (easy, label −1), (hard, label +1), and (hard, label −1)—as four mixture components over binary vectors of worker responses. While this allows us to apply tools from mixture model analysis, even then, the guarantees differ: Lu & Zhou (2016) show perfect clustering with probability $1 - e^{\sqrt{n}}$ when $n = O(log(d))$, whereas we achieve a stronger guarantee of $1 - e^{-n}$ under the same conditions.

A more direct comparison is with SBM exact-recovery thresholds. For the *general* two–block SBM with possibly different within-class probabilities $p_{11} = a \log d/d$, $p_{22} = c \log d/d$ and cross probability $p_{12} = b \log d/d$, Abbe (2018); Abbe & Sandon (2015) show that exact recovery is information-theoretically possible iff

$$\min_{x \in (0,1)} \left[ x(\sqrt{a} - \sqrt{b})^2 + (1-x)(\sqrt{c} - \sqrt{b})^2 \right] > 2,$$

(see Theorem 3 in Abbe (2018)). By contrast, our exact recovery result requires only $n = O(\log d)$ workers are available, and the reliability gap satisfies $\|r_e\|_2^2 - \|r_h\|_2^2 > \beta n$ for some absolute constant $\beta > 0$, a markedly weaker condition than the thresholds needed for SBM.

These differences highlight why SBM-style guarantees cannot be directly transferred to multi-type crowdsourcing and motivated the new perturbation analysis considered in this work.

## 3   Main Results

### Notation

For clarity, Table 1 summarizes the key symbols used throughout the paper. We follow standard conventions and do not use boldface for vectors; symbols like $r_e$, $r_h$, and $\hat{v}$ denote vectors, while their indexed versions (e.g., $r_{ei}$) refer to scalars.

For a vector $x \in \mathbb{R}^d$, we write $\|x\|_2$ for the Euclidean norm and $\|x\|_\infty = \max_j |x_j|$ for the max norm. For a square matrix $M \in \mathbb{R}^{d \times d}$, we define the infinity norm as $\|M\|_\infty = \max_i \sum_{j=1}^d |M_{ij}|$, i.e., the maximum absolute row sum.

| Symbol | Meaning / Definition |
|---|---|
| $[m]$ | Index set $\{1, \dots, m\}$ |
| $n$, $d$ | Number of workers and number of tasks, respectively |
| $X_{ij} \in \{-1, +1\}$ | Response given by worker $i$ on task $j$ |
| $y_j \in \{-1, +1\}$ | Unknown ground-truth label of task $j$ |
| $k_j \in \{e, h\}$ | Type of task $j$: either easy $(e)$ or hard $(h)$ |
| $p_{ki}$ | Probability that worker $i$ correctly labels a task of type $k$ |
| $r_{ki} = 2p_{ki} - 1$ | Reliability of worker $i$ on type-$k$ tasks |
| $r_e$, $r_h$ | Reliability vectors for easy and hard tasks, respectively |
| $d_e$, $d_h$ | Number of easy and hard tasks, respectively $(d_e + d_h = d)$ |
| $\alpha$ | Proportion of easy tasks: $d_e = \alpha d$, $d_h = (1 - \alpha)d$ |
| $\rho$ | The amount by which $p_{ki}$ is bounded away from 0 and 1 |
| $\bar{r}$ | Lower bound on average reliability: $\frac{1}{n} \sum_i r_{ki} \geq \bar{r}$ |
| $T = \frac{1}{n} X^\top X$ | Task-similarity matrix |
| $\hat{v}$ | Principal eigenvector of $T$ used for clustering |
| $\hat{\mu} = \frac{1}{d} \sum_j |\hat{v}_j|$ | Threshold for clustering tasks based on $\hat{\boldsymbol{v}}$ |
| $R_y$ | Low rank signal matrix defined in Lemma 1 |
| $\nu(n^{-1} R_y)$ | Normalized eigengap of $R_y$ as defined in Lemma 1 |
| $\hat{r}_{ki}$ | Estimated reliability of worker $i$ for type-$k$ tasks |
| $\eta$ | Fraction of tasks mis-clustered by the algorithm |
| $s$ | Ratio between the distinct entries of the principal eigenvector of $\mathbb{E}[T]$ |
| $D(r_e, r_h, \alpha, d)$ | Problem-dependent term in the perfect clustering bound in Theorem 1 |
| $\Phi_n(r_k)$ | Error exponent for DS-based label estimation on type-$k$ tasks |

Table 1: Summary of notation used throughout the paper. Subscripts $k \in \{e, h\}$ indicate task type when relevant.

## 3.1 Algorithm

We proposed a clustering algorithm for clustering tasks by type from the observation matrix $X$. For the goal of estimating the ground truth label $y_j$ for each task $j \in [d]$, we propose a two-step approach:

1. **Clustering Tasks by Type:** Separate the tasks into two clusters using Algorithm 1.

2. **DS Algorithm for Label Estimation:** Use the following DS model-based algorithm on each of those clusters to estimate the labels within each cluster:

   (a) Use the TE algorithm (see Appendix B for details of TE algorithm) to estimate the reliability vector for each cluster.

   (b) Estimate the labels using the plug-in NP rule as given in Equation 2.

The clustering method described in Algorithm 1 computes the principal eigenvector of the task-similarity matrix $T = n^{-1} X^T X$ denoted as $\hat{v}$. Each task $j$ is then assigned to one of the two clusters by thresholding the magnitude of the $j^{th}$ entry $\hat{v}_j$ with a threshold $\hat{\mu} = \frac{1}{d} \sum_j |\hat{v}_j|$. We adopt the convention that eigenvectors are unit norm.

---

**Algorithm 1** Clustering tasks into hard and easy types

---

**Input:** Worker responses $X \in \{-1, +1\}^{n \times d}$.
Compute the principal eigenvector $\hat{v}$ of the task-similarity matrix $T = n^{-1}X^T X$.
Set threshold $\hat{\mu} = \frac{1}{d}\sum_j |\hat{v}_j|$.
Classify task types by thresholding:
$$\hat{k}_j = \begin{cases} e & \text{if } |\hat{v}_j| \geq \hat{\mu} \\ h & \text{if } |\hat{v}_j| < \hat{\mu}. \end{cases}$$
**Return:** Task type estimates $\hat{k}_1, \ldots, \hat{k}_d$.

---

### 3.2 Clustering

In this sub-section, we analyze the performance of the clustering Algorithm 1. We note that our clustering algorithm only needs to classify tasks into two groups, as long as all the easy tasks fall into one group and all the hard tasks fall into the other group. Later, we will apply the DS-based algorithm separately to each cluster and hence, it does not matter which group we call hard and which group we call easy. Therefore, the clustering error associated with Algorithm 1 can be defined as

$$\eta := \min_{\pi : \{e,h\} \to \{e,h\}} \frac{1}{d}\sum_j \mathbf{1}\left\{\pi(\hat{k}_j) \neq k_j\right\}. \tag{3}$$

We show that the probability of perfectly recovering clusters, i.e. $\eta = 0$, approaches 1 with a rate exponentially fast in $n$. This is precisely stated and shown in Theorem 1.

To understand why task types can be perfectly recovered from clustering, we characterize the spectral properties of the task-similarity matrix $T$. For simplicity of analysis, we re-arrange the tasks such that easy tasks are in the first $d_e$ columns of $X$ and hard tasks are in the remaining columns. Knowing the arrangement of columns implies knowledge of task types, but we only use this to simplify exposition and note that this is not used by our algorithm and does not affect our analysis.

Denote $I_d$ and $1_{a \times b}$ to be the $d \times d$ identity matrix and the all-ones matrix of size $a \times b$, respectively. We first show that the expected task similarity matrix $\mathbb{E}[T] := \mathbb{E}[n^{-1}X^\top X]$ can be decomposed into a sum of low-rank and sparse components.

**Lemma 1** *Define the matrix*

$$n^{-1}R_y$$
$$:= n^{-1}\text{diag}(y)\begin{pmatrix} \|r_e\|_2^2 1_{d_e \times d_e} & r_e^T r_h 1_{d_e \times d_h} \\ r_h^T r_e 1_{d_h \times d_e} & \|r_h\|_2^2 1_{d_h \times d_h} \end{pmatrix}\text{diag}(y)$$

*and a diagonal matrix*

$$S - I_d - \frac{1}{n}\text{diag}\left([\|r_e\|_2^2 1_{1 \times d_e}, \|r_h\|_2^2 1_{1 \times d_h}]^T\right).$$

*The matrix $n^{-1}R_y$ is rank-$\ell$ with $\ell \leq 2$, and its normalized eigen-gap $\nu(n^{-1}R_y) := d^{-1}(\lambda_1(n^{-1}R_y) - \lambda_2(n^{-1}R_y))$ between its two largest eigenvalues $\lambda_1, \lambda_2$ can be expressed as:*

$$\nu(n^{-1}R_y) = \frac{\sqrt{[d_e\|r_e\|_2^2 - d_h\|r_h\|_2^2]^2 + 4d_e d_h (r_e^T r_h)^2}}{nd}. \tag{4}$$

*Further, we have the decomposition*

$$\mathbb{E}[T] = n^{-1}R_y + S. \tag{5}$$

The proof is provided in Appendix E.

Our key motivation for Algorithm 1 is based on the observation in Lemma 2, where the principal eigenvector $v(n^{-1}R_y)$ of the low-rank signal matrix $n^{-1}R_y$ has a special structure. Specifically, there exists a bijection between the magnitudes of entries in the principal eigenvector and task types. The proof follows from the eigendecomposition of matrix $n^{-1}R_y$ and is presented in Appendix E.

**Lemma 2** *Suppose $r_e^\top r_h \neq 0$. Then, the principal eigenvector of the matrix $n^{-1}R_y$ has the following form:*

$$v(n^{-1}R_y) = \text{diag}(y) \begin{bmatrix} \frac{s}{\sqrt{s^2 d_e + d_h}} 1_{d_e \times 1} \\ \frac{1}{\sqrt{s^2 d_e + d_h}} 1_{d_h \times 1} \end{bmatrix} \tag{6}$$

*where*

$$s = \omega + \sqrt{\omega^2 + \frac{d_h}{d_e}} \tag{7}$$

*and*

$$\omega = \frac{d_e \|r_e\|_2^2 - d_h \|r_h\|_2^2}{2 d_e r_e^T r_h}.$$

*In the alternative case that $r_e^\top r_h = 0$, we have that*

$$v(n^{-1}R_y) = \text{diag}(y) \begin{bmatrix} \frac{1}{\sqrt{d_e}} 1_{d_e \times 1} \\ 0_{d_h \times 1} \end{bmatrix}. \tag{8}$$

Denote the distinct magnitudes in $v(n^{-1}R_y)$ corresponding to easy and hard tasks as $\mu_e(n^{-1}R_y)$ and $\mu_h(n^{-1}R_y)$ respectively. It turns out that $\mu_e(n^{-1}R_y) \neq \mu_e(n^{-1}R_y)$ as long as $\|r_e\|_2 \neq \|r_h\|_2$. Consequently under this condition, if we have access to the signal matrix $n^{-1}R_y$, we can differentiate tasks of one type from another by inspecting the entries of the vector $v(n^{-1}R_y)$. Specifically, the task type can be recovered by thresholding the magnitudes of the entries with the average

$$\mu(n^{-1}R_y) = \frac{d_e}{d} \mu_e(n^{-1}R_y) + \frac{d_h}{d} \mu_h(n^{-1}R_y). \tag{9}$$

However, we can only access the principal eigenvector $\hat{v}$ of $T = n^{-1}X^T X$ instead of $v(n^{-1}R_y)$. The following theorem shows that the noisy entries in $\hat{v}$ are sufficiently concentrated to those in $v$ such that the threshold rule applied to $\hat{v}$ perfectly recovers the task types.

**Theorem 1** *Under Assumption 1, if the number of tasks $d$ satisfies*

$$d \geq \frac{C_1}{\sqrt{D(r_e, r_h, \alpha, d)}}, \tag{10}$$

*then Algorithm 1 returns task type estimates such that*

$$P(\eta = 0) \geq 1 - 2d^2 \exp\left(-C_2 n D(r_e, r_h, \alpha, d)\right), \tag{11}$$

*where the problem-dependent quantity $D(r_e, r_h, \alpha, d)$ characterizing the error exponent and the requirement on $d$ is defined as follows:*

$$D(r_e, r_h, \alpha, d) = \begin{cases} \left(\frac{(1-\alpha)^5 \rho}{\alpha} \frac{\nu(n^{-1}R_y)||s|-1|}{\sqrt{s^2+1}}\right)^2 & \text{when, } r_e^\top r_h \neq 0, \\ \left(\frac{(1-\alpha)^5 \rho}{\alpha} \nu(n^{-1}R_y)\right)^2 & \text{when, } r_e^\top r_h = 0 \end{cases} \tag{12}$$

*and $C_1$ and $C_2$ are universal constants, independent of the problem parameters.*

**Remark 1** *Inspecting the problem-dependent exponent $D(r_e, r_h, \alpha, d)$ in Theorem 1 reveals that, when the reliability vectors are not orthogonal, the exponent increases (as $|s|$ is an increasing function of $|\omega|$ and $|s| > 1$ under Assumption 1) with*

$$|\omega| = \frac{\left|d_e \|r_e\|_2^2 - d_h \|r_h\|_2^2\right|}{2 d_e \left|r_e^\top r_h\right|}.$$

*This implies that the probability of perfect clustering improves (i.e., has a better exponent) when there is a large gap in the norms of the reliability vectors and/or the angle between them is large.*

### 3.2.1 $\log(d)$ Workers Suffice for Perfect Clustering

From the above Theorem 1, we show that for achieving a clustering with $\eta = 0$ using Algorithm 1, we only need the number of workers $n$ to be of order $O(\log(d))$ under our model assumptions. From Theorem 1, the requirement on $n$ for the event of perfect clustering with probability $\geq 1 - \delta$ becomes:

$$n \geq \frac{\log\left(\frac{2d^2}{\delta}\right)}{C_2 D(r_e, r_h, \alpha, d)}.$$

The next lemma provides an intuitive condition on the reliability vectors $r_e$ and $r_h$ showing that $O(\log(d))$ workers suffice for perfect clustering.

**Corollary 1** *If there exist some universal constant $\beta$ with $0 < \beta \leq 1$,*

$$\frac{\|r_e\|_2^2 - \|r_h\|_2^2}{n} \geq \beta \tag{13}$$

*then, under Assumption 1, Algorithm 1 achieves clustering error $\eta = 0$ with probability at least $1 - \delta$ when*

$$n \geq \frac{C_6 \alpha^2 \log\left(\frac{2d}{\delta}\right)}{(1-\alpha)^{12} \rho^2 \beta^4} \tag{14}$$

*where $C_6$ is an absolute constant given as $C_6 = \frac{160}{C_2}$.*

Recall from our model assumptions, we indeed have $\|r_e\|_2^2, \|r_h\|^2 = O(n)$. Hence, the above condition in Equation 13 is quite practical. It requires that the normalized norm gap between the reliability vectors is bounded away from zero.

The idea behind proving Corollary 1 is to show that the problem-dependent parameter $D(r_e, r_h, \alpha, d)$ is of order $O(1)$. This is shown in detail in Section F.5.

### 3.2.2 Proof Sketch of Theorem 1

Building upon the above discussion, we give an outline of the key ideas involved in proving Theorem 1 below. The detailed proof of Theorem 1 is given in Appendix F:

1. Recall the structure of $v(n^{-1}R_y)$, the principal eigenvector of the signal matrix $n^{-1}R_y$ from Lemma 2. We prove that the magnitudes of $v(n^{-1}R_y)$ corresponding to different types are separated when $\|r_e\|_2 \neq \|r_h\|_2$. This suggests that under this condition if we had access to $v(n^{-1}R_y)$, then we can cluster tasks by using a threshold to differentiate the magnitudes of the elements of $v(n^{-1}R_y)$. But we do not have access to this eigenvector, therefore the rest of the proof shows that the eigenvector we have access to is a small perturbation of $v(n^{-1}R_y)$.

2. We note that $T = n^{-1}R_y + S + N$, where $N$ is a random matrix noise term given by $N = T - \mathbb{E}[T]$. We use matrix Hoeffding inequality to show that this noise term is small in the infinity-norm sense. This is shown in Lemma 3.

   **Lemma 3** *For any $t > 0$ and any positive values of $n$ and $d$, the task-similarity matrix $T$ concentrates around its expectation as given by the noise matrix concentration follows:*

   $$\mathbb{P}\left(\|N\|_\infty \geq t\right) \leq 2d^2 \exp\left(-\frac{nt^2}{2d^2}\right). \tag{15}$$

   The proof is given in the appendix F.1.

3. Since $S$ is a diagonal matrix, it can be easily shown that its spectral norm is sufficiently small when the number of tasks is large, which is the case in crowdsourcing models. This observation, along with

Lemma 3, implies that the spectral norm of $S + N$ is sufficiently small with high probability. Then, using the result of Fan et al. (2018), we show that the principal eigenvector of the matrix $T$ which is denoted as $\hat{v}$ has a structure similar to that of $v(n^{-1}R_y)$, i.e., $\hat{v}$ is a perturbed version of $v(n^{-1}R_y)$, in the $l_\infty$-norm sense, where the perturbation is small under our model. This is shown in Lemma 4.

**Lemma 4** *If $\nu(n^{-1}R_y)$ satisfies $\frac{C_3(1-\alpha)^4\rho}{\alpha}\nu(n^{-1}R_y)d - 1 > 0$, then, for every $0 < \epsilon < C_3(1 - \alpha)^4\nu(n^{-1}R_y)d - 1$, the event*

$$\min_{\theta \in \{-1,+1\}} \|\theta\hat{v} - v(n^{-1}R_y)\|_\infty \geq \frac{C_4\alpha}{(1-\alpha)^4\rho\nu(n^{-1}R_y)d\sqrt{d}}(\epsilon + 1) \tag{16}$$

*occurs with probability at most $2d^2 \exp\left(-n\frac{\epsilon^2}{2d^2}\right)$ where $C_3$ and $C_4$ are universal positive constants.*

The proof of Lemma 4 is quite involved and is provided in Appendix F.2. Below, we outline the key intuition behind the approach. Let $\zeta$ denote the angle between reliability vectors $r_e$ and $r_h$. Our analysis distinguishes between two regimes:

- **Sufficiently large $\zeta$:** In this case, we approximate the expected task-similarity matrix using a rank-2 signal matrix.
- **Small $\zeta$:** Here, we employ a rank-1 approximation of the task-similarity matrix.

A crucial aspect of our analysis is identifying the transition between these regimes, which requires a careful, structure-aware examination of the task-similarity matrix.

4. The $l_\infty$-norm concentration in Lemma 4 yields a sufficient condition for perfect clustering. In particular, we show that $\|\hat{v} - v(n^{-1}R_y)\|_\infty$ is with high probability, at most $\frac{1}{2}\min(m_e(n^{-1}R_y), m_h(n^{-1}R_y))$, where $m_e(n^{-1}R_y) = |\mu_e(n^{-1}R_y) - \mu(n^{-1}R_y)|$ and $m_h(n^{-1}R_y) = |\mu(n^{-1}R_y) - \mu_h(n^{-1}R_y)|$. A little thought shows that this would imply that all tasks are clustered perfectly.

**Remark 2** *In the above proof sketch, we leveraged the $l_\infty$-norm perturbation of the eigenvectors of the task-similarity matrix, as established in Lemma 4 to derive the perfect clustering result in Theorem 1 where $\log(d)$ order of workers suffice. Next, we discuss why a direct application of the Davis-Kahan theorem (Yu et al., 2014), one of the most commonly used perturbation results in clustering literature, yields vacuous bounds in this context.*

*The Davis-Kahan theorem characterizes eigenvector perturbations as a function of matrix perturbations in the $l_2$-norm of the eigenvectors. However, it is ineffective for obtaining meaningful $l_\infty$-norm bounds on eigenvector perturbations. A standard approach to convert an $l_2$-norm bound into an $l_\infty$-norm bound relies on the inequality $\|x\|_\infty \geq \frac{1}{\sqrt{d}}\|x\|_2$ for a vector $x$ in $\mathbb{R}^d$. This introduces an undesirable $\sqrt{d}$ factor, leading to a requirement that the number of workers must scale polynomially with the number of tasks which is an impractical condition for crowdsourcing applications.*

*This limitation highlights the necessity of a more refined analysis, as developed in our approach, to ensure that perfect clustering is achievable under realistic conditions. Notably, the Davis-Kahan theorem does not exploit any special structure of the matrix that is being perturbed while the result in Fan et al. (2018) allows us to exploit a low-rank structure that we have identified in the crowdsourcing task-similarity matrix.*

### 3.3 How Useful is Clustering for Labeling?

A natural question to ask in this two-type model is how important is the clustering step for the subsequent label estimation. Can one use a weighted majority voting estimate for the labels using a single weight vector across all tasks? The following proposition gives a lower bound on the expected labeling error for such type-agnostic weighted majority voting (TA-WMV) algorithms in this context.

**Proposition 1** *Let the WMV estimate using a single weight vector across all task $j$ is defined as:*

$$\hat{y}_j^{WMV}(w) := \text{sgn}\left(\sum_{i=1}^{n} w_i X_{ij}\right), \forall j \in [d]$$

*for some weight vector w. We consider weight vectors belonging to the set $w_l \leq |w_i| \leq w_u$ for all workers i with $w_l$ and $w_u$ two positive constants such that $0 < w_l \leq w_u < \infty$. Under this construction, for any $y \in \{-1, +1\}^d$, the average labeling error rate for the type-agnostic WMV algorithm can be lower bounded as*

$$\liminf_{n \to \infty} \frac{1}{n} \log \min_w \mathbb{E} \left( \frac{1}{d} \sum_j \mathbf{1} \left( \hat{y}_j^{WMV}(w) \neq y_j \right) \right) \geq - \limsup_{n \to \infty} \max_w \min_k \varphi_n(w, r_k),$$

*for any ground-truth vector $y \in \{-1, +1\}^d$ where the error exponent $\varphi_n(w, r_k)$ is given by*

$$\varphi_n(w, r_k) = - \inf_{t \geq 0} \frac{1}{n} \sum_{i=1}^n \log \left( e^{tw_i} \frac{1 - r_{ki}}{2} + e^{-tw_i} \frac{1 + r_{ki}}{2} \right). \tag{17}$$

The above result is a generalization of Theorem 5.1 in Gao et al. (2016); our proposition uses weighted majority voting for arbitrary weights for a type $k$, whereas their result is for majority voting. The proof of Proposition 1 is given in Appendix G. It's worth noting that the paper Gao & Zhou (2013) has shown lower bounds on labeling performance for a two-type model for these two algorithms: projected expected maximization and majority voting (Theorems 4.2 and 4.3 in Gao & Zhou (2013)). Compared to them, we have a lower bound on the performance of labeling for the weighted majority voting.

To understand the limitation of TA-WMV algorithms, it is instructive to compare the error rates in Proposition 1 with the achievable rates by an algorithm that accounts for type difference among different tasks under the setting when task types are known but the reliability vectors $(r_e, r_h)$ are unknown.

**Proposition 2** *Assume $V_k = \min_i \max_{a,b \neq i} \sqrt{|r_{ka} r_{kb}|} > 0$ for each $k \in \{e, h\}$ which is satisfied if there are at least two workers with non-zero reliability values for each type. If the number of workers n satisfies $n \geq \sqrt{3\rho/\bar{r}}$, and the number of tasks per type satisfies*

$$d_k \geq C_5 \frac{n^2}{V_k^4 \min(\rho^2, \bar{r}^2)} \left( n \Phi_n(r_k) + \log(6n^2) \right). \tag{18}$$

*for some universal constant $C_5$ then, the TE algorithm to estimate the reliability vectors followed by NP-WMV for label estimation separately for each type (when type information is known) achieves a labeling error rate satisfying*

$$\mathbb{E} \left( \frac{1}{d} \sum_j \mathbf{1} \left( \hat{y}_j \neq y_j \right) \right) \leq 3 \sum_{k \in \{e,h\}} \frac{d_k}{d} \exp \left( -n \Phi_n(r_k) \right),$$

*where $\hat{y}_j$ and $y_j$ are the estimated and true labels of task j, respectively, and*

$$\Phi_n(r_k) = - \frac{1}{n} \sum_{i=1}^n \log \left( \sqrt{(1 + r_{ki})(1 - r_{ki})} \right). \tag{19}$$

The error exponent Equation 19[1] for type-dependent weighted majority voting can be related to the error exponent for the type-agnostic weighted majoring voting in Equation 17 through the identity

$$\Phi_n(r_k) = \max_w \varphi_n(w, r_k).$$

Recall from Proposition 1, the lower bound on the error exponent for type-agnostic Weighted Majority Vote is $\max_w \min_k \varphi(w, r_k)$ and from the definition of $\Phi_n(r_k)$, it is clear that $\max_w \min_k \varphi(w, r_k) \leq \Phi_n(r_k), \forall k \in \{e, h\}$. It is easy to see that, in most cases, the inequality is strict for both $k \in \{e, h\}$. Therefore, TA-WMV is strictly worse than WMV applied to each task type separately. The proof of Proposition 2 is provided in the Appendix H.

---

[1]This error exponent also serves as the asymptotic lower bound for the labeling error for a one-coin DS model corresponding to a reliability type $r_k$ (Gao et al., 2016).

### 3.3.1 Labeling Error Guarantee for Our Two-Step Approach

By combining the perfect clustering result from Theorem 1 with the labeling error guarantee for the known-type case from Proposition 2, we immediately obtain the labeling error guarantee for our two-step approach. This approach consists of: (1) clustering tasks by type using Algorithm 1, and (2) applying the DS-based algorithm TE with NP-WMV for label estimation within each cluster. For completeness, the labeling guarantee of this two-step approach is provided in Theorem 2 in the Appendix D with its proof in Appendix I.

## 4 Experiments

In this paper, we present experiments with real-world datasets, pseudo-real datasets, and synthetic datasets to supplement the theory presented in the previous sections. By pseudo-real datasets, we mean the following: some real-world sets do not contain all the information we need to run our experiments and therefore, we generate some of the data we need using the available data in the datasets. In such cases, we will explain how we filled in the required data.

1. First, we compare our two-step algorithm (clustering tasks and then applying a DS algorithm to each type of task) with a single-step DS algorithm (i.e., applying a DS algorithm to all the tasks). Our experiments clearly show the benefit of clustering. Although our theoretical analysis primarily employs TE followed by WMV with NP-weights as the Dawid-Skene (DS) algorithm, we also compare DS algorithms with and without clustering across various other DS algorithms to demonstrate the benefits of clustering: unweighted majority vote (MV), ratio of eigenvectors (ER, Dalvi et al. 2013), TE (Bonald & Combes (2017)), and Plug-in gradient descent (PGD, Ma et al. 2022). A large number of algorithms have been proposed for crowdsourcing including Spectral-EM (Zhang et al. (2016)), and message-passing (Karger et al. (2014a)) to name just a few. Exhaustively comparing with all the algorithms is difficult, so we have chosen to compare our algorithm to ER, TE, and PGD for the following reason: many algorithms have been compared in Dalvi et al. (2013), Bonald & Combes (2017) and Ma et al. (2022), where it was shown that ER, TE, and PGD consistently out-perform other algorithms.

2. Next, we compare our algorithm with other algorithms that also consider tasks of different types. We demonstrate that our algorithm performs better on the datasets considered.

The datasets we used for our experiments are the following:

1. Two of the real-world datasets we used which are called the "Bluebird" (Welinder et al., 2010) and "HC-TREC" (Buckley et al., 2010) are complete datasets, i.e., the response matrix has no missing entry.

2. Three other real-world datasets, "Dog" (Deng et al., 2009), "Temp" (Snow et al., 2008), and "RTE" (Snow et al., 2008) are sparse datasets that do not provide responses corresponding to all worker-task pairs as in our motivating example in the introduction. To handle this, for the "Dog" dataset that contains 4 classes, we converted it to binary groups $\{0, 2\}$ vs. $\{1, 3\}$ following Bonald & Combes (2017). Then we calculate the fraction of correct labels (given by workers) for each task based on the ground truth and the available responses and classify half of them (the half with the most accurate worker responses) as easy tasks and the rest as hard tasks. Then, we estimate the empirical reliabilities of the workers for each type of task and use this to generate synthetic entries for the missing worker-task pairs in the response matrix. Similar treatments for the no-response entries are done for, "RTE" and "Temp", each of which contains binary truth values. The number of workers and tasks for all five datasets ("Bluebird", "HC-TREC", "Dog", "Temp", and "RTE") are provided in Table 2.

3. Obtaining real-world crowdsourcing datasets for healthcare examples that we mention in the Introduction is difficult due to privacy reasons. With the limited information available from a radiology dataset, we created a synthetic dataset and we report the results from the dataset in Appendix C.1.

Table 2: Dataset Descriptions

| Dataset | # Workers | # Tasks |
|---------|-----------|---------|
| Bluebird | 39 | 108 |
| Dog | 78 | 807 |
| RTE | 164 | 800 |
| HC-TREC | 10 | 1000 |
| Temp | 76 | 462 |

Table 3: Label estimation errors for different crowdsourced datasets. "TA" and "C" indicate that labels were estimated without (type-agnostic) or with clustering.

| Dataset | MV | ER | TE | PGD |
|---------|-----|-----|-----|-----|
| Bluebird-TA | 24.07 | 27.78 | 17.59 | 25.93 |
| Bluebird-C | 24.07 | 11.11 | 12.96 | 12.96 |
| Gain | 0.00 | 16.67 | 4.63 | 12.97 |
| Dog-TA | 26.15 | 19.85 | 13.64 | 19.01 |
| Dog-C | 26.15 | 0.78 | 12.23 | 20.56 |
| Gain | 0.00 | 19.07 | 1.41 | -1.64 |
| HC-TREC-TA | 33.70 | 68.80 | 67.30 | 30.80 |
| HC-TREC-C | 33.70 | 40.90 | 30.60 | 30.80 |
| Gain | 0.00 | 27.90 | 36.6 | 0.00 |

**Comparing with Traditional DS Algorithms:** We observe that clustering improves performance in the dataset considered. In the case of RTE and Temp datasets, with or without clustering, the accuracy of label estimation is 100%, which is why we did not include them in Table 3. Hence, our results show that clustering does not hurt the accuracy even in cases where it may not be required.

**Comparison with Task-Specific Reliability Models:** As discussed in the related work section, several previous papers address models with multiple types of tasks and use different task-specific reliability models to infer task labels. Notable works include Khetan & Oh (2016), Shah et al. (2021), Shah & Lee (2018), Kim et al. (2024) to name a few. The model in Khetan & Oh (2016) assumes that $\mathbb{E}[T]$ is a rank-1 matrix. Clearly, this is not true if there is more than one type of task. The algorithm in Shah et al. (2021) involves a large number of parameters, leading to very poor performance on the datasets we used, therefore we are not comparing it with our model. Thus, we restrict the comparison of our algorithm to those in Shah & Lee (2018) and Kim et al. (2024).

Table 4: Comparison of our approach with Task-specific Reliability Models. 'TE-C' is our two-step approach - clustering followed by TE-WMV.

| Dataset | TE-C | SDP | SS |
|---------|------|------|------|
| Bluebird | 12.96 | 24.81 | 22.62 |
| Dog | 12.23 | 34.56 | 51.70 |
| TREC | 30.6 | 38.22 | 49.39 |
| Temp | 0 | 1.93 | 50.35 |

In Table 4, columns "TE-C", "SDP" and "SS" correspond to our two-step approach, SDP-based algorithm in Kim et al. (2024) and SS algorithm from Shah & Lee (2018), respectively. We used the MATLAB code provided by the authors in Kim et al. (2024) for running different 'SDP' and 'SS' algorithms and listed the error minimized over the input parameter the number of specializations from $\{2, 3, 4\}$ . We see that our algorithm outperforms SDP and SS in the datasets considered above.

**Runtime Comparison:** We report the runtime performance of our proposed two-step algorithm to illustrate its practicality. In particular, we compare the type-agnostic DS-based TE algorithm (denoted **TE-TA**) with our type-dependent pipeline (denoted **TE-C**), which consists of spectral clustering followed by TE and NP-WMV applied separately to each task cluster.

The clustering step is dominated by the computation of the principal eigenvector of the task-similarity matrix of size $d \times d$, which can be performed in $\mathcal{O}(d^2)$ time using power iteration. The TE algorithm (see Appendix B) is dominated by the computation of the worker-similarity matrix and can be implemented in $\mathcal{O}(n^2 d)$ time.

Table 5 presents the average runtime (in seconds) of the algorithms on the three real-world datasets used in our experiments. All runtimes were measured on a standard CPU using Google Colab. We observe that the overhead due to clustering is negligible in practice. We observe that the clustering runtime increases from Bluebird to HC-TREC, consistent with its polynomial dependence on the number of tasks $d$. In contrast, the TE labeling step, which scales as $\mathcal{O}(n^2 d)$, is fastest on HC-TREC due to its small number of workers ($n = 10$), and slowest on Dog, which has the largest $n = 78$.

| Dataset (workers, tasks) | Clustering Step | TE-TA (DS-based) | TE-C (Our approach) |
|---|---|---|---|
| Bluebird (39, 108) | 0.0998 | 0.0318 | 0.1280 |
| Dog (78, 807) | 0.2217 | 0.2064 | 0.5935 |
| HC-TREC (10, 1000) | 0.2927 | 0.0212 | 0.3207 |

Table 5: Runtime comparison (in seconds) of TE algorithms with and without clustering.

For comparison with other type-dependent algorithms, we used the MATLAB implementation of the SDP-based method proposed by Kim et al. (2024). Due to the complexity of parameter tuning and the reliance on a semi-definite program solver, the runtime for this method is substantially higher, ranging from several minutes to over 30 minutes on a standard laptop with 16GB RAM.

**A broader question:** A broader question in crowdsourcing, beyond the scope of this paper, is assessing the validity of the DS model and its extensions in a data-driven manner for a given dataset. In Appendix C.2, we present a dataset where plain majority voting outperforms weighted majority voting. This suggests that the DS model or its variants may not be suitable in such cases, either because the underlying mathematical assumptions do not hold or there is insufficient data to accurately estimate worker reliabilities.

# 5 Discussion: Extensions and Practical Implications

The analysis in this paper focuses on a two-type crowdsourcing model where each worker labels every task. The main contribution is identifying a separability condition for perfect clustering, which implies that $O(\log(d))$ workers suffice for perfect clustering (1). In this section, we discuss possible extensions of our algorithm to more general settings and the practical implications of our work.

**Sparse response matrices.** Unlike some datasets considered in the paper (e.g., "Blue-bird", "HC-TREC"), many practical datasets yield sparse response matrices. Real-world datasets with sparse responses examined here include "Dog", "Temp", and "RTE". To fit these into our model, we preprocess the sparse response matrices by imputing missing entries using only the observed labels (without access to ground truth), consistent with common approaches in the literature. We then evaluate the performance of our clustering-based pipeline against existing algorithms designed for multi-type crowdsourcing. Extending the theoretical analysis to sparse response matrices is an important direction for future work.

**Extension to more than two types of tasks and when clustering is beneficial.** Our algorithm can, in principle, be extended to more than two types (e.g., via $k$-means clustering on the top eigenvectors). However, accurate estimation in such settings requires that type-specific reliability vectors be sufficiently separated and that each type has enough labeled examples. The algorithmic pipeline presented in this work is most useful when worker skills vary by task type. Heuristically, a clear eigengap after the first two components

can be used as a diagnostic indicator of multi-type structure. If no eigengap is present, a type-agnostic aggregator (e.g., DS) is a reasonable fallback. Extending the theoretical guarantees to more than two types remains an important direction for future work.

## 6 Conclusion

We considered a crowdsourcing model which is more appropriate than the Dawid-Skene model when there are tasks that require different levels of skill sets. Then we described a spectral clustering algorithm that clusters tasks by difficulty and analyzed its performance to characterize the condition for perfect clustering. Experiments with real-life datasets demonstrate the benefits of in label estimation when combined with TE and NP-WMV for each task type separately.

An intriguing direction for future research is extending our clustering approach to models with more than two task types. While our algorithm naturally generalizes to multiple types—by applying k-means clustering to entries corresponding to dominant eigenvectors—accurate label estimation in such settings requires sufficiently distinct reliability vectors across task types. Moreover, estimating multiple reliability vectors introduces additional data requirements. Though our focus in this work has been on the two-type setting, the extension to more than two types presents rich theoretical challenges, making it a promising avenue for further exploration.

### Broader Impact Statement

In accordance with the TMLR guidelines on potential societal impacts and ethical conduct, the authors are not aware of any direct or indirect negative implications of this work. The proposed method may help reduce labeling costs in crowdsourcing by enabling more accurate label aggregation in heterogeneous task settings. Our findings suggest that two task-type modeling approach captures meaningful structure in real datasets, and may serve as a foundation for more general task-type inference in future work.

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

## A  Contents: Appendix Sections

The appendix sections are organized as follows:

1. In Appendix B, a detail description of a DS algorithm is provided. The algorithm we discussed is TE followed by NP-WMV following the discussion in the related work: Dawid-Skene model from the main paper.

2. In Appendix C.3, we plotted the eigenspectram of the datasets used in the experiments. The description of the datasets is given in Table 2 in the main paper. The idea here is to give an intuition on the benefit of clustering.

3. In Appendix C.1, we provide some additional experiments. Here we synthetically generate different datasets from the meta-data available from a radiology database.

4. In Appendix D, we provide a performance guarantee (Theorem 2) on the label estimation of our two-step approach described in the main paper: clustering by Algorithm 1 plus label estimation using TE followed by NP-WMV.

5. Appendix E establishes the spectral properties of the signal matrix $n^{-1}R_y$ which serves as a key motivation for our analysis in the main paper. It also proves Lemma 1 and Lemma 2.

6. Appendix F proves the main result in the paper: Theorem 1.

7. The proof of the Proposition 2 is given in Appendix H.

8. We provide the proof of the Theorem 2 in Appendix I.

9. In Appendix G, we prove Proposition 1 of the main paper.

## B  Detail Description of DS Algorithms: TE and NP-WMV

In this section, we provide a brief description of the DS-based algorithm used in our experiments for label estimation for each task type after separating the tasks into different clusters according to their types. It consists of two steps: first, estimate the reliability vector and then use a weighted majority vote (WMV) algorithm for estimating task tasks. We will review the WMV algorithm first. Consider the Dawid-Skene model so that the distribution of the binary worker response matrix $X \in \{-1, +1\}^{n \times d}$ is determined by a single reliability vector $r \in [-1, +1]^n$, i.e. all tasks are of the same type. Given *known* reliabilities $r$ and focusing on a single task with worker responses $x = (x_1, \ldots, x_n)$, the maximum likelihood decision rule for a given task $j$ is then given by the map

$$g^*(x) = \text{sgn}\left(\sum_{i=1}^{n} w_i x_i\right), \tag{20}$$

with (possibly infinite) weights

$$w_i = \log \frac{1 + r_i}{1 - r_i}. \tag{21}$$

Based on this observation, a common approach is to estimate the reliability vector $r$ from the responses $X$, denoted as $\hat{r}$, and use the Nitzan-Paroush decision rule (Nitzan & Paroush, 1983) to infer the labels as

$$\hat{y}_j^{NP} = \text{sgn}\left(\sum_{i=1}^{n} \log \frac{1 + \hat{r}_i}{1 - \hat{r}_i} X_{ij}\right), \forall j \in [d].$$

Equation 2 corresponds to a *weighted majority vote* of the form Equation 20 with weights $w_i = \log \frac{1+\hat{r}_i}{1-\hat{r}_i}$.

Next, we review the TE algorithm for estimating reliabilities proposed in Bonald & Combes (2017), which we will use in our theoretical results. The reason we focus on this algorithm is that it has been compared to

other algorithms and shown to perform better in real datasets. Additionally, by comparing the probability of labeling error expression derived from Bonald & Combes (2017) with the lower bounds in Gao et al. (2016), it can be seen that the algorithm is provably asymptotically optimal. We give a brief description of the TE algorithm for completeness. The TE algorithm designed for estimating a reliability vector for the DS model first computes the worker-covariance matrix

$$W_{ab} = \frac{1}{d} \sum_{j=1}^{d} X_{aj} X_{bj}, \forall a, b \in [n].$$

For every worker $i \in [n]$, the most informative pair of co-workers $\arg\max_{a,b \in [n]: a \neq b \neq i} |W_{ab}|$ denoted by $(a_i, b_i)$ is computed, and the magnitude of the $i$th worker's reliability is estimated as

$$|\hat{r}_i| = \begin{cases} \left[ \sqrt{\left| \frac{W_{a_i i} W_{b_i i}}{W_{a_i b_i}} \right|} \right]_{[2\rho-1, 1-2\rho]} & \text{if } |W_{a_i b_i}| > 0 \\ 0 & \text{else} \end{cases}. \tag{22}$$

The sign of $\hat{r}_i$ is estimated by letting

$$i^* = \arg\max_{i \in [n]} \left| \hat{r}_i^2 + \sum_{j \in [n]: j \neq i} W_{ji} \right|.$$

and by setting the sign of $\hat{r}$ according to

$$\text{sgn}(\hat{r}_i) = \begin{cases} \text{sgn}\left( \hat{r}_{i^*}^2 + \sum_{j \in [n]: j \neq i^*} W_{ji^*} \right) & \text{if } i = i^* \\ \text{sgn}\left( \hat{r}_{i^*} W_{ii^*} \right) & \text{else} \end{cases}.$$

This concludes our discussion of the TE algorithm.

## C  Additional Experiments :

### C.1  Synthetically Generated Radiology Data

Obtaining real-world datasets for healthcare examples mentioned in the Introduction is difficult. Due to privacy reasons, such datasets do not contain much of the information we require, including ground truths and responses. Nevertheless, we considered one radiology dataset: the Japanese Society of Radiological Technology (JSRT) Database and its report (Shiraishi et al., 2000) to conduct a synthetic experiment. These datasets only contain information about the reliability of the doctors who looked at the data. In other words, this dataset only provides a range of realistic reliabilities, but we had to generate synthetic ground truths and response matrices.

### C.1.1  Setup

In this subsection, we describe how we generate our synthetic datasets from the JSRT report in Shiraishi et al. (2000). The JSRT report contains the performance of 20 radiologists for identifying solitary pulmonary

Table 6: JSRT dataset. Size is in millimeters, and a subtlety of 0 indicates that a nodular pattern is absent.

| Subtlety | 0 | 1 | 2 | 3 | 4 | 5 |
|---|---|---|---|---|---|---|
| Count | 93 | 25 | 29 | 50 | 38 | 12 |
| Size | 0.0 | 23.0 | 17.9 | 17.2 | 16.4 | 14.6 |
| Mean sensitivity (accuracy) of experts | 80.9 | 99.6 | 92.6 | 75.7 | 54.7 | 29.6 |

nodules in chest radiographs. Its dataset statistics are summarized in Table 6. Expert performances are

reported for various levels of subtlety defined by the size of nodular patterns. It is clear that detecting nodular patterns becomes significantly more difficult as the size is decreased, demonstrating a multi-type phenomenon with varying levels of task difficulty. Our setup for the JSRT experiments is given as follows. There is a total of 6 types according to the mean sensitivity reported across all radiologists for 6 different subtlety levels. These values are used as the accuracy for each type as described next.

1. For the JSTR-6 data, we use the reported means and standard deviations of sensitivities of a type $k' \in [6]$: $(\tilde{r}_{k'}, \sigma_{k'})$ as: for each type, we sample the probability parameter for each worker $i$, $p_{k'i}$ as a sample from the uniform distribution with support $\tilde{r}_{k'} \pm \sigma_{k'}$. Then we set $r_{k'i} = \frac{p_{k'i}+1}{2}$.

2. To get an easy-hard model from this, we generate the dataset JSRT-2. Here, we combine the higher and lower 3 accuracy parameters: for the easy type, the sensitivity is estimated as having a mean of $\frac{1}{3}\sum_{k'=1}^{3}\tilde{r}_{k'}$ and standard deviation as the root mean square of the standard deviation of the first three subtlety levels. The parameters for the hard types are generated similarly from the next 3 subtlety levels.

Each truth value $y_j$ is drawn randomly from its class distribution defined by the sample mean of positive (presence of nodules) cases. We then sample the crowd's response following the number of tasks per type in Table 6.

### C.1.2 Results

Table 7: Label estimation errors (%) for the JSRT experiments. "TA" and "C" after dataset names indicate whether label estimation was performed without (type-agnostic) or with clustering, respectively.

| Dataset | MV | ER | TE | PGD |
|---|---|---|---|---|
| JSRT-2-TA | 5.65 | 5.65 | 4.74 | 5.06 |
| JSRT-2-C | 5.65 | 4.39 | 3.16 | 3.81 |
| Gain | 0.00 | 1.26 | 1.58 | 1.25 |
| JSRT-6-TA | 10.30 | 10.30 | 9.96 | 9.72 |
| JSRT-6-C | 10.30 | 10.02 | 9.84 | 9.76 |
| Gain | 0.00 | 0.28 | 0.12 | -0.04 |

The performance of crowdsourcing algorithms with and without our clustering algorithm on the JSRT-6 and JSRT-2 datasets is shown in Table 7. As shown, separation consistently increases accuracy over Dawid-Skene algorithms. Because experts labeled the JSRT dataset, we observe a high accuracy using the simple majority vote. However, failing to identify nodules can be consequential and even a small gain in accuracy is critical.

### C.2 An Example of Majority Voting Performing Better than Weighted Majority Voting

Working with the "Duck" dataset (Welinder et al., 2010), we observed an interesting phenomenon: majority voting outperforms weighted majority voting when using existing algorithms for this dataset. The label estimation errors of the various algorithms considered in this paper for this dataset are presented in Table 8. The plain majority voting is denoted as "MV-TA" and the weighted majority voting algorithms without clustering are "ER-TA", "TE-TA" and "PGD-TA". The "Duck" dataset consists of 53 workers labeling 240 tasks. To align this dataset with the framework used in this paper, we handle missing entries similarly to other datasets such as "Dog", "Temp", and "RTE". Specifically, we compute the fraction of correct labels provided by workers for each task based on ground truth and available responses. We then classify the half of tasks with the most accurate worker responses as easy tasks and the rest as hard tasks. Using this classification, we estimate the empirical reliabilities of workers for each task type and generate synthetic entries for the missing worker-task pairs in the response matrix.

As discussed in the Experiment Section 4, this finding suggests that the DS model and its variants may not be applicable in certain cases, either because the underlying mathematical assumptions do not hold or due to insufficient data to accurately estimate worker reliabilities. From Table 8, we observe that the SDP-based

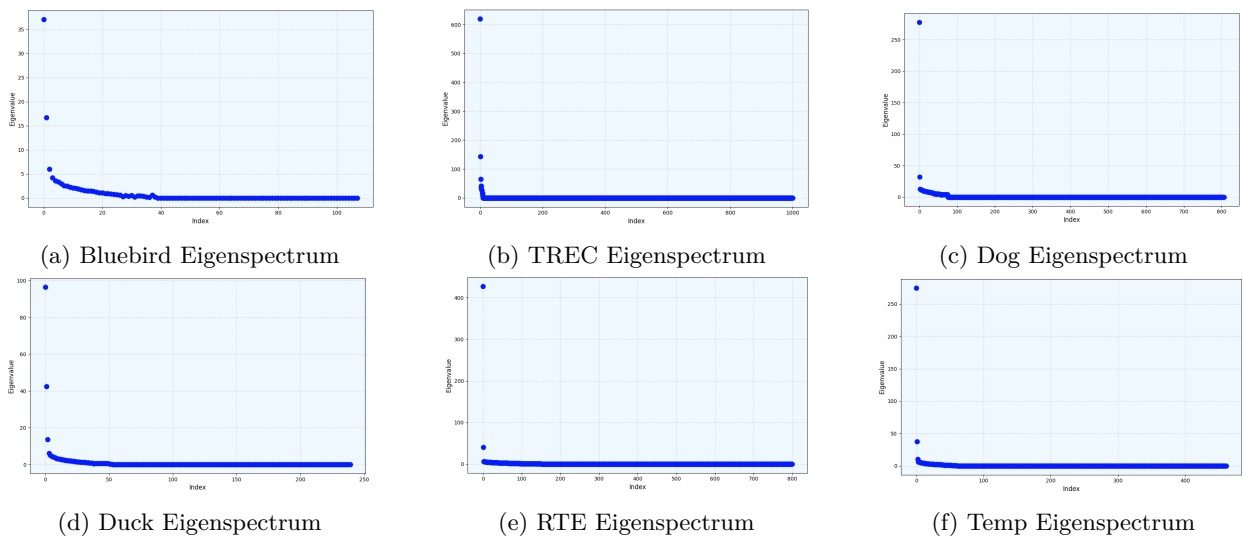

Figure 1: Eigenspectrum of $T$ for different datasets: (a) Bluebird, (b) TREC, (c) Dog, (d) Duck, (e) RTE, and (f) Temp. For each plot, the y-axis represents the eigenvalues, and the x-axis represents the corresponding index of each eigenvalue.

algorithm outperforms all other methods, with TE with clustering and plain majority voting coming in second and third, respectively. Notably, the SDP algorithm clusters workers and tasks separately and then applies plain majority voting for label estimation. On the other hand, the main approach in the paper: TE with clustering uses weighted majority voting based on the reliability estimation by the TE algorithm.

These results highlight an open question in crowdsourcing: how can we determine when majority voting outperforms weighted majority voting? A data-driven approach to this decision could improve label aggregation in cases where standard models like DS may not apply.

Table 8: Label estimation errors for the "Duck" dataset using different algorithms. "-TA" and "-C" indicate that labels were estimated without (type-agnostic) or with clustering. Algorithms compared are: unweighted majority vote (MV), ratio of eigenvectors (ER, Dalvi et al. 2013), TE (Bonald & Combes (2017)), and Plug-in gradient descent (PGD, Ma et al. 2022), SDP-based algorithm in (Kim et al., 2024)(SDP) and SS algorithm from (Shah & Lee, 2018)(SS), respectively

| MV-TA | MV-C | ER-TA | ER-C | TE-TA | TE-C | PGD-TA | PGD-C | SDP | SS |
|-------|------|-------|------|-------|------|--------|-------|-----|-----|
| 32.58 | 32.58 | 59.37 | 24.33 | 41.04 | 41.67 | 38.96 | 32.58 | 19.88 | 56.58 |

## C.3 Eigenspectrum of Task-Similarity Matrix in the Datasets Used for Experiments

In the Table 3 of the main draft, we have seen that clustering improves performance if used before a DS-based algorithm. To get an intuition of why this is the case, we plotted the eigenspectrum of the matrix $T$ in Figure 1. As we can see, all the datasets exhibit at least two eigenvalues which are larger than the rest of them which are close to zero, thus indicating that there is more than one type of task. Therefore, clustering helps to separate tasks by their reliabilities.

## C.4 Synthetic Validation of the Clustering Performance

We simulate a controllable synthetic setting to verify the clustering bound in Theorem 1.

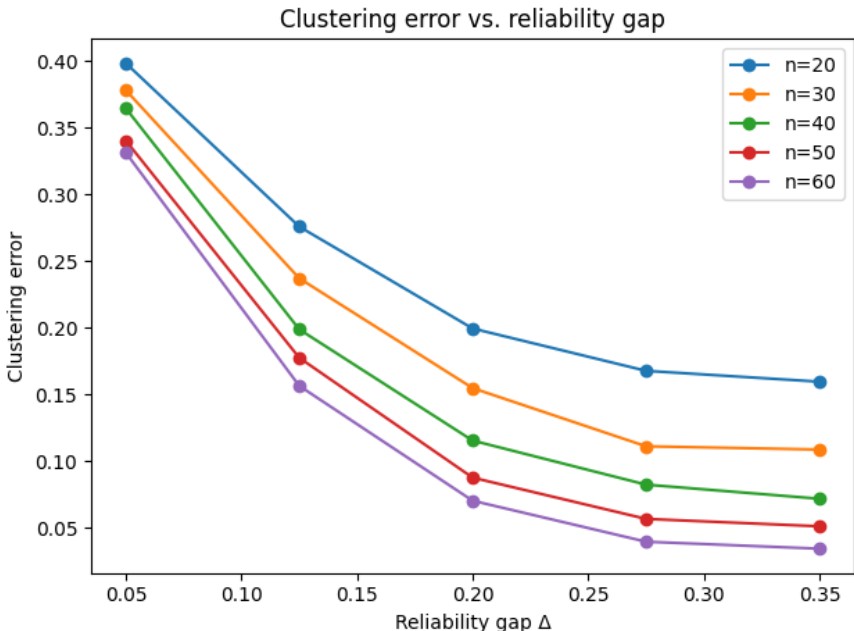

Figure 2: Clustering error on synthetic data versus reliability gap $\Delta$.

**Setup.** We create $d = 1000$ tasks (600 easy, 400 hard) and vary the number of workers $n \in \{20, 30, 40, 50, 60\}$. For each worker $i$ we draw a base accuracy $q_i \sim \text{Unif}[0.6, 0.9]$ and set

$$r_{e,i} = 2q_i - 1, \qquad r_{h,i} = 2 \max\{q_i - \Delta, 0.51\} - 1,$$

where the reliability gap $\Delta \in \{0.05, 0.10, 0.15, 0.20, 0.25, 0.30, 0.35\}$ simultaneously controls the $\ell_2$-norm gap and the angle between $r_e$ and $r_h$. Responses are generated via $X_{ij} = y_j$ with probability $(r_{k,i} + 1)/2$, else $-y_j$, with $y_j \overset{\text{iid}}{\sim}$ Rademacher. We apply the clustering step of Algorithm 1 and report the fraction of mis-clustered tasks, averaged over five independent trials.

**Results.** Figure 2 shows the average clustering error versus $\Delta$. Error decreases (i) as the reliability gap widens and (ii) as the worker count rises.

## D  Label Estimation for Hard-Easy Tasks

In this section, we provide a performance guarantee of the overall clustering plus label estimation in terms of the expected labeling error. If we denote $\hat{y}$ as the label estimation in our approach, then the expected labeling error is defined as: $\mathbb{E}\left(\frac{1}{d} \sum_j \mathbf{1}(\hat{y}_j \neq y_j)\right)$. Before giving an upper bound on the expected labeling error by our overall algorithm, few quantities and notations are to be introduced as follows.

After having divided the tasks into two clusters, in practice, one can simply apply a DS algorithm, such as TE, to each task type separately. However, analyzing such an algorithm is difficult because the clustering step and label estimation steps are correlated due to the fact that we use the same dataset for both. Therefore, as is common in the literature (see Shah et al. (2021), for example), we split the $n$ workers into two disjoint groups and use the responses of one group for clustering and the other group for label estimation. We present these details next.

For the following analysis, let $\mathcal{N}_{cl}$ be the set of workers used for clustering, and define $\mathcal{N}_{rl} = [n] - \mathcal{N}_{cl}$ to be the set of workers that is used for reliability estimation as well as label estimation. Let the responses of the workers in the set $\mathcal{N}_{cl}$ be denoted by

$$X_{cl} := (X_{ij} : (i,j) \in \mathcal{N}_{cl} \times [d]),$$

and the worker responses of the set $\mathcal{N}_{rl}$ be

$$\mathrm{X}_{rl} := (X_{ij} : (i,j) \in \mathcal{N}_{rl} \times [d]).$$

We cluster the tasks in $\mathrm{X}_{cl}$ using Algorithm 1 (with the substitution $X = \mathrm{X}_{cl}$) resulting in the following type assignment for all task $j \in [d]$:

$$\mathcal{T}_k = \left\{ j \in [d] : \hat{k}_j = k \right\}, k \in \{e, h\}.$$

We then use the TE algorithm to estimate reliabilities $\hat{r}_k = (\hat{r}_{ki} : i \in \mathcal{N}_{rl})$ from the responses $(X_{ij} : (i,j) \in \mathcal{N}_{rl} \times \mathcal{T}_k)$ for each $k$. Lastly, the labels $y_j$ are estimated using the NP decision rule

$$\hat{y}_j^{TE} = \mathrm{sgn} \left( \sum_{i \in \mathcal{N}_{rl}} \log \frac{1 + \hat{r}_{\hat{k}_j i}}{1 - \hat{r}_{\hat{k}_j i}} X_{ij} \right). \tag{23}$$

Now we are ready to present the theorem characterizing the accuracy of our combined clustering and label estimation algorithm. Let $n_{cl}$ and $n_{rl}$ be the number of workers in the sets $\mathcal{N}_{cl}$ and $\mathcal{N}_{rl}$, respectively. Let $r_k(\mathcal{N}_{cl})$ and $r_k(\mathcal{N}_{rl})$ be the reliability vector associated with each task type $k$ for the set of workers $\mathcal{N}_{cl}$ and $\mathcal{N}_{rl}$, respectively.

**Theorem 2** *Suppose $(n_{rl}, d_e, d_h, r_k(\mathrm{N}_{rl}))$ satisfy the conditions stated in Proposition 2 and $(n_{cl}, d_e, d_h, r_k(\mathcal{N}_{cl}))$ satisfy the conditions from Theorem 1. Then, for the hard-easy crowdsourcing model under Assumption 1, the labels $\hat{y}$ estimated using Equation 23 satisfy*

$$\mathbb{E} \left( \frac{1}{d} \sum_j \mathbf{1}\left( \hat{y}_j \neq y_j \right) \right)$$

$$\leq 3 \left[ \sum_{k \in \{e,h\}} \frac{d_k}{d} \exp\left( -n_{rl} \Phi_{k,\mathcal{N}_{rl}} \right) \right] + 2d^2 \exp\left( -C_2 n_{cl} D(r_e(\mathcal{N}_{cl}), r_h(\mathcal{N}_{cl}), \alpha, d) \right)$$

*where $\Phi_{k,\mathcal{N}_{rl}} := \Phi_{\mathcal{N}_{rl}}(r_k(\mathcal{N}_{rl}))$ and $D(r_e(\mathcal{N}_{cl}), r_h(\mathcal{N}_{cl}), \alpha, d)$ is defined similarly to $D(r_e, r_h, \alpha, d)$ in Equation 12, with the obvious changes to account for the fact that we are only using the reduced dataset $\mathrm{X}_{cl}$ for clustering. Here, $C_2$ is the same positive universal constant as in the Theorem 1 in the main paper.*

The proof of Theorem 2 is an immediate application of Theorem 1 and is provided in Appendix I.

## E  Spectral Properties of the Expected Task-Similarity Matrix

In this section, we establish the spectral properties of the signal matrix $n^{-1} R_y$ and give the proofs to the Lemma 1 and Lemma 2.

Given the ordered response matrix $X$ we consider in Section 3.2, where the easy and hard tasks are listed consecutively in the columns, the true response matrix with arbitrary task ordering is obtained by a column permutation of $X$. It is easy to see that the ordered task-similarity matrix $T = n^{-1} X^T X$ is then related to the true task-similarity matrix with type-permutations by a similarity transform. All eigenvalues and eigen-spectrum are therefore related by the same permutations, and as long as the algorithm does not utilize an unknown prior on the ordering of these types, its analysis still pertains to the un-ordered case.

Recall the decomposition of the expected task-similarity matrix $\mathbb{E}[T]$ into

$$\mathbb{E}[T] = \underbrace{n^{-1} \mathrm{diag}(y) \begin{pmatrix} \|r_e\|_2^2 1_{d_e \times d_e} & r_e^T r_h 1_{d_e \times d_h} \\ r_h^T r_e 1_{d_h \times d_e} & \|r_h\|_2^2 1_{d_h \times d_h} \end{pmatrix} \mathrm{diag}(y)}_{n^{-1} R_y} \underbrace{-n^{-1} \mathrm{diag}\left( [\|r_e\|_2^2 1_{1 \times d_e}, \|r_h\|_2^2 1_{1 \times d_h}]^T \right) + I_d}_{S}$$

$$= n^{-1} R_y + S, \tag{24}$$

where $S$ is a diagonal matrix. First, we prove the above decomposition by analyzing the entries of the expected task-similarity matrix $\mathbb{E}[T]$. From the definition, $T = n^{-1}O^{\top}O$. That is for all $j_1 \in [d], j_2 \in [d]$, we have, $T_{j_1 j_2} = n^{-1}\sum_{i=1}^{n} O_{ij_1}O_{ij_2}$. Now, from the probabilistic crowdsourcing model considered in Section 2.1, if $j \in [d_e]$, then, $\mathbb{E}[O_{ij}] = r_{ei}y_j$. Similarly, if $j \notin [d_e]$, then, $\mathbb{E}[O_{ij}] = r_{hi}y_j$. Now if $j_2 = j_1$, then, $\mathbb{E}[T_{j_1 j_2}] = \mathbb{E}[T_{j_1 j_1}] = 1$. But for $j_1 \neq j_2$, we have,

$$\mathbb{E}[T_{j_1 j_2}] = \begin{cases} n^{-1}\langle r_e, r_e \rangle = n^{-1}\|r_e\|_2^2 & \text{if } j_1 \in [d_e], j_2 \in [d_e], j_1 \neq j_2 \\ n^{-1}\langle r_e, r_h \rangle & \text{if } j_1 \in [d_e], j_2 \notin [d_e], \text{ else if, } j_1 \notin [d_e], j_2 \in [d_e] \\ n^{-1}\langle r_h, r_h \rangle = n^{-1}\|r_h\|_2^2 & \text{if } j_1 \notin [d_e], j_2 \notin [d_e], j_1 \neq j_2 \end{cases}$$

where, the above relations for $j_1 \neq j_2$ is due to the fact that for a worker $i$, the labels provided to different tasks are independent of each other. The above observations yield us with the decomposition of the matrix $\mathbb{E}[T]$.

### E.1 Proof of Lemma 1 and Lemma 2: Spectral Properties of $n^{-1}R_y$

We restate Lemma 1 and Lemma 2 here.

**Restatement of Lemma 1:**

Define the matrix

$$n^{-1}R_y$$
$$:= n^{-1}\text{diag}(y)\begin{pmatrix} \|r_e\|_2^2 1_{d_e \times d_e} & r_e^T r_h 1_{d_e \times d_h} \\ r_h^T r_e 1_{d_h \times d_e} & \|r_h\|_2^2 1_{d_h \times d_h} \end{pmatrix}\text{diag}(y)$$

and a diagonal matrix

$$S - I_d - \frac{1}{n}\text{diag}\left([\|r_e\|_2^2 1_{1 \times d_e}, \|r_h\|_2^2 1_{1 \times d_h}]^T\right).$$

The matrix $n^{-1}R_y$ is rank-$\ell$ with $\ell \leq 2$, and its normalized eigen-gap $\nu(n^{-1}R_y) := d^{-1}(\lambda_1(n^{-1}R_y) - \lambda_2(n^{-1}R_y))$ between its two largest eigenvalues $\lambda_1, \lambda_2$ can be expressed as:

$$\nu(n^{-1}R_y) = \frac{\sqrt{[d_e\|r_e\|_2^2 - d_h\|r_h\|_2^2]^2 + 4d_e d_h(r_e^T r_h)^2}}{nd}.$$

Further, we have the low-rank decomposition

$$\mathbb{E}[T] = n^{-1}R_y + S.$$

**Restatement of Lemma 2:**

Suppose $r_e^{\top} r_h \neq 0$. Then, the principal eigenvector of the matrix $n^{-1}R_y$ has the following form:

$$v(n^{-1}R_y) = \text{diag}(y)\begin{bmatrix} \frac{s}{\sqrt{s^2 d_e + d_h}} 1_{d_e \times 1} \\ \frac{1}{\sqrt{s^2 d_e + d_h}} 1_{d_h \times 1} \end{bmatrix}$$

where

$$s = \omega + \sqrt{\omega^2 + \frac{d_h}{d_e}}$$

and

$$\omega = \frac{d_e\|r_e\|_2^2 - d_h\|r_h\|_2^2}{2d_e r_e^T r_h}.$$

In the alternative case that $r_e^{\top} r_h = 0$, we have that

$$v(n^{-1}R_y) = \text{diag}(y)\begin{bmatrix} \frac{1}{\sqrt{d_e}} 1_{d_e \times 1} \\ 0_{d_h \times 1} \end{bmatrix}.$$

**Proof:**

Recall, $n^{-1}R_y$ is defined as, $n^{-1}R_y = n^{-1}\text{diag}(y) \begin{pmatrix} \|r_e\|_2^2 1_{d_e \times d_e} & r_e^T r_h 1_{d_e \times d_h} \\ r_h^T r_e 1_{d_h \times d_e} & \|r_h\|_2^2 1_{d_h \times d_h} \end{pmatrix} \text{diag}(y)$. Clearly, the $n^{-1}R_y$ is a rank-$\ell$ matrix with $\ell \leq 2$. Specifically,

$$\ell = \begin{cases} 1, & \text{when } r_e \text{ and } r_h \text{ are collinear} \\ 2, & \text{else} \end{cases}$$

Next, we calculate the eigenspectrum of $n^{-1}R_y$. First consider the case when $r_e^\top r_h \neq 0$.

**Case 1: When $r_e^\top r_h \neq 0$:**

Consider a generic vector $q$ of the form $\text{diag}(y)[\bar{s}1_{1 \times d_e}, 1_{1 \times d_h}]^T$ for some $\bar{s}$ as the ratio of the magnitude between the entries of the vector corresponding to different types of tasks. A normalization of $q$ serves as a candidate eigenvector for the matrix $n^{-1}R_y$, where

$$\frac{q}{\|q\|_2} = \text{diag}(y) \begin{bmatrix} \frac{\bar{s}}{\sqrt{d_e \bar{s}^2 + d_h}} 1_{d_e \times 1} \\ \frac{1}{\sqrt{d_e \bar{s}^2 + d_h}} 1_{d_h \times 1} \end{bmatrix}.$$

The eigen-pair equation for the candidate eigenvector above is calculated to be:

$$\frac{1}{n}R_y q = \text{diag}(y) \begin{bmatrix} \left[\frac{1}{n}(\bar{s}d_e \|r_e\|_2^2 + d_h r_e^T r_h)\right] 1_{d_e \times 1} \\ \left[\frac{1}{n}(\bar{s}d_e r_e^T r_h + d_h \|r_h\|_2^2)\right] 1_{d_h \times 1} \end{bmatrix} = \left[\frac{1}{n}(\bar{s}d_e r_e^T r_h + d_h \|r_h\|_2^2)\right] q. \quad (25)$$

Now as $\bar{s}$ is the ratio between the quantity $\frac{1}{n}(\bar{s}d_e \|r_e\|_2^2 + d_h r_e^T r_h)$ and $\frac{1}{n}(\bar{s}d_e r_e^T r_h + d_h \|r_h\|_2^2)$, we can write:

$$\bar{s}\left[\frac{1}{n}(\bar{s}d_e r_e^T r_h + d_h \|r_h\|_2^2)\right] = \frac{1}{n}(\bar{s}d_e \|r_e\|_2^2 + d_h r_e^T r_h). \quad (26)$$

The solutions to this quadratic equation are given by

$$\bar{s} = \frac{d_e \|r_e\|_2^2 - d_h \|r_h\|_2^2 \pm \sqrt{[d_e \|r_e\|_2^2 - d_h \|r_h\|_2^2]^2 + 4 d_e d_h (r_e^T r_h)^2}}{2 d_e r_e^T r_h}. \quad (27)$$

Let us call the solution $\frac{d_e \|r_e\|_2^2 - d_h \|r_h\|_2^2 + \sqrt{[d_e \|r_e\|_2^2 - d_h \|r_h\|_2^2]^2 + 4 d_e d_h (r_e^T r_h)^2}}{2 d_e r_e^T r_h}$ as $s$ and the other solution as $s_2$. The eigenvalues $n^{-1}(\bar{s}d_e r_e^T r_h + d_h \|r_h\|_2^2)$ of $n^{-1}R_y$ corresponding to solutions $s$ and $s_2$ respectively are

$$\lambda_1(n^{-1}R_y) = \frac{d_e \|r_e\|_2^2 + d_h \|r_h\|_2^2 + \sqrt{[d_e \|r_e\|_2^2 - d_h \|r_h\|_2^2]^2 + 4 d_e d_h (r_e^T r_h)^2}}{2n}$$

and

$$\lambda_2(n^{-1}R_y) = \frac{d_e \|r_e\|_2^2 + d_h \|r_h\|_2^2 - \sqrt{[d_e \|r_e\|_2^2 - d_h \|r_h\|_2^2]^2 + 4 d_e d_h (r_e^T r_h)^2}}{2n}, \quad (28)$$

where $\lambda_1(n^{-1}R_y) \geq \lambda_2(n^{-1}R_y)$. By Assumption 1, we have $\|r_e\|_2 > 0$. Hence, for $d_e \geq 1$ and $d_h \geq 1$, we can write, $\lambda_1(n^{-1}R_y) > 0$ and $\lambda_2(n^{-1}R_y) \geq 0$. When $r_e$ and $r_h$ are co-linear, $\lambda_2(n^{-1}R_y) = 0$.

**Case 2: when $r_e^\top r_h = 0$:**

When the reliability vectors are orthogonal, we can write

$$n^{-1}R_y = n^{-1}\|r_e\|_2^2 \text{diag}(y_{1:d_e}) 1_{d_e \times d_e} \text{diag}(y_{1:d_e}) \oplus \|r_h\|_2^2 \text{diag}(y_{d_e+1:d}) 1_{d_h \times d_h} \text{diag}(y_{d_e+1:d}), \quad (29)$$

where $y_{1:d_e}$ and $y_{d_e+1:d}$ are the ground truth vectors corresponding to type easy tasks and type hard tasks respectively and $\oplus$ is the notation for a direct sum. From the expression Equation 29, it is clear that $\text{rank}(n^{-1}R_y) = 2$ when $\|r_h\|_2 \neq 0$ with the following eigenvalues:

$$\lambda_1(n^{-1}R_y) = n^{-1} \max_{k \in \{e,h\}} d_k \|r_k\|_2^2 = n^{-1}d_e \|r_e\|_2^2 \geq \lambda_2(n^{-1}R_y) = n^{-1} \min_{k \in \{e,h\}} d_k \|r_k\|_2^2 = n^{-1}d_h \|r_h\|_2^2,$$

with $\lambda_2(n^{-1}R_y) \geq \lambda_j(n^{-1}R_y) = 0$ for all $j = 3, \ldots, d$.

Also, the eigenvectors of $n^{-1}R_y$ corresponding to the eigenvalues $n^{-1}d_e\|r_e\|_2^2$ and $n^{-1}d_h\|r_h\|_2^2$ are respectively

$$\text{diag}(y)\begin{bmatrix} \frac{1}{\sqrt{d_e}}1_{d_e \times 1} \\ 0_{d_h \times 1} \end{bmatrix} \quad \text{and} \quad \text{diag}(y)\begin{bmatrix} 0_{d_e \times 1} \\ \frac{1}{\sqrt{d_h}}1_{d_h \times 1} \end{bmatrix}.$$

# F    Proof of Theorem 1: Perfect Clustering

This section gives the complete proof of Theorem 1 in the main paper. We restate the theorem here:

**Restatement of Theorem 1:**

Under Assumption 1, if the number of tasks $d$ satisfies

$$d \geq \frac{C_1}{\sqrt{D(r_e, r_h, \alpha, d)}},$$

then Algorithm 1 returns task type estimates such that

$$P(\eta = 0) \geq 1 - 2d^2 \exp\left(-C_2 n D(r_e, r_h, \alpha, d)\right),$$

where the problem-dependent quantity $D(r_e, r_h, \alpha, d)$ characterizing the error exponent and the requirement on $d$ is defined as follows:

$$D(r_e, r_h, \alpha, d) = \begin{cases} \left(\frac{(1-\alpha)^5\rho}{\alpha} \frac{\nu(n^{-1}R_y)\|s\|-1|}{\sqrt{s^2+1}}\right)^2 & \text{when, } r_e^\top r_h \neq 0, \\ \left(\frac{(1-\alpha)^5\rho}{\alpha}\nu(n^{-1}R_y)\right)^2 & \text{when, } r_e^\top r_h = 0 \end{cases}$$

and $C_1$ and $C_2$ are universal constants, independent of the problem parameters.

As discussed in the proof sketch of the theorem, the first step is to show that the principal eigenvector $v(n^{-1}R_y)$ of the signal matrix $n^{-1}R_y$ reveals the type information for each task. This is discussed in detail in Lemma 1 and Lemma 2 and proved in Appendix E. Building upon the Lemma 2, the rest of the proof of Theorem 1 is given in this section as enlisted below.

1. First, we prove Lemma 3 in Subsection F.1.

2. Then we show that the principal eigenvector $\hat{v}$ of the task-similarity matrix $T$ is a small perturbation of $v(n^{-1}R_y)$ in the $l_\infty$ norm sense. This is stated in Lemma 4 and proved in the following Subsection F.2.

3. Next, we relate the event of perfect clustering, that is $\{\eta = 0\}$ with a sufficient condition on the concentration of $\hat{v}$ with respect to $v(n^{-1}R_y)$ (see Proposition 3 in Subsection F.3).

4. Finally, we prove that the condition described in the Proposition 3 is satisfied with high probability. See Subsection F.4 for this final step.

## F.1    Proof of Lemma 3: Concentration of the Noise Matrix $N$

**Restatement of Lemma 3:**

For any $\epsilon > 0$ and any positive values of $n$ and $d$, the task-similarity matrix $T$ concentrates around its expectation as follows:

$$\mathbb{P}\left(\|N\|_\infty \geq \epsilon\right) \leq 2d^2 \exp\left(-\frac{n\epsilon^2}{2d^2}\right).$$

**Proof:**

The proof of the Lemma 3 stating the concentration of $N$ is given as:

$$\mathbb{P}\left(\|N\|_\infty \geq \epsilon\right) = \mathbb{P}\left(\max_{i\in[d]}\sum_{j=1}^d |T_{ij} - \mathbb{E}[T_{ij}]| \geq \epsilon\right) \underbrace{\leq}_{(a)} \sum_{i=1}^d \mathbb{P}\left(\sum_{j=1}^d |T_{ij} - \mathbb{E}[T_{ij}]| \geq \epsilon\right)$$

$$\leq \sum_{i=1}^d \mathbb{P}\left(\max_{j\in[d]} |T_{ij} - \mathbb{E}[T_{ij}]| \geq \frac{\epsilon}{d}\right) \underbrace{\leq}_{(b)} \sum_{i=1}^d \sum_{j=1}^d \mathbb{P}\left(|T_{ij} - \mathbb{E}[T_{ij}]| \geq \frac{\epsilon}{d}\right)$$

$$= \sum_{i=1}^d \sum_{j=1}^d \mathbb{P}\left(\left|\frac{1}{n}\sum_{l=1}^n (X_{li}X_{lj} - \mathbb{E}[X_{li}X_{lj}])\right| \geq \frac{\epsilon}{d}\right)$$

$$\underbrace{\leq}_{(c)} 2d^2 \exp\left(-n\frac{\epsilon^2}{2d^2}\right).$$

In $(a)$ and $(b)$ we use the union bound, and in $(c)$ we employ Hoeffding's inequality for the independent bounded random variables $X_{li}, X_{lj} \in \{\pm 1\}$.

### F.2   $l_\infty$-norm Concentration of the Principal Eigenvector

We prove the Lemma 4 here.

**Restatement of Lemma 4:** If $\nu(n^{-1}R_y)$ satisfies : $\frac{C_3(1-\alpha)^4\rho}{\alpha}\nu(n^{-1}R_y)d - 1 > 0$, then, for every $0 < \epsilon < C_3(1-\alpha)^4\nu(n^{-1}R)d - 1$, the event

$$\min_{\theta\in\{-1,+1\}}\|\theta\hat{v} - v(n^{-1}R_y)\|_\infty$$

$$\geq \frac{C_4\alpha}{(1-\alpha)^4\rho\nu(n^{-1}R_y)d\sqrt{d}}(\epsilon+1)$$

occurs with probability at most $2d^2 \exp\left(-n\frac{\epsilon^2}{2d^2}\right)$ where $C_3$ and $C_4$ are universal positive constants.

**Proof:**

We use a result from the paper Fan et al. (2018) that turns out to be more useful than the standard Devis-Kahan perturbation result (Yu et al., 2014) for $l_\infty$ norm perturbation bounds on the eigenvectors of a perturbed matrix in certain scenarios. In our case, recall that the task-similarity matrix $T$ has the following decomposition :

$$T = n^{-1}R_y + S + N.$$

It turns out that the low-rank structure of $n^{-1}R_y$ and the fact that $S$ is a diagonal matrix with a matrix inf-norm as $d_e n^{-1}\|r_e\|^2$ makes the above decomposition of $T$ a suitable setting for getting a useful $l_\infty$ norm perturbation bound on the principal eigenvector of $T$ treating $n^{-1}R_y$ as the signal matrix. From the Lemma 1, we have $S = -\frac{1}{n}\text{diag}\left([\|r_e\|_2^2 1_{1\times d_e}, \|r_h\|_2^2 1_{1\times d_h}]^T\right) + I_d$. Here we are interested in the distance between $\hat{v}$ which is the principal eigenvector of $T$ and $v(n^{-1}R_y)$ induced by the infinity norm.

Let $n^{-1}R_{y,1}$ be the rank-1 approximation of the signal matrix $n^{-1}R_y$. First, we state the result from Fan et al. (2018) on $l_\infty$ norm perturbation that we use in our paper. Before stating the result from Fan et al. (2018), we need to define a quantity called the coherence of the signal matrix $n^{-1}R_y$ and the coherence of its best rank-1 approximation $n^{-1}R_{y,1}$. Writing the modal matrix of $n^{-1}R_y$ which is of size $d \times \ell$ as $V$ so that its columns correspond to the unit-norm eigenvectors of $n^{-1}R_y$, the coherence $M$ of matrix $n^{-1}R_y$ is defined as

$$M = \frac{d}{\ell}\max_{j\in[d]}\sum_{g=1}^\ell V_{jg}^2. \tag{30}$$

Similarly the coherence $M^1$ of the matrix $n^{-1}R_{y,1}$ is defined as

$$M^1 = d\max_{j\in[d]}(v(n^{-1}R_y)[j])^2. \tag{31}$$

We utilize the following result by Fan et al. (2018), cf. Theorem 3.[2]

**Lemma 5** *Let $\tilde{C}_1$ and $\tilde{C}_2$ be two universal constants. Consider a d-dimensional rank-2 symmetric matrix $A$ and its eigen-decomposition*

$$A = \sum_{g=1}^{2} \lambda_g(A) v_g(A) v_g(A)^T.$$

*Let $m \in \{1, 2\}$. We call $A_m$ as the best rank-m approximation of $A$. Clearly for our construction, $A_2 = A$. Define $\gamma_m$ as $\gamma_m = \|A - A_m\|_\infty$. Clearly, $\gamma_2 = 0$. Denote $M(A_m), \lambda_1(A)$ and $\lambda_2(A)$ as the coherence of the matrix $A_m$, the largest and second largest eigenvalue of $A$. Let a perturbation of $A$ be $\tilde{A}$ and the perturbation $\tilde{A} - A$ is also symmetric with the same dimension as $A$. Then, for each $m \in \{1, 2\}$, if $\lambda_m(A)$ satisfies:*

$$|\lambda_m(A)| - \gamma_m \geq \tilde{C}_1 r^3 (M(A_m))^2 \|\tilde{A} - A\|_\infty \tag{32}$$

*and if*

$$\min_{g \leq m}(\lambda_g(A) - \lambda_{g+1}(A)) > \|\tilde{A} - A\|_2 \tag{33}$$

*with a notation $\lambda_3(A) = 0$, then,*

$$\min_{\theta \in \{-1,+1\}} \|v_1(A) - \theta v_1(\tilde{A})\|_\infty \leq \tilde{C}_2 \left( \frac{m^4 (M(A_m))^2 \|\tilde{A} - A\|_\infty}{(|\lambda_m| - \gamma_m)\sqrt{d}} + \frac{m^{\frac{3}{2}} \sqrt{M(A_m)} \|\tilde{A} - A\|_2}{\min_{g \leq m}(\lambda_g(A) - \lambda_{g+1}(A))\sqrt{d}} \right),$$

*where, $v_1(\tilde{A})$ denotes the principal eigenvector of the matrix $\tilde{A}$.*

### F.2.1   Characterizing the Suitable $m$ Based on the Angle between $r_e$ and $r_h$

Before applying the above Lemma 5, we first characterize which $m$ from the set $\{1, 2\}$ is more suitable in this setting to apply the lemma based on the angle between the vectors $r_e$ and $r_h$. Let us call the angle between $r_e$ and $r_h$ as $\zeta$. The idea is that if $|\sin \zeta|$ is sufficiently small, we use $m = 1$, and if it is sufficiently large, we use $m = 2$.

To understand this, we study the error of the best rank-1 approximation of $n^{-1}R_y$ defined as: $\gamma_1 := \|n^{-1}R_y - n^{-1}R_{y,1}\|_\infty$ and its implication in Equation 32 for the case of $m = 1$ with the signal matrix $A$ and the perturbed matrix $\tilde{A}$ being replaced by $n^{-1}R_y$ and $T$. Let us denote $v_2(n^{-1}R_y)$ as the eigenvector corresponding to the eigenvalue $\lambda_2(n^{-1}R_y)$ of $n^{-1}R_y$. Then, we have

$$\gamma_1 = |\lambda_2(n^{-1}R_y)| \|v_2(n^{-1}R_y) v_2(n^{-1}R_y)^\top\|_\infty = |\lambda_2(n^{-1}R_y)| \max_{j \in [d]} |v_{2j}(n^{-1}R_y)| \sum_{j=1}^{d} |v_{2j}(n^{-1}R_y)|$$

where $v_{2j}(n^{-1}R_y)$ is the $j^{th}$ element of the vector $v_2(n^{-1}R_y)$. Let us first consider the case when $\lambda_2(n^{-1}R_y) \neq 0$. We know from Appendix E, that the magnitude of the elements vector $v_{2j}(n^{-1}R_y)$ are from the set $\left\{ \frac{|s_2|}{\sqrt{d_e s_2^2 + d_h}}, \frac{1}{\sqrt{d_e s_2^2 + d_h}} \right\}$ with $\frac{|s_2|}{\sqrt{d_e s_2^2 + d_h}}$ corresponding to easy tasks and $\frac{1}{\sqrt{d_e s_2^2 + d_h}}$ corresponding to hard tasks where

$$s_2 = \frac{d_e \|r_e\|_2^2 - d_h \|r_h\|_2^2 - \sqrt{[d_e \|r_e\|_2^2 - d_h \|r_h\|_2^2]^2 + 4 d_e d_h (r_e^T r_h)^2}}{2 d_e r_e^T r_h}.$$

---

[2]The theorem in Fan et al. (2018) is for a matrix of rank $\ell$ where $\ell$ can take any finite value, we simplified it for our purpose when $\ell = 2$.

Under our Assumption 1 in the main draft, we have $d_e\|r_e\|_2^2 - d_h\|r_h\|_2^2 \geq 0$. Hence, we have $0 \leq |s_2| \leq 1$. Thus, we can write,

$$\gamma_1 = |\lambda_2(n^{-1}R_y)| \frac{1}{\sqrt{d_e s_2^2 + d_h}} \left( \frac{d_e|s_2|}{\sqrt{d_e s_2^2 + d_h}} + \frac{d_h}{\sqrt{d_e s_2^2 + d_h}} \right)$$

$$= |\lambda_2(n^{-1}R_y)| \frac{d_e|s_2| + d_h}{d_e(s_2)^2 + d_h}$$

$$= |\lambda_2(n^{-1}R_y)| \frac{\frac{d_e}{d_h}|s_2| + 1}{\frac{d_e}{d_h}(s_2)^2 + 1}.$$

Now a bit of calculus shows that the quantity $\frac{\frac{d_e}{d_h}|s_2|+1}{\frac{d_e}{d_h}|s_2|^2+1}$ as a function of $|s_2|$ with $0 \leq |s_2| \leq 1$ achieves a its maxima at a value $\frac{\frac{d_e}{d_h}}{2\left(\sqrt{1+\frac{d_e}{d_h}}-1\right)}$ which can be further upper-bounded by $1.25\frac{d_e}{d_h}$ as $\frac{d_e}{d_h} \geq 1$. Thus, we can write:

$$\gamma_1 \leq 1.25 \frac{d_e}{d_h} |\lambda_2(n^{-1}R_y)|.$$

Next, we would characterize the quantity $|\lambda_1(n^{-1}R_y)| - \gamma_1$ which should be sufficiently large if we put $m = 1$ in the application of Lemma 5 in light of Equation 32. Recall from Appendix E, we can write:

$$\lambda_1(n^{-1}R_y) = \frac{d_e\|r_e\|_2^2 + d_h\|r_h\|_2^2 + \sqrt{\left[d_e\|r_e\|_2^2 - d_h\|r_h\|_2^2\right]^2 + 4d_e d_h(r_e^T r_h)^2}}{2n} \geq 0.$$

$$\lambda_2(n^{-1}R_y) = \frac{d_e\|r_e\|_2^2 + d_h\|r_h\|_2^2 - \sqrt{\left[d_e\|r_e\|_2^2 - d_h\|r_h\|_2^2\right]^2 + 4d_e d_h(r_e^T r_h)^2}}{2n}$$

$$= \frac{d_e\|r_e\|_2^2 + d_h\|r_h\|_2^2 - \sqrt{\left[d_e\|r_e\|_2^2 + d_h\|r_h\|_2^2\right]^2 - 4d_e d_h(\|r_e\|_2^2\|r_h\|_2^2 - (r_e^T r_h)^2)}}{2n} \geq 0.$$

Then, we can write:

$$4d_e d_h(\|r_e\|_2^2\|r_h\|_2^2 - (r_e^T r_h)^2) = 4d_e d_h\|r_e\|_2^2\|r_h\|_2^2(1 - \cos\zeta^2) = 4d_e d_h\|r_e\|_2^2\|r_h\|_2^2(\sin\zeta)^2.$$

Clearly, the quantity $|\lambda_1(n^{-1}R_y)| - \gamma_1$ can be lower-bounded as:

$$|\lambda_1(n^{-1}R_y)| - \gamma_1 \geq \lambda_1(n^{-1}R_y) - 1.25\frac{d_e}{d_h}\lambda_2(n^{-1}R_y)$$

$$= \left(1.25\frac{d_e}{d_h} + 1\right) \frac{\sqrt{\left[d_e\|r_e\|_2^2 + d_h\|r_h\|_2^2\right]^2 - 4d_e d_h\|r_e\|_2^2\|r_h\|_2^2(\sin\zeta)^2}}{2n} - \left(1.25\frac{d_e}{d_h} - 1\right) \frac{(d_e\|r_e\|_2^2 + d_h\|r_h\|_2^2)}{2n}.$$

Now since, $\left[d_e\|r_e\|_2^2 + d_h\|r_h\|_2^2\right]^2 - 4d_e d_h\|r_e\|_2^2\|r_h\|_2^2 = \left[d_e\|r_e\|_2^2 - d_h\|r_h\|_2^2\right]^2 \geq 0$, we have, $\left[d_e\|r_e\|_2^2 + d_h\|r_h\|_2^2\right]^2 \geq 4d_e d_h\|r_e\|_2^2\|r_h\|_2^2$. Hence, $\left[d_e\|r_e\|_2^2 + d_h\|r_h\|_2^2\right]^2 - 4d_e d_h\|r_e\|_2^2\|r_h\|_2^2(\sin\zeta)^2 \geq \left[d_e\|r_e\|_2^2 + d_h\|r_h\|_2^2\right]^2 (1 - (\sin\zeta)^2) = \left[d_e\|r_e\|_2^2 + d_h\|r_h\|_2^2\right]^2 \cos\zeta^2$ giving us the following:

$$|\lambda_1(n^{-1}R_y)| - \gamma_1 \geq \left(\left(1.25\frac{d_e}{d_h} + 1\right)|\cos\zeta| - \left(1.25\frac{d_e}{d_h} - 1\right)\right) \frac{(d_e\|r_e\|_2^2 + d_h\|r_h\|_2^2)}{2n}$$

$$= \left((1 + |\cos\zeta|) - 1.25\frac{d_e}{d_h}(1 - |\cos\zeta|)\right) \frac{(d_e\|r_e\|_2^2 + d_h\|r_h\|_2^2)}{2n}.$$

Clearly, if $|\cos\zeta| \geq 1 - \frac{4d_h}{5d_e}$, we have, $\left((1 + |\cos\zeta|) - 1.25\frac{d_e}{d_h}(1 - |\cos\zeta|)\right) \geq 1 - \frac{4d_h}{5d_e} \geq \frac{1}{5}$. A sufficient condition of $|\cos\zeta| \geq 1 - \frac{4d_h}{5d_e}$ is $(\sin\zeta)^2 \leq \frac{4d_h}{25d_e}$. So we have arrived at the following fact:

**fact:** When the angle between $r_e$ and $r_h$ satisfy $(\sin\zeta)^2 \leq \frac{4d_h}{25d_e}$, we can write:

$$|\lambda_1(n^{-1}R_y)| - \gamma_1 \geq \frac{(d_e\|r_e\|_2^2 + d_h\|r_h\|_2^2)}{10n}.$$

In the alternative case of $((\sin\zeta)^2 > \frac{4d_h}{25d_e})$, the quantity of interest in light of the condition given in Equation 32 in Lemma 5 is the second largest eigenvalue of $n^{-1}R_y$ as the approximation error for a rank-2 approximation in this case is 0. Next, we lower-bound the second largest eigenvalue of $n^{-1}R_y$ as follows:

$$\lambda_2(n^{-1}R_y) = \frac{d_e\|r_e\|_2^2 + d_h\|r_h\|_2^2 - \sqrt{\left[d_e\|r_e\|_2^2 + d_h\|r_h\|_2^2\right]^2 - 4d_ed_h\|r_e\|_2^2\|r_h\|_2^2(\sin\zeta)^2}}{2n}$$

$$\underbrace{\geq}_{(d)} \frac{4d_ed_h\|r_e\|_2^2\|r_h\|_2^2(\sin\zeta)^2}{2n\left(d_e\|r_e\|_2^2 + d_h\|r_h\|_2^2 + \sqrt{\left[d_e\|r_e\|_2^2 + d_h\|r_h\|_2^2\right]^2 - 4d_ed_h\|r_e\|_2^2\|r_h\|_2^2(\sin\zeta)^2}\right)}$$

$$\geq \frac{4d_ed_h\|r_e\|_2^2\|r_h\|_2^2(\sin\zeta)^2}{4n(d_e\|r_e\|_2^2 + d_h\|r_h\|_2^2)}$$

$$\underbrace{\geq}_{(e)} \frac{d_h\|r_h\|_2^2(\sin\zeta)^2}{n}$$

where in $(d)$, we multiply the numerator and the denominator by $\left(d_e\|r_e\|_2^2 + d_h\|r_h\|_2^2 + \sqrt{\left[d_e\|r_e\|_2^2 + d_h\|r_h\|_2^2\right]^2 - 4d_ed_h\|r_e\|_2^2\|r_h\|_2^2(\sin\zeta)^2}\right)$ assuming $|\sin\zeta| \neq 0$. In $(e)$, we use that $d_e \geq d_h$ and $\|r_e\|_2 \geq \|r_h\|_2$.

Hence, for the case of $(\sin\zeta)^2 > \frac{4d_h}{25d_e}$, we have,

$$\lambda_2(n^{-1}R_y) > \frac{2d_h^2\|r_h\|_2^2}{25d_en}.$$

Hence, the idea is to use $m=1$ in the Lemma 5 when $(\sin\zeta)^2 \leq \frac{4d_h}{25d_e}$, otherwise use $m=2$.

### F.2.2 Characterizing the Upper and Lower Bound on the Coherence Terms $M$ and $M^1$

Before applying the Lemma 5 in our case, we want to give an upper bound on the coherence parameter $M$ and $M^1$ defined in Equation 30 and Equation 31 that will be used in the proof of this section. Recall the definition of $M$ and $M^1$ as:

$$M = \frac{d}{\ell}\max_{j\in[d]}\sum_{g=1}^{\ell}V_{jg}^2$$

and

$$M^1 = d\max_{j\in[d]}(v(n^{-1}R_y)[j])^2.$$

From the Lemma 2 of the main draft, the elements of $v(n^{-1}R_y)$ corresponding to easy and hard tasks are as $\frac{s}{\sqrt{d_es^2+d_h}}$ and $\frac{1}{\sqrt{d_es^2+d_h}}$, respectively. Similarly, for non-collinear $r_e$ and $r_h$, the two non-zero eigenvectors for the signal matrix the corresponding entries of the second eigenvector of $n^{-1}R_y$ would be $\frac{s_2}{\sqrt{d_es_2^2+d_h}}$ and $\frac{1}{\sqrt{d_es_2^2+d_h}}$. Here $s$ and $s_2$ takes the following values:

$$s, s_2 = \frac{d_e\|r_e\|_2^2 - d_h\|r_h\|_2^2 \pm \sqrt{\left[d_e\|r_e\|_2^2 - d_h\|r_h\|_2^2\right]^2 + 4d_ed_h(r_e^Tr_h)^2}}{2d_er_e^Tr_h}.$$

From the expressions obtained above, we can write the coherence terms defined in Equation 30 and Equation 31 as

$$M = \frac{d}{\ell}\max_{i\in[d]}\sum_{j=1}^{\ell}V_{ij}^2 = \frac{d}{2}\max\left\{\frac{s^2}{d_es^2+d_h} + \frac{s_2^2}{d_es_2^2+d_h}, \frac{1}{d_es^2+d_h} + \frac{1}{d_es_2^2+d_h}\right\}$$

$$M^1 = d\max\left\{\frac{s^2}{d_es^2+d_h}, \frac{1}{d_es^2+d_h}\right\}.$$

Hence, we can lower bound the coherence terms as:

$$M \geq \frac{1}{2}\left(d_e\frac{s^2}{d_e s^2 + d_h} + d_h\frac{1}{d_e s^2 + d_h} + d_e\frac{s_2^2}{d_e s_2^2 + d_h} + d_h\frac{1}{d_e s_2^2 + d_h}\right) = 1.$$

$$M^1 \geq d_e\frac{s^2}{d_e s^2 + d_h} + d_h\frac{1}{d_e s^2 + d_h} = 1.$$

Moreover, we can upper bound the coherence terms as:

$$M \leq \frac{1}{2}\left(\frac{ds^2 + d}{d_e s^2 + d_h} + \frac{ds_2^2 + d}{d_e s_2^2 + d_h}\right) \underbrace{\leq}_{(f)} \frac{1}{2}\left(\frac{s^2 + 1}{\alpha s^2 + (1-\alpha)} + \frac{s_2^2 + 1}{\alpha s_2^2 + (1-\alpha)}\right) \leq \frac{1}{1-\alpha}$$

$$M^1 \leq \frac{ds^2 + d}{d_e s^2 + d_h} \underbrace{\leq}_{(g)} \frac{s^2 + 1}{\alpha s^2 + (1-\alpha)} \leq \frac{1}{1-\alpha}$$

where in $(f)$ and $(g)$, we use Assumption 1 in the main draft, specifically the assumption $d_e = \alpha d$ and $d_h = (1-\alpha)d$ and $\alpha \geq 0.5$.

### F.2.3   When $(\sin\zeta)^2 > \frac{4d_h}{d_e}$

In this case, we apply Lemma 5 with $m = 2$. We substitute the matrix $A$ with $n^{-1}R_y$ and the perturbation $\tilde{A} - A$ with $S + N$. The conditions to satisfy according to Equations 32 and 33 are:

$$\lambda_2(n^{-1}R_y) \geq \tilde{C}_1 2^3 M^2\|S + N\|_\infty$$

and

$$\min(\lambda_1(n^{-1}R_y) - \lambda_2(n^{-1}R_y), \lambda_2(n^{-1}R_y)) > \|S + N\|_2.$$

Recall from the discussion above, $M \geq 1$ and $M \leq \frac{1}{1-\alpha}$. Also for a symmetric matrix $B$, $\|B\|_\infty \geq \|B\|_2$. Hence, letting $\tilde{C}_3 = \max\{1, \tilde{C}_1 2^3\}$ the sufficient condition to satisfy Equations 32 and 33 can be stated as

$$\min\{\lambda_1(n^{-1}R_y) - \lambda_2(n^{-1}R_y), \lambda_2(n^{-1}R_y)\} \geq \frac{\tilde{C}_3}{(1-\alpha)^2}\|S + N\|_\infty$$

or equivalently

$$\|S + N\|_\infty \leq \frac{(1-\alpha)^2}{\tilde{C}_3}\min\{\lambda_1(n^{-1}R_y) - \lambda_2(n^{-1}R_y), \lambda_2(n^{-1}R_y)\}.$$

Define the event $E_N$ as:

$$E_N := \left\{\|N\|_\infty \leq \frac{(1-\alpha)^2}{\tilde{C}_3}\min\{\lambda_1(n^{-1}R_y) - \lambda_2(n^{-1}R_y), \lambda_2(n^{-1}R_y)\} - 1\right\}.$$

Clearly, on the event $E_N$, the conditions given by Equations 32 and 33 are satisfied by the use of the triangle inequality with the fact that $\|S\|_\infty = 1 - n^{-1}\|r_h\|_2^2 \leq 1$ for the diagonal matrix $S$.

Now conditioning on the event $E_N$ we can use Lemma 5 as:

$$\min_{\theta\in\{-1,+1\}} \|v(n^{-1}R_y) - \theta\hat{v}\|_\infty \leq \tilde{C}_2\left(\frac{2^4 M^2\|S + N\|_\infty}{(\lambda_2(n^{-1}R_y))\sqrt{d}} + \frac{2^{\frac{3}{2}}\sqrt{M}\|S + N\|_2}{\min\{\lambda_1(n^{-1}R_y) - \lambda_2(n^{-1}R_y), \lambda_2(n^{-1}R_y)\}\sqrt{d}}\right)$$

$$\underbrace{\leq}_{(h)} \frac{\tilde{C}_4\|S + N\|_\infty}{(1-\alpha)^2\min\{\lambda_1(n^{-1}R_y) - \lambda_2(n^{-1}R_y), \lambda_2(n^{-1}R_y)\}\sqrt{d}}$$

$$\underbrace{\leq}_{(i)} \frac{\tilde{C}_4\left[\|N\|_\infty + 1\right]}{(1-\alpha)^2\min\{\lambda_1(n^{-1}R_y) - \lambda_2(n^{-1}R_y), \lambda_2(n^{-1}R_y)\}\sqrt{d}}. \tag{34}$$

In (h), we let $\tilde{C}_4 = \tilde{C}_2(2^4\tilde{C}_3 + 2^{\frac{3}{2}})$ and we use the fact that $1 \le M \le \frac{1}{1-\alpha}$, in (i), we use $\|S\|_\infty \le 1$.

We are interested in the event $E_N \cap \{\|N\|_\infty \le \epsilon\}$ for some $\epsilon$ such that, $0 < \epsilon \le \frac{(1-\alpha)^2}{\tilde{C}_3} \min\{\lambda_1(n^{-1}R_y) - \lambda_2(n^{-1}R_y), \lambda_2(n^{-1}R_y)\} - 1$. On the event $E_N \cap \{\|N\|_\infty \le \epsilon\}$, the following is satisfied using Equation 34:

$$\min_{\theta \in \{-1,+1\}} \|v(n^{-1}R_y) - \theta\hat{v}\|_\infty \le \tilde{C}_4 \frac{\epsilon + 1}{(1-\alpha)^2 \min\{\lambda_1(n^{-1}R_y) - \lambda_2(n^{-1}R_y), \lambda_2(n^{-1}R_y)\}\sqrt{d}}. \tag{35}$$

It remains to show that the event $E_N \cap \{\|N\|_\infty \le \epsilon\}$ for some $\epsilon$ in the range $0 < \epsilon \le \frac{(1-\alpha)^2}{\tilde{C}_3} \min\{\lambda_1(n^{-1}R_y) - \lambda_2(n^{-1}R_y), \lambda_2(n^{-1}R_y)\} - 1$ occurs with high probability:

$$\mathbb{P}\left(E_N \cap \{\|N\|_\infty \le \epsilon\}\right) \underbrace{=}_{(j)} 1 - \mathbb{P}(\{\|N\|_\infty \le \epsilon\}^c) \underbrace{\ge}_{(k)} 1 - 2d^2 \exp\left(\frac{-n\epsilon^2}{2d^2}\right)$$

where in $(j)$ we use the fact that the event $\{\|N\|_\infty \le \epsilon\}$ is a subset of the event $E_N$ and in $(k)$, we use Lemma 3 in the main draft.

### F.2.4 When $(\sin\zeta)^2 \le \frac{4d_h}{d_e}$

In this case, we apply Lemma 5 with $m = 1$. The steps are similar to the other case with a few differences. The conditions to satisfy according to Equations 32 and 33 are:

$$\lambda_1(n^{-1}R_y) - \gamma_1 \ge \tilde{C}_1(M^1)^2\|S + N\|_\infty$$

and

$$\lambda_1(n^{-1}R_y) - \lambda_2(n^{-1}R_y) > \|S + N\|_2.$$

Recall from the bounds on the coherence terms, $M^1 \ge 1$ and $M^1 \le \frac{1}{1-\alpha}$. Hence, letting $\tilde{C}_3 = \max\{1, \tilde{C}_1\}$ the sufficient condition to satisfy Equations 32 and 33 can be stated as

$$\min\{\lambda_1(n^{-1}R_y) - \lambda_2(n^{-1}R_y), \lambda_1(n^{-1}R_y) - \gamma_1\} \ge \frac{\tilde{C}_3}{(1-\alpha)^2}\|S + N\|_\infty$$

or equivalently

$$\|S + N\|_\infty \le \frac{(1-\alpha)^2}{\tilde{C}_3} \min\{\lambda_1(n^{-1}R_y) - \lambda_2(n^{-1}R_y), \lambda_1(n^{-1}R_y) - \gamma_1\}.$$

Define the event $E_N^2$ as:

$$E_N^2 := \left\{\|N\|_\infty \le \frac{(1-\alpha)^2}{\tilde{C}_3} \min\{\lambda_1(n^{-1}R_y) - \lambda_2(n^{-1}R_y), \lambda_1(n^{-1}R_y) - \gamma_1\} - 1\right\}.$$

Clearly, on the event $E_N^2$ the conditions given by Equations 32 and 33 are satisfied by the use of the triangle inequality with the fact that $\|S\|_\infty = 1 - n^{-1}\|r_h\|_2^2 \le 1$ for the diagonal matrix $S$.

Now conditioning on the event $E_N^2$ we can use the Lemma 5 as:

$$\min_{\theta \in \{-1,+1\}} \|v(n^{-1}R_y) - \theta\hat{v}\|_\infty \le \tilde{C}_2 \left(\frac{(M^1)^2\|S + N\|_\infty}{(\lambda_1(n^{-1}R_y) - \gamma_1)\sqrt{d}} + \frac{\sqrt{M^1}\|S + N\|_2}{(\lambda_1(n^{-1}R_y) - \lambda_2(n^{-1}R_y))\sqrt{d}}\right)$$

$$\underbrace{\le}_{(l)} \frac{\tilde{C}_4\|S + N\|_\infty}{(1-\alpha)^2 \min\{\lambda_1(n^{-1}R_y) - \lambda_2(n^{-1}R_y), \lambda_1(n^{-1}R_y) - \gamma_1\}\sqrt{d}}$$

$$\underbrace{\le}_{(m)} \frac{\tilde{C}_4\left[\|N\|_\infty + 1\right]}{(1-\alpha)^2 \min\{\lambda_1(n^{-1}R_y) - \lambda_2(n^{-1}R_y), \lambda_1(n^{-1}R_y) - \gamma_1\}\sqrt{d}}. \tag{36}$$

In (l) use the fact that $1 \leq M^1 \leq \frac{1}{1-\alpha}$, in (m), we use $\|S\|_\infty \leq 1$ .

We are interested in the event $E_N^2 \cap \{\|N\|_\infty \leq \epsilon\}$ for some $\epsilon$ such that, $0 < \epsilon \leq \frac{(1-\alpha)^2}{\tilde{C}_3} \min\{\lambda_1(n^{-1}R_y) - \lambda_2(n^{-1}R_y), \lambda_1(n^{-1}R_y) - \gamma_1\} - 1$. On the event $E_N^2 \cap \{\|N\|_\infty \leq \epsilon\}$, the following is satisfied using Equation 36:

$$\min_{\theta \in \{-1,+1\}} \|v(n^{-1}R_y) - \theta\hat{v}\|_\infty \leq \tilde{C}_4 \frac{\epsilon + 1}{(1-\alpha)^2 \min\{\lambda_1(n^{-1}R_y) - \lambda_2(n^{-1}R_y), \lambda_1(n^{-1}R_y) - \gamma_1\}\sqrt{d}}. \tag{37}$$

It remains to show that the event $E_N^2 \cap \{\|N\|_\infty \leq \epsilon\}$ for some $\epsilon$ in the range $0 < \epsilon \leq \frac{(1-\alpha)^2}{\tilde{C}_3} \min\{\lambda_1(n^{-1}R_y) - \lambda_2(n^{-1}R_y), \lambda_1(n^{-1}R_y) - \gamma_1\} - 1$ occurs with high probability:

$$\mathbb{P}\left(E_N \cap \{\|N\|_\infty \leq \epsilon\}\right) \underbrace{=}_{(n)} 1 - \mathbb{P}(\{\|N\|_\infty \leq \epsilon\}^c) \underbrace{\geq}_{(o)} 1 - 2d^2 \exp\left(\frac{-n\epsilon^2}{2d^2}\right)$$

where in $(n)$ we use the fact that the event $\{\|N\|_\infty \leq \epsilon\}$ is a subset of the event $E_N^2$ and in $(o)$, we use the lemma 3 in the main draft.

### F.2.5 Combining the Two Regimes: Completing the Proof of Lemma 4

Recall the following fact proved before in this section:
**fact:** When the angle between $r_e$ and $r_h$ satisfy $(\sin\zeta)^2 \leq \frac{4d_h}{25d_e}$, we can write:

$$|\lambda_1(n^{-1}R_y)| - \gamma_1 \geq \frac{(d_e\|r_e\|_2^2 + d_h\|r_h\|_2^2)}{10n}$$

and for the case of $(\sin\zeta)^2 > \frac{4d_h}{25d_e}$, we have,

$$\lambda_2(n^{-1}R_y) > \frac{2d_h^2\|r_h\|_2^2}{25d_e n}.$$

We use the above fact to combine the two regimes to complete the proof of Lemma 4 of the main draft. When $(\sin\zeta)^2 > \frac{4d_h}{25d_e}$, we can write:

$$\min\{\lambda_1(n^{-1}R_y) - \lambda_2(n^{-1}R_y), \lambda_2(n^{-1}R_y)\} \geq \min\{\lambda_1(n^{-1}R_y) - \lambda_2(n^{-1}R_y), \frac{2d_h^2\|r_h\|_2^2}{25d_e n}\}$$

$$= d\min\{d^{-1}(\lambda_1(n^{-1}R_y) - \lambda_2(n^{-1}R_y)), \frac{2(1-\alpha)^2\|r_h\|_2^2}{25\alpha n}\}.$$

Now recall from Lemma 1 of the main draft,

$$\lambda_1(n^{-1}R_y) - \lambda_2(n^{-1}R_y) = \frac{\sqrt{[d_e\|r_e\|_2^2 - d_h\|r_h\|_2^2]^2 + 4d_e d_h(r_e^T r_h)^2}}{n}$$

$$\leq \frac{\left[d_e\|r_e\|_2^2 + d_h\|r_h\|_2^2\right]^2}{n} \leq d.$$

Also, from the Assumption 1 of the main draft, we have, $\|r_h\|^2 \geq 2\rho n$. From the above two observations, we can write, when $(\sin\zeta)^2 > \frac{4d_h}{25d_e}$,

$$\min\{\lambda_1(n^{-1}R_y) - \lambda_2(n^{-1}R_y), \lambda_2(n^{-1}R_y)\} \geq \frac{\tilde{C}_5(1-\alpha)^2\rho}{\alpha}(\lambda_1(n^{-1}R_y) - \lambda_2(n^{-1}R_y))$$

where, we let $\tilde{C}_5 = \frac{4}{25}$.
Now for the alternative case of $(\sin\zeta)^2 \leq \frac{4d_h}{25d_e}$, we can write,

$$\min\{\lambda_1(n^{-1}R_y) - \lambda_2(n^{-1}R_y), \lambda_1(n^{-1}R_y) - \gamma_1\} \geq \min\{\lambda_1(n^{-1}R_y) - \lambda_2(n^{-1}R_y), \frac{d_e\|r_e\|_2^2 + d_h\|r_h\|_2^2}{10n}\}.$$

Now as shown above, we have, $\lambda_1(n^{-1}R_y) - \lambda_2(n^{-1}R_y) \leq \frac{d_e\|r_e\|_2^2 + d_h\|r_h\|_2^2}{n}$, giving us: for the case of $(\sin\zeta)^2 \leq \frac{4d_h}{25d_e}$

$$\min\{\lambda_1(n^{-1}R_y) - \lambda_2(n^{-1}R_y), \lambda_1(n^{-1}R_y) - \gamma_1\} \geq \frac{1}{10}(\lambda_1(n^{-1}R_y) - \lambda_2(n^{-1}R_y)).$$

Hence, we can combine the two cases in the following statement. Let $\tilde{C}_6 = \min\{\tilde{C}_5, \frac{1}{10}\}$. If $\nu(n^{-1}R_y)$ satisfies:

$$\frac{\tilde{C}_6(1-\alpha)^4\rho}{\tilde{C}_3\alpha}\nu(n^{-1}R_y)d - 1 > 0$$

then, for every $\epsilon$ such that $0 < \epsilon < \frac{\tilde{C}_6(1-\alpha)^4\rho}{\tilde{C}_3\alpha}\nu(n^{-1}R_y)d - 1$, we have the following:

$$\mathbb{P}\left(\min_{\theta\in\{-1,+1\}}\|v(n^{-1}R_y) - \theta\hat{v}\|_\infty \geq \frac{\tilde{C}_4\alpha}{\tilde{C}_6(1-\alpha)^4\rho}\frac{\epsilon+1}{\nu(n^{-1}R_y)d\sqrt{d}}\right) \leq 2d^2\exp\left(\frac{-n\epsilon^2}{2d^2}\right),$$

where we used the following notation from the main draft: $\nu(n^{-1}R_y) = d^{-1}(\lambda_1(n^{-1}R_y) - \lambda_2(n^{-1}R_y))$. Letting $C_3 = \frac{\tilde{C}_6}{\tilde{C}_3}$ and $C_4 = \frac{\tilde{C}_4}{\tilde{C}_6}$, we arrive at the statement in Lemma 4.

### F.3 Sufficient Condition for Perfect Clustering

Here, we relate the event of perfect clustering with the concentration of the principal eigenvector $\hat{v}$ with respect to $v(n^{-1}R_y)$.

**Proposition 3** *Under the stated assumptions, Algorithm 1 achieves perfect clustering, that is $\eta = 0$ when the following event occurs :*

$$E_{l_\infty} := \left\{\min_{\theta\in\{-1,+1\}}\|v(n^{-1}R_y) - \theta\hat{v}\|_\infty < \frac{1}{2}\min\{m_e(n^{-1}R_y), m_h(n^{-1}R_y)\}\right\}.$$

The proof of the above proposition is given in Appendix F.6.3

### F.4 Proof of the Theorem 1: Perfect Clustering

Now we complete the proof of the clustering Theorem 1. From Proposition 3, we know that,

$$\mathbb{P}(\eta = 0) \geq \mathbb{P}\left(\min_{\theta\in\{-1,+1\}}\|v(n^{-1}R_y) - \theta\hat{v}\|_\infty < \frac{1}{2}\min\{m_e(n^{-1}R_y), m_h(n^{-1}R_y)\}\right).$$

Now we show that the right hand side of the above equation is close to 1 for large values of $n$ using lemma 4. We also derive the corresponding necessary conditions on the problem parameters $n$ and $d$.

One requirement of Lemma 4 is that $\frac{C_3(1-\alpha)^4\rho}{\alpha}\nu(n^{-1}R_y)d - 1 > 0$. This leads to the following requirement on $d$:

$$d > \frac{\alpha}{C_3(1-\alpha)^4\rho\nu(n^{-1}R_y)}. \tag{38}$$

Under Equation 38, we have from Lemma 4, for every $0 < \epsilon < \frac{C_3(1-\alpha)^4\rho}{\alpha}\nu(n^{-1}R_y)d - 1$,

$$\mathbb{P}\left(\min_{\theta\in\{-1,+1\}}\|\theta\hat{v} - v(n^{-1}R_y)\|_\infty \geq C_4\frac{\alpha(\epsilon+1)}{(1-\alpha)^4\rho\nu(n^{-1}R_y)d\sqrt{d}}\right) \leq 2d^2\exp\left(-n\frac{\epsilon^2}{2d^2}\right),$$

Next, we choose a suitable $\epsilon$ with $0 < \epsilon < \frac{C_3(1-\alpha)^4\rho}{\alpha}\nu(n^{-1}R_y)d - 1$ such that

$$C_4\frac{\alpha(\epsilon+1)}{(1-\alpha)^4\rho\nu(n^{-1}R_y)d\sqrt{d}} \leq \frac{1}{2}\min\{m_e(n^{-1}R_y), m_h(n^{-1}R_y)\}.$$

The following choice of $\epsilon$ satisfies the above requirement :

$$\epsilon = \frac{1}{4\max\{C_3, C_4, 1\}} \frac{(1-\alpha)^4 \rho}{\alpha} \nu(n^{-1}R_y) d \min(m_e(n^{-1}R_y)d^{\frac{1}{2}}, m_h(n^{-1}R_y)d^{\frac{1}{2}}, 1)$$

, when we impose :

$$d > \frac{4\max\{C_3, C_4, 1\}\alpha}{(1-\alpha)^4 \rho \nu(n^{-1}R_y) \min\left\{m_e(n^{-1}R_y)d^{1/2}, m_h(n^{-1}R_y)d^{1/2}, 1\right\}}. \tag{39}$$

Notice that the requirement on $d$ in Equation 39 is stronger than the requirement in Equation 38. Putting it together, we get, when $d$ satisfies Equation 39 the perfect clustering is guaranteed as

$$\mathbb{P}\left(\eta = 0\right) \geq \mathbb{P}\left(\min_{\theta \in \{-1, +1\}} \|v(n^{-1}R_y) - \theta\hat{v}\|_\infty < \frac{1}{2}\min\left\{m_e(n^{-1}R_y), m_h(n^{-1}R_y)\right\}\right)$$

$$\geq 1 - 2d^2 \exp\left(-\frac{n}{2}\left(\frac{(1-\alpha)^4 \rho \nu(n^{-1}R_y) \min\left\{m_e(n^{-1}R_y)d^{1/2}, m_h(n^{-1}R_y)d^{1/2}, 1\right\}}{4\max\{C_3, C_4, 1\}\alpha}\right)^2\right).$$

When $r_e^\top r_h = 0$, from the analysis of Appendix E.1 we have,

$$m_e(n^{-1}R_y) = \mu_e(n^{-1}R_y) - \mu(n^{-1}R_y) = \frac{d_h}{d}(\mu_e(n^{-1}R_y) - \mu_h(n^{-1}R_y)) = \frac{d_h}{d}\frac{1}{\sqrt{d_e}}.$$

$$m_h(n^{-1}R_y) = \mu(n^{-1}R_y) - \mu_h(n^{-1}R_y) = \frac{d_e}{d}(\mu_e(n^{-1}R_y) - \mu_h(n^{-1}R_y)) = \frac{d_e}{d}\frac{1}{\sqrt{d_e}}.$$

Now, $d_h = (1 - \alpha)d$ and $d_e = \alpha d$ with $0 < \alpha < 1$. Hence for this case, we have $\min\left\{m_e(n^{-1}R_y)d^{1/2}, m_h(n^{-1}R_y)d^{1/2}, 1\right\} \geq 1 - \alpha$. On the other hand when, $r_e^\top r_h \neq 0$, it is convenient to express the absolute margins $m_e(n^{-1}R_y)$ and $m_h(n^{-1}R_y)$ as a function of the ratio $s = \mu_e(n^{-1}R_y)/\mu_h(n^{-1}R_y)$ between the easy and hard magnitudes $\mu_e(n^{-1}R_y), \mu_h(n^{-1}R_y)$ so that

$$m_e(n^{-1}R_y) = \mu_e(n^{-1}R_y) - \mu(n^{-1}R_y) = \frac{d_h}{d}(\mu_e(n^{-1}R_y) - \mu_h(n^{-1}R_y)) = \frac{d_h}{d}\frac{||s| - 1|}{\sqrt{d_e s^2 + d_h}}. \tag{40}$$

$$m_h(n^{-1}R_y) = \mu(n^{-1}R_y) - \mu_h(n^{-1}R_y) = \frac{d_e}{d}(\mu_e(n^{-1}R_y) - \mu_h(n^{-1}R_y)) = \frac{d_e}{d}\frac{||s| - 1|}{\sqrt{d_e s^2 + d_h}}. \tag{41}$$

Hence, we can lower bound the term $\min\{m_e(n^{-1}R_y)d^{1/2}, m_h(n^{-1}R_y)d^{1/2}, 1\}$ as follows:

$$\min\{m_e(n^{-1}R_y)d^{1/2}, m_h(n^{-1}R_y)d^{1/2}, 1\} = \min\left\{\frac{d_h}{d}\frac{||s| - 1|}{\sqrt{d_e s^2 + d_h}}d^{1/2}, \frac{d_e}{d}\frac{||s| - 1|}{\sqrt{d_e s^2 + d_h}}d^{1/2}, 1\right\}$$

$$= \min\left\{\frac{\alpha||s| - 1|}{\sqrt{\alpha s^2 + (1-\alpha)}}, \frac{(1-\alpha)||s| - 1|}{\sqrt{\alpha s^2 + (1-\alpha)}}, 1\right\} \underset{(p)}{\geq} \min\left\{\alpha, 1 - \alpha\right\}\frac{||s| - 1|}{\sqrt{s^2 + 1}}$$

where in $(p)$, we use the fact that $\min\left\{\alpha, 1 - \alpha\right\}\frac{||s|-1|}{\sqrt{s^2+1}} \leq 1$. From the above bounds on $M$ and $\min\{m_e(n^{-1}R_y)d^{1/2}, m_h(n^{-1}R_y)d^{1/2}, 1\}$, we can write the sufficient number of tasks required for perfect clustering as:

$$d \geq \frac{C_1}{\sqrt{D(r_e, r_h, \alpha, d)}}$$

and the probability guarantee of perfect clustering as

$$\mathbb{P}(\eta = 0) \geq 1 - 2d^2 \exp\left(-C_2 n D(r_e, r_h, \alpha, d)\right),$$

where the problem-dependent quantity $D(r_e, r_h, \alpha, d)$ characterizing the error exponent and the requirement on $d$ is given by

$$D(r_e, r_h, \alpha, d) = \begin{cases} \left(\frac{(1-\alpha)^5 \rho}{\alpha}\frac{\nu(n^{-1}R_y)||s|-1|}{\sqrt{s^2+1}}\right)^2 & \text{when, } r_e^\top r_h \neq 0, \\ \left(\frac{(1-\alpha)^5 \rho}{\alpha}\nu(n^{-1}R_y)\right)^2 & \text{when, } r_e^\top r_h = 0, \end{cases}$$

with the positive universal constants $C_1 = 4\max\{C_3, C_4, 1\}$ and $C_2 = \frac{1}{2^5(\max\{C_3, C_4, 1\})^2}$.

### F.5 Proof of Corollary 1

**Restatement of Corollary 1:**

If there exist some universal constant $\beta$ with $0 < \beta \leq 1$,

$$\frac{\|r_e\|_2^2 - \|r_h\|_2^2}{n} \geq \beta$$

then, under Assumption 1, Algorithm 1 achieves clustering error $\eta = 0$ with probability at least $1 - \delta$ when

$$n \geq \frac{C_6 \alpha^2 \log\left(\frac{2d}{\delta}\right)}{(1-\alpha)^{12} \rho^2 \beta^4}$$

where $C_6 = \frac{160}{C_2}$

**Proof:**

We are assuming that there exist a positive $\beta$ such that $\beta \leq 1$, the following is true :

$$\|r_e\|^2 - \|r_h\|^2 \geq \beta n.$$

We would like to lower-bound the following term:

$$D(r_e, r_h, \alpha, d) = \begin{cases} \left(\frac{(1-\alpha)^5 \rho}{\alpha} \frac{\nu(n^{-1} R_y)||s|-1|}{\sqrt{s^2+1}}\right)^2 & \text{when, } r_e^\top r_h \neq 0, \\ \left(\frac{(1-\alpha)^5 \rho}{\alpha} \nu(n^{-1} R_y)\right)^2 & \text{when, } r_e^\top r_h = 0. \end{cases}$$

Let us first lower-bound $\nu(n^{-1} R_y)$. We have the following:

$$\begin{aligned} \nu(n^{-1} R_y) &= \frac{\lambda_1(n^{-1} R_y) - \lambda_2(n^{-1} R_y)}{nd} \\ &= \frac{\sqrt{(d_e\|r_e\|_2^2 - d_h\|r_h\|_2^2)^2 + 4 d_e d_h (r_e^\top r_h)^2}}{nd} \\ &\geq \frac{d_e\|r_e\|_2^2 - d_h\|r_h\|_2^2}{nd} \\ &\geq \frac{d_e\|r_e\|_2^2 - d_e\|r_h\|_2^2 + d_e\|r_h\|_2^2 - d_h\|r_h\|_2^2}{nd} \\ &\underbrace{\geq}_{(p)} \frac{d_e(\|r_e\|_2^2 - \|r_h\|_2^2)}{nd} \\ &\geq \alpha\beta \end{aligned}$$

where in $(p)$, we used the assumptions $d_e \geq d_h$ and $\|r_e\|_2 \geq \|r_h\|_2$. Next, let us lower-bound the term $\frac{(|s|-1)^2}{s^2+1}$. Recall from the Lemma 2 of the main draft, we have:

$$s = \omega + \sqrt{\omega^2 + \frac{d_h}{d_e}}$$

with

$$\omega = \frac{d_e\|r_e\|_2^2 - d_h\|r_h\|^2}{2 d_e r_e^\top r_h}.$$

Using simple calculus, we can show that, $\frac{(|s|-1)^2}{s^2+1} \geq \min\left\{\frac{1}{25}, \frac{1}{20}\left(|\omega| - \frac{d_e - d_h}{2d_e}\right)^2\right\}$. The rest is to lower-bound $\left(|\omega| - \frac{d_e - d_h}{2d_e}\right)$. We can write:

$$
\begin{aligned}
|\omega| - \frac{d_e - d_h}{2d_e} &= \frac{d_e\|r_e\|^2 - d_h\|r_h\|_2^2 - (d_e - d_h)|r_e^\top r_h|}{2d_e|r_e^\top r_h|} \\
&= \frac{d_e\|r_e\|^2 - d_h\|r_e\|_2^2 + d_h(\|r_e\|^2 - \|r_h\|_2^2) - (d_e - d_h)|r_e^\top r_h|}{2d_e|r_e^\top r_h|} \\
&\underbrace{\geq}_{(q)} \frac{d_h(\|r_e\|^2 - \|r_h\|_2^2)}{2d_e|r_e^\top r_h|} \\
&\geq \frac{d_h\beta}{2d_e} = \frac{\beta(1-\alpha)}{2\alpha}
\end{aligned}
$$

where in $(q)$ we use $\|r_e\|_2 \geq \|r_h\|_2$ and in $(c)$, we used that $|r_e^\top r_h| \leq n$. Thus, we can lower-bound $\frac{(|s|-1)^2}{s^2+1}$ as:

$$
\frac{(|s|-1)^2}{s^2+1} \geq \min\left\{\frac{1}{25}, \frac{\beta^2(1-\alpha)^2}{2^4 \cdot 5\alpha^2}\right\}
$$

Giving us:

$$
D(r_e, r_h, \alpha, d) \geq \frac{(1-\alpha)^{12}\rho^2\beta^4}{2^4 \cdot 5\alpha^2}.
$$

From Theorem 1, the requirement on $n$ for an event of perfect clustering with probability $\geq 1 - \delta$ becomes:

$$
n \geq \frac{\log\left(\frac{2d^2}{\delta}\right)}{C_2 D(r_e, r_h, \alpha, d)}.
$$

Hence, we can write that, Algorithm 1 achieves clustering error $\eta = 0$ with probability at least $1 - \delta$ when

$$
n \geq \frac{C_6\alpha^2 \log\left(\frac{2d}{\delta}\right)}{(1-\alpha)^{12}\rho^2\beta^4}
$$

where $C_6 = \frac{160}{C_2}$

## F.6   Remaining Part of Proofs for Theorem 1

Here we characterize the relation between the $l_\infty$ norm concentration of the eigenvectors with the event of perfect clustering which leads to a proof of the Proposition 3.

### F.6.1   Relating the Event of Misclustering to Eigenvector Concentration

Before stating the sufficient condition for the perfect clustering, we state a more general result that provides the sufficient conditions for a clustering error $\eta \leq 1 - t$ for some $t \in [0, 1]$ in the following proposition.

**Proposition 4** *Let $\theta$ be the sign that resolves the eigenvector ambiguity*

$$
\theta = \arg\min_{\theta \in \{-1, +1\}} \|v(n^{-1}R_y) - \theta\hat{v}\|_2.
$$

*Fix any non-negative $t \leq 1$, Algorithm 1 returns cluster membership with $\eta \leq 1 - t$ on the following event on the random vector $\hat{v}$ and the random variable $\hat{\mu}$:*

$$
\frac{1}{d}\sum_{j=1}^{d} \mathbf{1}(E_{\hat{v},j}) \geq t \tag{42}
$$

*where,*

$$
E_{\hat{v},j} = \left\{|v_j(n^{-1}R_y) - \theta\hat{v}_j| + |\mu(n^{-1}R_y) - \hat{\mu}| < \min\{m_e(n^{-1}R_y), m_h(n^{-1}R_y)\}\right\}.
$$

**Proof 1** *Assume the event defined in Equation 42 is true for a fixed $t$ such that $0 \leq t \leq 1$. Under this event we show that there exists a permutation $\pi$ from $\{e, h\}$ to $\{e, h\}$ such that $\eta \leq 1 - t$.*

*First, consider the case of $\mu_e(n^{-1}R_y) \geq \mu_h(n^{-1}R_y)$. Our candidate permutation for this case is $\pi = \{e \mapsto e; h \mapsto h\}$. We claim that when event $E_{\hat{v}, j}$ is true, the task $j$ is clustered into group 1 if $k_j = e$ and into group 2 otherwise. Under this claim, it is easy to see that on the event Equation 42, at least $t$ fraction of tasks are correctly clustered, that is, $\eta \leq 1 - t$. We are left to prove the claim now. Consider the case $k_j = e$ for a task $j$. By definition of the absolute margins $m_e(n^{-1}R_y)$ and $m_h(n^{-1}R_y)$, we have that $\min\{m_e(n^{-1}R_y), m_h(n^{-1}R_y)\} \leq |v_j(n^{-1}R_y)| - \mu$. Suppose $E_{\hat{v}, j}$ is true. Then,*

$$
\begin{aligned}
|\hat{v}_j| - \hat{\mu} &= |v_j(n^{-1}R_y)| - \mu(n^{-1}R_y) + \mu(n^{-1}R_y) - \hat{\mu} + |\theta\hat{v}_j| - |v_j(n^{-1}R_y)| \\
&\geq \min\{m_e(n^{-1}R_y), m_h(n^{-1}R_y)\} - |v_j(n^{-1}R_y) - \theta\hat{v}_j| + |\mu(n^{-1}R_y) - \hat{\mu}| \\
&\underbrace{>}_{(r)} \min\{m_e(n^{-1}R_y), m_h(n^{-1}R_y)\} - \min\{m_e(n^{-1}R_y), m_h(n^{-1}R_y)\} = 0,
\end{aligned}
$$

*where $(r)$ is due to event $E_{\hat{v}, j}$. This implies $|\hat{v}_j| > \hat{\mu}$. This proves that task $j$ is correctly clustered as $\hat{k}_j = e$ and $\pi(\hat{k}_j) = e$. By the similar arguments for $k_j = h$, we obtain that $\pi(\hat{k}_j) = h$ in the same event.*

*Lastly, consider the case of $\mu_e(n^{-1}R_y) < \mu_h(n^{-1}R_y)$. The flow is almost identical for the case of $\mu_e(n^{-1}R_y) \geq \mu_h(n^{-1}R_y)$ but, it is given below for completeness. Our candidate permutation for this case is $\pi = \{e \mapsto h; h \mapsto e\}$. We claim that when event $E_{\hat{v}, j}$ is true, the task $j$ is clustered into group 1 if $k_j = h$ and into group 2 otherwise. Under this claim, it is easy to see that under event Equation 42, at least $t$ fraction of tasks are correctly clustered, that is, $\eta \leq 1 - t$. We are left to prove the claim now. Consider the case $k_j = e$ for a task $j$. By definition of the absolute margins $m_e(n^{-1}R_y)$ and $m_h(n^{-1}R_y)$, we have that $\min\{m_e(n^{-1}R_y), m_h(n^{-1}R_y)\} \leq \mu(n^{-1}R_y) - |v_j(n^{-1}R_y)|$. Suppose $E_{v, j}$ is true. Then,*

$$
\begin{aligned}
\hat{\mu} - |\hat{v}_j| &= \mu(n^{-1}R_y) - |v_j(n^{-1}R_y)| + \hat{\mu} - \mu(n^{-1}R_y) + |\theta\hat{v}_j| - |v_j(n^{-1}R_y)| \\
&\geq \min\{m_e(n^{-1}R_y), m_h(n^{-1}R_y)\} - |v_j(n^{-1}R_y) - \theta\hat{v}_j| + |\mu(n^{-1}R_y) - \hat{\mu}| \\
&\underbrace{>}_{(s)} \min\{m_e(n^{-1}R_y), m_h(n^{-1}R_y)\} - \min\{m_e(n^{-1}R_y), m_h(n^{-1}R_y)\} = 0,
\end{aligned}
$$

*where $(s)$ is due to the event $E_{\hat{v}, j}$. This implies $|\hat{v}_j| < \hat{\mu}$. This proves that task $j$ is correctly clustered as $\pi(\hat{k}_j) = e$. Repeating the same argument for $k_j = h$, we obtain that $\pi(\hat{k}_j) = h$ in the same event.*

### F.6.2 Concentration of the Threshold $\hat{\mu}$

Recall, the Algorithm 1 uses the following threshold to cluster the entries of $|\hat{v}|$ :

$$
\hat{\mu} = \frac{1}{d} \sum_{j=1}^{d} |\hat{v}_j|.
$$

**Fact:** For any vectors $v, \hat{v}$ of dimension $d$ the mean absolute error $|\mu - \hat{\mu}|$ between the average of magnitudes $\mu = d^{-1} \sum_{j=1}^{d} |v_j|$ and that of $\hat{v}$ satisfies

$$
|\mu - \hat{\mu}| \leq d^{-1/2} \min_{\theta \in \{-1, +1\}} \|v - \theta\hat{v}\|_2 \leq \min_{\theta \in \{-1, +1\}} \|v - \theta\hat{v}\|_\infty. \tag{43}
$$

**Proof 2**

$$
\hat{\mu} - \mu = \frac{1}{d} \sum_{j=1}^{d} (|\theta\hat{v}_j| - |v_j|) = \frac{1}{d} \sum_{j=1}^{d} (|\hat{v}_j| - |v_j|).
$$

*Taking the absolute value and using the triangle inequality, followed by the root mean square - arithmetic mean inequality,*

$$
|\hat{\mu} - \mu| \leq \frac{1}{d} \sum_{j=1}^{d} |\hat{v}_j - v_j| \leq d^{-1/2} \|\hat{v} - v\|_2 \leq \min_{\theta \in \{-1, +1\}} \|v - \theta\hat{v}\|_\infty.
$$

Using the above fact, we can relate the concentration of $\hat{\mu}$ with respect to $\mu(n^{-1}R_y) = \frac{1}{d}\sum_{j=1}^{d}|v_j(n^{-1}R_y)|$ as

$$|\hat{\mu} - \mu(n^{-1}R_y)| \leq \frac{1}{d}\sum_{j=1}^{d}|\hat{v}_j - v_j(n^1 R_y)| \leq d^{-1/2}\|\hat{v} - v(n^{-1}R_y)\|_2 \leq \min_{\theta \in \{-1,+1\}}\|v(n^{-1}R_y) - \theta\hat{v}\|_\infty. \quad (44)$$

### F.6.3 Proof of Proposition 3: Relating the Event of Perfect Clustering with Eigenvector Concentration

The Proposition 3 is an immediate implication of Proposition 4 and Equation 44. On the event $E_{l_\infty}$, using Equation 44, the following is satisfied : $|\hat{\mu} - \mu| < \frac{1}{2}\min\left\{m_e(n^{-1}R_y), m_h(n^{-1}R_y)\right\}$. Hence, the event $E_{\hat{v},j}$ is satisfied for all $j \in [n]$. Hence $\eta = 0$ is achieved from Proposition 4.

## G  Proof of Proposition 1: Error Rate Lower Bound of Type-Agnostic Weighted Majority Vote

We restate the proposition for easy reference:

**Restatement of Proposition 1**:

Let the WMV estimate using a single weight vector $w$ across all task $j$ is defined as:

$$\hat{y}_j^{WMV}(w) := \text{sgn}\left(\sum_{i=1}^{n} w_i X_{ij}\right), \forall j \in [d].$$

We consider weight vectors belonging to the set $w_l \leq |w_i| \leq w_u$ for all workers $i$ with $w_l$ and $w_u$ two positive constants such that $0 < w_l \leq w_u < \infty$. Under this construction, for any $y \in \{-1,+1\}^d$, the average labeling error rate for the type-agnostic WMV algorithm can be lower bounded as

$$\liminf_{n\to\infty} \frac{1}{n} \log \min_w \mathbb{E}\left(\frac{1}{d}\sum_j \mathbf{1}\left(\hat{y}_j^{WMV}(w) \neq y_j\right)\right)$$
$$\geq -\limsup_{n\to\infty} \max_w \min_k \varphi_n(w, r_k),$$

for any ground-truth vector $y \in \{-1,+1\}^d$ where the error exponent $\varphi_n(w, r_k)$ is given by

$$\varphi_n(w, r_k) = -\inf_{t\geq 0}\frac{1}{n}\sum_{i=1}^{n}\log\left(e^{tw_i}\frac{1-r_{ki}}{2} + e^{-tw_i}\frac{1+r_{ki}}{2}\right).$$

**Proof:**

This proof technique uses large deviation analysis on a sum of independent random variables (Srikant & Ying, 2013). Let us first fix a task index $j$ and let the type of that task be $k_j = k$ for some $k \in \{e, h\}$. For each worker $i$ and task $j$, let $G_{ij}$ be a random variable that takes the value $+1$ if worker $i$ correctly labels task $j$ and is $-1$ otherwise. In other words, $G_{ij} = y_j X_{ij}$, which is $+1$ with probability $\frac{1}{2}(1+r_{ki})$. Let the probability measure corresponding to type $k$ be denoted by $\mathbb{P}_k$ and we can write the probability of mislabeling task $j$ as:

$$\mathbb{P}_k\left(\hat{y}_j(w) \neq y_j\right) \geq \mathbb{P}_k\left(y_j\sum_{i=1}^{n}w_i X_{ij} < 0\right) = \mathbb{P}_k\left(\sum_{i=1}^{n}w_i G_{ij} < 0\right),$$

where the inequality follows from the observation that when $\sum_{i=1}^{n}w_i X_{ij} = 0$, we assign the label as $+1$. where we drop the superscript 'WMV' in this section from $\hat{y}_j^{WMV}(w)$. We notice that $\sum_{i=1}^{n}w_i G_{ij}$ can only take finitely many values and $\sum_{i=1}^{n}w_i G_{ij} \leq \sum_i |w_i|$. Consider the set $\mathbb{S} = \{s : s = \sum_i g_i, g_i \in \{-w_i, w_i\}\}$.

For any positive value of $S_k$ with $0 < S_k \leq \sum_i |w_i|$,

$$\mathbb{P}_k \left( \sum_{i=1}^n w_i G_{ij} < 0 \right) = \mathbb{P}_k \left( -\sum_{i=1}^n w_i G_{ij} > 0 \right) \geq \sum_{s \in \mathbb{S}: 0 < s < S_k} \mathbb{P}_k \left( -\sum_{i=1}^n w_i G_{ij} = s \right) \tag{45}$$

$$= \sum_{s \in \mathbb{S}: 0 < s < S_k} \sum_{\sum_i g_i = s, g_i \in \{-w_i, w_i\}} \prod_{i=1}^n \mathbb{P}_k \left( -w_i G_{ij} = g_i \right) \tag{46}$$

holds by the independence of the responses across workers. Now, we use a change of measure of the concerned random variable. Define a new random variable corresponding to each $i$ as $\tilde{G}_{ij}$ given by the following mass distribution for some $t_n(k) \geq 0$

$$Q_k^{t_n(k)} \left( \tilde{G}_{ij} = 1 \right) = \frac{(1 + r_{ki}) e^{-t_n(k) w_i}}{(1 + r_{ki}) e^{-t_n(k) w_i} + (1 - r_{ki}) e^{t_n(k) w_i}},$$

$$Q_k^{t_n(k)} \left( \tilde{G}_{ij} = -1 \right) = \frac{(1 - r_{ki}) e^{t_n(k) w_i}}{(1 + r_{ki}) e^{-t_n(k) w_i} + (1 - r_{ki}) e^{t_n(k) w_i}}.$$

Then we can express Equation 46 as

$$\sum_{s \in \mathbb{S}: 0 < s < S_k} \sum_{\sum_i g_i = s, g_i \in \{-w_i, w_i\}} \prod_{i=1}^n \mathbb{P} \left( -w_i G_{ij} = g_i \right)$$

$$\geq Q_k^{t_n(k)} \left( 0 < -\sum_i w_i \tilde{G}_{ij} < S_k \right) \frac{\prod_{i=1}^n \left( (1 + r_{ki}) e^{-t_n(k) w_i} + (1 - r_{ki}) e^{t_n(k) w_i} \right)}{2 e^{t_n(k) S_k}}$$

where to obtain the last step above, we have multiplied and divided each term in the product by $\frac{2 e^{t_n(k) g_i}}{(1 + r_{ki}) e^{-t_n(k) w_i} + (1 - r_{ki}) e^{t_n(k) w_i}}$ and used the bound $\sum_{i=1}^n g_i \leq S_k$. Recall the expression

$$\varphi_n(w, r_k) = -\inf_{t \geq 0} \frac{1}{n} \sum_i \log \left( \frac{1}{2} \left( (1 + r_{ki}) e^{-t w_i} + (1 - r_{ki}) e^{t w_i} \right) \right), \tag{47}$$

Define $t_n(k) = t_n^*(k)$ to be a minimizing argument of $\frac{1}{n} \sum_i \log \left( \frac{1}{2} \left( (1 + r_{ki}) e^{-t_n(k) w_i} + (1 - r_{ki}) e^{t_n(k) w_i} \right) \right)$ in the domain $t_n(k) \geq 0$. Now, putting the minimizing argument $t_n^*(k)$ in the place of $t_n(k)$ we obtain a lower bound for type $k$ as

$$\mathbb{P}_k \left( \hat{y}_j \neq y_j \right) \geq Q_k^{t_n^*(k)} \left( 0 < -\sum_i w_i \tilde{G}_{ij} < S_k \right) e^{-n \varphi_n(w, r_k) - t_n^*(k) S_k}.$$

Noting that the distribution of the random variable $\tilde{G}_{i,j}$ is invariant to task index $j$, we drop the index $j$ in the subsequent bounds on the error rate for positive values $S_k, \forall k \in \{e, h\}$ (note that the following holds for all $y$):

$$\mathbb{E} \left( \frac{1}{d} \sum_j \mathbf{1} \left( \hat{y}_j^{WMV}(w) \neq y_j \right) \right) \geq \sum_{k \in \{e, h\}} \frac{d_k}{d} Q_k^{t_n^*(k)} \left( 0 < -\sum_i w_i \tilde{G}_i < S_k \right) e^{-n \varphi_n(w, r_k) - t_n^*(k) S_k}. \tag{48}$$

To analyze this further, use the following Lemma on the distribution of $-\sum_i w_i \tilde{G}_i$, an extension to the asymptotic analysis of majority voting in Gao et al. (2016).

Recall our definition $\rho \leq \min_i \frac{1 + r_{ki}}{2} \leq 1 - \rho, \forall k \in \{e, h\}$. The following lemma is similar to Lemma 6.3 in Gao et al. (2016). The proof is given next to it for completeness.

**Lemma 6** *Let $\rho \leq \min_i \frac{1 \pm r_{ki}}{2} \leq 1 - \rho, \forall k \in \{e, h\}$, for some $\rho \in (0, 1/2)$.*

1. Let $t_n^*(k)$ be the minimizer of $\frac{1}{n}\sum_i \log\left(\frac{1}{2}\left((1+r_{ki})e^{-tw_i} + (1-r_{ki})e^{tw_i}\right)\right)$ . Then, $0 \leq t_n^*(k) < -\frac{n}{\|w\|_1}\log\rho, k \in \{e, h\}, \forall n \geq 1$.

2. For any $y \in \{\pm 1\}$ and any $t_n(k) \geq 0$,

$$\frac{\sum_{i=1}^n \left(-w_i\tilde{G}_i - \mathbb{E}_{Q_k^{t_n(k)}}[-w_i\tilde{G}_i]\right)}{\sqrt{\operatorname{Var}_{Q_k^{t_n(k)}}(-\sum_{i=1}^n w_i\tilde{G}_i)}} \xrightarrow[n\to\infty]{d} \mathcal{N}(0,1), \qquad \text{under the measure } Q_k^{t_n(k)}.$$

Moreover, at $t_n(k) = t_n^*(k)$,

$$\frac{-\sum_{i=1}^n w_i\tilde{G}_i}{\sqrt{\operatorname{Var}_{Q_k^{t_n^*(k)}}(-\sum_{i=1}^n w_i\tilde{G}_i)}} \xrightarrow[n\to\infty]{d} \mathcal{N}(0,1), \qquad \text{under the measure } Q_k^{t_n^*(k)}.$$

**Proof 3** *1. Let*

$$\beta_k(t_n(k)) = \prod_{i=1}^n \frac{1}{2}\left[(1+r_{ki})e^{-t_n(k)w_i} + (1-r_{ki})e^{t_n(k)w_i}\right].$$

*Then $\beta_k(0) = 1$ and $\forall t_n(k) \geq -\frac{n}{\|w\|_1}\log\rho$, we have that $\beta_k(t_n(k)) > \prod_{i=1}^n \left(\rho e^{t_n(k)|w_i|}\right) \geq 1$. Therefore, $t_n^*(k) \in \left[0, -\frac{n}{\|w\|_1}\log\rho\right)$.*

*2. For the second part, we use Lindeberg's condition for the Central Limit Theorem for the expression $\sum_{i=1}^n -w_i\tilde{G}_i$. The Lindeberg's condition in this context corresponds to*

$$\lim_{n\to\infty} \frac{\sum_{i=1}^n \mathbb{E}_{Q_k^{t_n(k)}}\left[\left(-w_i\tilde{G}_i - \mathbb{E}_{Q_k^{t_n(k)}}[-w_i\tilde{G}_i]\right)^2 \mathbf{1}\left\{\left|-w_i\tilde{G}_i - \mathbb{E}_{Q_k^{t_n(k)}}[-w_i\tilde{G}_i]\right| > \epsilon\sqrt{\operatorname{Var}_{Q_k^{t_n(k)}}\left(\sum_{i=1}^n -w_i\tilde{G}_i\right)}\right\}\right]}{\operatorname{Var}_{Q_k^{t_n(k)}}\left(\sum_{i=1}^n -w_i\tilde{G}_i\right)}$$

$$= 0, \forall \epsilon > 0.$$

*A direct calculation gives*

$$\mathbb{E}_{Q_k^{t_n(k)}}[-w_i\tilde{G}_i] \underbrace{=}_{(a)} w_i \frac{(1-p_{ki})e^{t_n(k)w_i} - p_{ki}e^{-t_n(k)w_i}}{(1-p_{ki})e^{t_n(k)w_i} + p_{ki}e^{-t_n(k)w_i}}$$

$$= \frac{\frac{d}{dt_n(k)}\left[(1-p_{ki})e^{t_n(k)w_i} + p_{ki}e^{-t_n(k)w_i}\right]}{(1-p_{ki})e^{t_n(k)w_i} + p_{ki}e^{-t_n(k)w_i}}$$

$$= \frac{d}{dt_n(k)}\log\left((1-p_{ki})e^{t_n(k)w_i} + p_{ki}e^{-t_n(k)w_i}\right),$$

*where in $(a)$, we used the following relation : $p_{ki} = \frac{1+r_{ki}}{2}, \forall k \in e, h, i \in [n]$.*

*The last two equalities imply: at $t_n(k) = t_n^*(k)$, $\mathbb{E}_{Q_k^{t_n(k)}}\left[\sum_{i=1}^n -w_i\tilde{G}_i\right] = 0$. Moreover, $\mathbb{E}_{Q_k^{t_n(k)}}[(-w_i\tilde{G}_i)^2] = w_i^2$. Therefore,*

$$\operatorname{Var}_{Q_k^{t_n(k)}}(-w_i\tilde{G}_i) = w_i^2\left[1 - \frac{[(1-p_{ki})e^{t_n(k)w_i} - p_{ki}e^{-t_n(k)w_i}]^2}{[(1-p_{ki})e^{t_n(k)w_i} + p_{ki}e^{-t_n(k)w_i}]^2}\right]$$

$$= w_i^2 \frac{4p_{ki}(1-p_{ki})}{[(1-p_{ki})e^{t_n(k)w_i} + p_{ki}e^{-t_n(k)w_i}]^2}$$

$$\geq \frac{4w_i^2\rho^2}{(1-\rho)^2[e^{t_n(k)w_i} + e^{-t_n(k)w_i}]} \geq \frac{2w_i^2\rho^2}{(1-\rho)^2 e^{t_n(k)|w_i|}} \geq \frac{2w_l^2\rho^2}{(1-\rho)^2 e^{t_n(k)w_u}},$$

*and hence, $\operatorname{Var}_{Q_k^{t_n(k)}}(\sum_{i=1}^n -w_i\tilde{G}_i) \geq n\frac{2w_l^2\rho^2}{(1-\rho)^2 e^{t_n(k)w_u}} \to \infty$ as $n \to \infty$.*

Additionally, $\left|-w_i\tilde{G}_i - \mathbb{E}_{Q_k^{t_n(k)}}[-w_i\tilde{G}_i]\right| \leq 2|w_i| \leq 2w_u$ almost surely (and therefore, $\text{Var}_{Q_k^{t_n(k)}}(-w_i\tilde{G}_i) \leq 4w_u^2$). Thus, for every $\epsilon > 0$ we have that

$$\mathbf{1}\left\{\left|w_i\tilde{G}_i - \mathbb{E}_{Q_k^{t_n(k)}}[-w_i\tilde{G}_i]\right| > \epsilon\sqrt{\text{Var}_{Q_k^{t_n(k)}}\left(\sum_{i=1}^n -w_i\tilde{G}_i\right)}\right\} = 0, \quad almost\ surely$$

for $n > \frac{2w_u^2(1-\rho)^2 e^{t_n(k)w_u}}{\epsilon^2 w_l^2 \rho^2}$. Lindeberg's condition now follows.

**Remark 3** We can see that $\text{Var}_{Q_k^{t_n^*(k)}}(-w_i\tilde{G}_i) > 0$ as $t_n^*(k) < -\frac{n}{\|w\|_1}\log\rho$.

Now, let us go back to proving the lower bound. We have the following:

$$\mathbb{E}\left(\frac{1}{d}\sum_j \mathbf{1}\left(\hat{y}_j^{WMV}(w) \neq y_j\right)\right) \geq \sum_{k\in\{e,h\}}\frac{d_k}{d}Q_k^{t_n^*(k)}\left(0 < -\sum_i w_i\tilde{G}_i < S_k\right)e^{-n\varphi_n(w,r_k) - t_n^*(k)S_k}.$$

Setting $S_k = \sqrt{\text{Var}_{Q_k^{t_n^*(k)}}(\sum_i -w_i\tilde{G}_{ij})}$, we write the following

$$Q_k^{t_n^*(k)}\left(0 < -\sum_i w_i\tilde{G}_i < S_k\right)$$

$$= Q_k^{t_n^*(k)}\left(0 < -\sum_i w_i\tilde{G}_i < \sqrt{\text{Var}_{Q_k^{t_n^*(k)}}\left(\sum_i -w_i\tilde{G}_{ij}\right)}\right)$$

$$\underbrace{=}_{(b)} Q_k^{t_n^*(k)}\left(0 < \frac{-\sum_i w_i\tilde{G}_i}{\sqrt{\text{Var}_{Q_k^{t_n^*(k)}}(\sum_i -w_i\tilde{G}_{ij})}} < 1\right).$$

In $(b)$, we use Remark 3 from the proof of Lemma 6: $\sqrt{\text{Var}_{Q_k^{t_n^*(k)}}\left(\sum_i -w_i\tilde{G}_{ij}\right)} > 0$ at $t_n(k) = t_n^*(k)$. Also,

$$\exp\left(-n\varphi_n(w,r_k) - t_n^*(k)S_k\right) = \exp\left(-n\varphi_n(w,r_k) - t_n^*(k)\sqrt{\text{Var}_{Q_k^{t_n^*(k)}}\left(\sum_i -w_i\tilde{G}_i\right)}\right).$$

Evaluating $\text{Var}_Q(\sum_i -w_i\tilde{G}_i) \leq \sum_i w_i^2$ and using the following bounds on the entries of $w$ : $w_l \leq |w_i| \leq w_u, \forall i \in [n]$, and using the upper-bound on $t_n^*(k)$ from the Lemma 6,

$$\exp\left(-n\varphi_n(w,r_k) - t_n^*(k)\sqrt{\text{Var}_{Q_k^{t_n^*(k)}}\left(\sum_i -w_i\tilde{G}_i\right)}\right) \geq \exp\left(-n\frac{\|w\|_2|\log(\rho)|}{\|w\|_1} - n\varphi_n(w,r_k)\right)$$

$$\geq \exp\left(-\sqrt{n}\frac{w_u|\log(\rho)|}{w_l} - n\varphi_n(w,r_k)\right).$$

Putting it all together, We can write from Equation 48,

$$\mathbb{E}\left(\frac{1}{d}\sum_j \mathbf{1}\left(\hat{y}_j^{WMV}(w) \neq y_j\right)\right)$$

$$\geq \sum_{k\in\{e,h\}}\frac{d_k}{d}Q_k^{t_n^*(k)}\left(0 < \frac{-\sum_i w_i\tilde{G}_i}{\sqrt{\text{Var}_{Q_k^{t_n^*(k)}}(\sum_i -w_i\tilde{G}_{ij})}} < 1\right)\exp\left(-\sqrt{n}\frac{w_u|\log(\rho)|}{w_l} - n\varphi_n(w,r_k)\right)$$

$$\geq \min_k\frac{d_k}{d}\exp\left(-\sqrt{n}\frac{w_u|\log(\rho)|}{w_l} - n\min_k\varphi_n(w,r_k)\right)\min_k Q_k^{t_n^*(k)}\left(0 < \frac{\sum_i -w_i\tilde{G}_i}{\sqrt{\text{Var}_{Q_k^{t_n^*(k)}}(\sum_i -w_i\tilde{G}_{ij})}} < 1\right).$$

By first taking a minimum over weight vector $w$ and then taking the $\liminf$ as $n \to \infty$ and using the Lemma 6,

$$\liminf_{n \to \infty} \frac{1}{n} \log \mathbb{E} \left( \frac{1}{d} \sum_j \mathbf{1} \left( \hat{y}_j^{WMV}(w) \neq y_j \right) \right) \geq -\limsup_{n \to \infty} \max_w \min_k \varphi_n(w, r_k).$$

## H    Proof of Proposition 2

**Restatement of Proposition 2:** Assume $V_k = min_i max_{a,b \neq i} \sqrt{|r_{ka} r_{kb}|} > 0$ for each $k \in \{e, h\}$ which is satisfied if there are at least two workers with non-zero reliability values for each type. If the number of workers $n$ satisfies $n \geq \sqrt{3\rho/\bar{r}}$, and the number of tasks per type satisfies

$$d_k \geq C_5 \frac{n^2}{V_k^4 \min(\rho^2, \bar{r}^2)} \left( n\Phi_n(r_k) + \log(6n^2) \right).$$

for some universal constant $C_5$ then, the TE algorithm to estimate the reliability vectors followed by NP-WMV for label estimation separately for each type (when type information is known) achieves a labeling error rate satisfying

$$\mathbb{E} \left( \frac{1}{d} \sum_j \mathbf{1} \left( \hat{y}_j \neq y_j \right) \right) \leq 3 \sum_{k \in \{e,h\}} \frac{d_k}{d} \exp \left( -n\Phi_n(r_k) \right),$$

where $\hat{y}_j$ and $y_j$ are the estimated and true labels of task $j$, respectively, and

$$\Phi_n(r_k) = -\frac{1}{n} \sum_{i=1}^{n} \log \left( \sqrt{(1 + r_{ki})(1 - r_{ki})} \right).$$

**Proof:**

The statement is obtained by appropriately modifying Theorem 4.1 in Gao et al. (2016) and Theorem 2 in Bonald & Combes (2017). For the known type case, we separate the tasks according to their type, and each type is dealt with separately as two Dawid-Skene problem instances. Because task types are known for this setting, the TE algorithm is applied separately to each type independently of the other to estimate each type's reliability vectors. Labels are estimated using the corresponding NP-WMV.

From Theorem 2 [3] in Bonald & Combes (2017)[4], we have that if the number of workers $n$ satisfies

$$n^2 \geq \frac{3\rho}{\bar{r}} \tag{49}$$

and the number of tasks $d_k$ per type $k \in \{e, h\}$ is

$$d_k \geq \max \left( 120 \times 24^2 \frac{n^2}{V_k^4 \rho^2} (n\Phi_n(r_k) + \log(6n^2)), 30 \times 8^2 \frac{n}{V_k^2 \bar{r}^2} (n\Phi_n(r_k) + \log(4n^2)) \right), \tag{50}$$

then

$$\mathbb{P} \left( \|\hat{r}_k - r_k\|_\infty \geq \frac{\rho}{n} \right) \leq \exp(-n\Phi_n(r_k)). \tag{51}$$

The sufficient condition of $d$ can also be written as

$$d_k \geq C_5 \frac{n^2}{V_k^4 \min(\rho^2, \bar{r}^2)} \left( n\Phi_n(r_k) + \log(6n^2) \right)$$

---

[3]According to the TE algorithm described in Section B, the estimated reliabilities are projected onto the set $\rho \leq \frac{1+\hat{r}_{ki}}{2} \leq 1 - \rho$. This step was not included in the original TE algorithm proposed by Bonald & Combes (2017). Nevertheless, the concentration of the reliability estimate derived from Theorem 2 of Bonald & Combes (2017) in the max-norm sense also holds under this projection, as it acts as a contraction operator.

[4]One difference between our model and the model considered in Bonald & Combes (2017) is that we consider the true labels as deterministic quantity and Bonald & Combes (2017) considers them to be random variables. The TE algorithm uses the worker-similarity matrix and we can easily show that the worker-similarity matrix is independent of the true labels and thus the performance bound on the TE algorithm in Theorem 2 in Bonald & Combes (2017) is valid for deterministic labels

with $C_5 = 15 \times 2^9$.

Using the inequality $|\log x - \log y| \leq \frac{|x-y|}{\min\{x,y\}}, \forall x, y > 0$ implied by $\log x \leq x - 1$ for positive $x$, we have that when $d_k$ satisfies Equation 50,

$$\sum_i \max\left\{\left|\log \frac{1+\hat{r}_{ki}}{1+r_{ki}}\right|, \left|\log \frac{1-\hat{r}_{ki}}{1-r_{ki}}\right|\right\} \leq \frac{1}{2}$$

with probability $\geq 1 - \exp(-n\Phi_n(r_k))$. Now define the event

$$E_k := \left\{\sum_i \max\left(\left|\log \frac{1+\hat{r}_{ki}}{1+r_{ki}}\right|, \left|\log \frac{1-\hat{r}_{ki}}{1-r_{ki}}\right|\right) \leq \frac{1}{2}\right\}.$$

Under this event, the weights used by the NP estimate are approximately equal to the maximum likelihood weights. Applying Equation 51,

$$\mathbb{P}(E_k^c) \leq \exp(-n\Phi_n(r_k)).$$

Without loss of generality, consider $y_j = 1$ so that $\hat{y}_j \neq y_j$ implies $\hat{y}_j = -1$. Let the type for task $j$ be $k_j$.

$$\mathbb{P}(\hat{y}_j \neq y_j) \leq \mathbb{P}(\{\hat{y}_j \neq y_j\} \cap E_{k_j}) + \mathbb{P}(E_{k_j}^c)$$

$$= \mathbb{P}\left(\left\{\sum_i \left(\log \frac{1+\hat{r}_{k_j i}}{1-\hat{r}_{k_j i}} X_{ij}\right) < 0\right\} \cap E_{k_j}\right) + \mathbb{P}(E_{k_j}^c)$$

$$= \mathbb{P}\left(\left\{\prod_i \left(\frac{1-\hat{r}_{k_j i}}{1+\hat{r}_{k_j i}}\right)^{\mathbf{1}(X_{ij}=1)} \left(\frac{1+\hat{r}_{k_j i}}{1-\hat{r}_{k_j i}}\right)^{\mathbf{1}(X_{ij}=-1)} \geq 1\right\} \cap E_{k_j}\right) + \mathbb{P}(E_{k_j}^c). \qquad (52)$$

Define the two random variables

$$A_1 = \prod_i \left(\frac{1-r_{k_j i}}{1+r_{k_j i}}\right)^{\mathbf{1}(X_{ij}=1)} \left(\frac{1+r_{k_j i}}{1-r_{k_j i}}\right)^{\mathbf{1}(X_{ij}=-1)}$$

$$A_2 = \prod_i \left(\frac{(1-\hat{r}_{k_j i})(1+r_{k_j i})}{(1-r_{k_j i})(1+\hat{r}_{k_j i})}\right)^{\mathbf{1}(X_{ij}=1)} \left(\frac{(1+\hat{r}_{k_j i})(1-r_{k_j i})}{(1+r_{k_j i})(1-\hat{r}_{k_j i})}\right)^{\mathbf{1}(X_{ij}=-1)}.$$

Then, the expression $\prod_i \left(\frac{1-\hat{r}_{k_j i}}{1+\hat{r}_{k_j i}}\right)^{\mathbf{1}(X_{ij}=1)} \left(\frac{1+\hat{r}_{k_j i}}{1-\hat{r}_{k_j i}}\right)^{\mathbf{1}(X_{ij}=-1)}$ in the above probability is given by the product of $A_1$ and $A_2$. On the event $E_{k_j}$,

$$A_2 \leq \exp\left(2\sum_i \max\left(\left|\log \frac{1+\hat{r}_{k_j i}}{1+r_{k_j i}}\right|, \left|\log \frac{1-\hat{r}_{k_j i}}{1-r_{k_j i}}\right|\right)\right) \leq \exp(1).$$

Therefore,

$$\mathbb{P}\left(\{A_1 A_2 \geq 1\} \cap E_{k_j}\right) \leq \mathbb{P}\left(\{A_1 \geq \exp(-1)\} \cap E_{k_j}\right)$$

$$\underbrace{\leq}_{(a)} \mathbb{P}\left(\left\{A_1^{\frac{1}{2}} \geq \exp\left(-\frac{1}{2}\right)\right\} \cap E_{k_j}\right)$$

$$\leq \mathbb{P}\left(\left\{A_1^{\frac{1}{2}} \geq \exp\left(-\frac{1}{2}\right)\right\}\right)$$

$$\underbrace{\leq}_{(b)} \exp\left(\frac{1}{2}\right) \mathbb{E}[A_1^{1/2}],$$

where in $(a)$ we used the observation that $A_1 > 0$ and in $(b)$ we used Markov's inequality on the random variable $A_1^{1/2} > 0$. Evaluating the expectation,

$$\mathbb{E}[A_1^{\frac{1}{2}}] = \prod_i \left[ \left( \frac{1 - r_{k_j i}}{1 + r_{k_j i}} \right)^{\frac{1}{2}} \frac{1 + r_{k_j,i}}{2} + \left( \frac{1 + r_{k_j i}}{1 - r_{k_j i}} \right)^{\frac{1}{2}} \frac{1 - r_{k_j,i}}{2} \right]$$

$$= \exp \left( \frac{1}{2} \sum_i \log \left( 1 - r_{ki}^2 \right) \right) = \exp(-n\Phi_n(r_k)).$$

Returning to Equation 52, we have that for a task $j$ with type $k$,

$$\mathbb{P}_k \left( \hat{y}_j \neq y_j \right) \leq \exp \left( \frac{1}{2} \right) \mathbb{E}[A_1^{1/2}] + \mathbb{P}\left(E_k\right) \leq \left( \exp \left( \frac{1}{2} \right) + 1 \right) \exp \left( -n\Phi_n(r_k) \right).$$

Averaging for all tasks $j \in [d]$, we get the error rate for known types as

$$\mathbb{E} \left( \frac{1}{d} \sum_j \mathbf{1} \left( \hat{y}_j \neq y_j \right) \right) = \frac{1}{d} \sum_{j=1}^d \mathbb{P}\left( \hat{y}_j \neq y_j \right) \leq 3 \sum_{k \in \{e,h\}} \frac{d_k}{d} \exp \left( -n\Phi_n(r_k) \right).$$

## I  Label Estimation Performance: Proof of Theorem 2

The expected rate of labeling error using the law of total expectation can be decomposed as :

$$\mathbb{E} \left( \frac{1}{d} \sum_j \mathbf{1} \left( \hat{y}_j \neq y_j \right) \right) = \underbrace{\mathbb{E} \left( \frac{1}{d} \sum_j \mathbf{1} \left( \hat{y}_j \neq y_j \right) | E_{pc} \right) \mathbb{P}(E_{pc})}_{I} + \underbrace{\mathbb{E} \left( \frac{1}{d} \sum_j \mathbf{1} \left( \hat{y}_j \neq y_j \right) | E_{pc}^c \right) \mathbb{P}(E_{pc}^c)}_{II}$$

where $E_{pc}$ is defined as the event of perfect clustering, that is $\eta = 0$. We upper bound the second term $II$ as $II \leq \mathbb{P}(E_{pc}^c)$. Now, when the condition described in the Equation 10 of Theorem 1 in the main paper is satisfied by $(n_{cl}, d, r_k(\mathcal{N}_{cl}))$ for each $k \in \{e, h\}$ it is characterized by Theorem 1 as :

$$\mathbb{P}(E_{pc}^c) \leq 2d^2 \exp \left( -C_2 n_{cl} D(r_e(\mathcal{N}_{cl}), r_h(\mathcal{N}_{cl}), \alpha, d) \right).$$

To upper bound the term $I$, we invoke the Proposition 2. Recall the partition of the set of workers to mutually exclusive sets $\mathcal{N}_{cl}$ and $\mathcal{N}_{rl}$ for clustering and the labeling steps, respectively. Hence, given the event $E_{pc}$, the labeling step has perfect knowledge of each task's type, and $NP - WMV$ for the known type model would yield the following error rate when $(n_{rl}, d_e, d_h, r_k(\mathcal{N}_{rl}))$ satisfy the conditions stated in Proposition 2:

$$I \leq 3 \sum_{k \in \{e,h\}} \frac{d_k}{d} \exp \left( -n_{rl}\Phi_{k,\mathcal{N}_{rl}} \right).$$

