# OpenReview forum: "Spectral Clustering and Labeling for Crowdsourcing with Inherently Distinct Task Types"
_TMLR — Accepted by TMLR_

### Review · Reviewer_eMiz · 2025-05-05

**Summary Of Contributions:**

This paper addresses the problem of crowdsourcing when tasks inherently fall into “easy” and “hard” types. Building on the classical Dawid–Skene one-coin model, the authors propose a two-type DS model in which each worker has separate reliability parameters for each task type. The authors first introduce a simple spectral method to cluster tasks by types , with a theoretical guarantee that O(log d) workers suffice for perfect clustering. Then, the authors provide an $ℓ_∞$-norm perturbation bound enabling a perfect-recovery result. The proposed method is validated on five public datasets, showing that clustering followed by DS-style algorithms reduces error compared to DS alone and outperforms other task-specialization methods.

**Audience:**

Yes

**Broader Impact Concerns:**

The authors did not provide a discussion related to broader impact. The authors might want to discuss the potential benefits of the method such as reduce labeling costs or whether the method is likely to mitigate or introduce new bias in crowdsourcing analysis.

**Claims And Evidence:**

Yes

**Requested Changes:**

Can the authors provide evidence whether real-world datasets can be separate into "easy" and "hard" clusters? It would be beneficial if the authors can illustrate a clear gap between "easy" and "hard" clusters in real-world datasets.

Section 3 contains a lot of technical details. It would be beneficial if the authors can provide more commentary after each lemma or theory, explaining what is proven in plain English. This section also involves many mathematical symbols, and it would be beneficial if the authors can summarize the symbols in a table.

**Strengths And Weaknesses:**

Strengths

The authors provide theoretical guarantees for the proposed two-step model.

Evaluation on multiple real and synthetic datasets demonstrates the benefit of clustering, and comparisons to alternative specialization models bolster the empirical case.

Weaknesses

The model assumes every worker labels every task, which may not hold in most real-world crowdsourcing settings.

The authors assume that the tasks can be separated into  “easy” and “hard” clusters. However, the authors did not provide evidence whether this assumption holds in the real-world dataset. I suspect whether there exists a clear gap between "easy" and "hard" clusters or there might exist more than two clusters in the real-world datasets.

---

> ### Author Response · Authors · 2025-05-07
> **Clarification on Worker-Task Assumptions, Illustrating Task Type Separation, and Presentation Improvements**
>
> We thank the reviewer for the thoughtful review and valuable suggestions. Below, we address the concerns raised and outline the changes we plan to incorporate in the revised version of the paper.
>
> **Worker-Task Sparsity Assumption**
>
> We acknowledge the reviewer’s concern about the assumption that each worker labels every task. As stated in the paper:
>
> "It is worth noting that our model assumes all workers respond to all tasks, as it is motivated by applications where an institution contracts professionals to label a dataset. In this paper, we are not interested in applications that use platforms such as Amazon Mechanical Turk, in which workers independently select a sparse subset of tasks to label."
>
> Our primary goal is to theoretically establish a novel perfect clustering result in multi-type crowdsourcing settings under practically motivated assumptions on worker-task complexity. While extending the theory to handle sparsity is an important direction, we deliberately chose to focus on the fully connected setting to emphasize the core contribution. However, in our experiments, we do analyze sparse real-world datasets. To fit them into our model, we preprocess the sparse response matrices by imputing missing entries using only the observed labels (without access to ground truth), consistent with common approaches in the literature. We then evaluate the performance of our clustering-based pipeline against existing algorithms designed for multi-type crowdsourcing.
>
> **Easy-Hard Task Separation in Real-World Datasets**
>
> We appreciate the reviewer’s question regarding the presence of a clear easy-hard or multi-type separation in real-world datasets. In our experience, real-world datasets tend to reflect multi-type task behavior more frequently than a single-type DS model. Our motivation is best illustrated through the JSRT[1] dataset, as described in Appendix Section C.1:
>
> "Expert performances are reported for various levels of subtlety defined by the size of nodular patterns. It is clear that detecting nodular patterns becomes significantly more difficult as the size is decreased, demonstrating a multi-type phenomenon with varying levels of task difficulty."
>
> We placed this example in the appendix because the response matrix for JSRT[1] was synthetically generated. As noted in the introduction, access to real-world healthcare datasets with both ground truth and full response information is highly limited due to privacy concerns.
>
> To provide evidence of multi-type structure in the real-world datasets used in our main experiments, we have added a plot of the eigenvalue spectrum of the (preprocessed) response matrices in Appendix Section C.3. As shown, each dataset exhibits at least two dominant eigenvalues, with the remaining eigenvalues close to zero. This spectral gap is a strong indicator of the presence of multiple task types.
>
> While our algorithm can in principle be extended to more than two types (e.g., via k-means clustering on the top eigenvectors), accurate label estimation in such settings would require the reliability vectors of each type to be sufficiently distinct, and a larger number of responses to estimate them reliably. In our experiments, we found that modeling these datasets using a two-type separation offers a practical and effective trade-off between model complexity and performance. Exploring data-driven methods for estimating the number of task types is an exciting direction for future work.
>
> **Requested Revisions to Improve Clarity**
>
> Thank you for the helpful suggestions regarding presentation. We will incorporate the following changes in the revised version:
>
> A table of notation to summarize key mathematical symbols used in the paper.
>
> Further plain-English explanations following each technical lemma and theorem in Section 3, to clarify their implications and intuition.
>
> **Broader Impact Statement**
>
> We thank the reviewer for the suggestion to elaborate on the broader impact of our work. In the revised version, we will include the following updated Broader Impact Statement:
>
> In accordance with the TMLR guidelines on potential societal impacts and ethical conduct, the authors are not aware of any direct or indirect negative implications of this work. The proposed method may help reduce labeling costs in crowdsourcing by enabling more accurate label aggregation in heterogeneous task settings. While we adopt a two-type model in this work, our findings suggest that this modeling approach captures meaningful structure in real datasets, and may serve as a foundation for more general task-type inference in future work.
>
>
> *[1] Junji Shiraishi, Shigehiko Katsuragawa, Junpei Ikezoe, Tsuneo Matsumoto, Takeshi Kobayashi, Ken-ichi
> Komatsu, Mitate Matsui, Hiroshi Fujita, Yoshie Kodera, and Kunio Doi. Development of a digital image
> database for chest radiographs with and without a lung nodule. American Journal of Roentgenology*

---

> ### Author Response · Authors · 2025-07-09
> **Note on Revised Submission and Request for Further Review**
>
> Dear Reviewer eMiz,
>
> We hope our previous responses to your comments were helpful and addressed your concerns satisfactorily. Following that exchange, we have submitted a revised version of the paper, with all changes highlighted in blue. A summary of the key updates is included in the revision, but we would also like to take this opportunity to highlight specific changes made in response to your suggestions.
>
> In particular, based on your suggestions, we added a notation table, along with a geometric interpretation of the main clustering result to aid understanding. And, we have elaborated on and expanded the discussion in the broader impact section.
>
> We respectfully request that you evaluate our paper based on this revised version. Please let us know if you have any further suggestions or concerns—we would greatly appreciate your feedback.
>
> Looking forward to hearing from you.

---

### Review · Reviewer_Yfhp · 2025-05-11

**Summary Of Contributions:**

The paper introduces a new crowdsourcing model where each task belongs to one of two latent types (easy or hard), and each worker has type-dependent reliability. It proposes a spectral clustering method that recovers task types using the principal eigenvector of a task similarity matrix, followed by type-aware label aggregation using the Dawid-Skene model. Theoretical analysis shows that perfect clustering is achievable with high probability when the number of workers scales logarithmically with the number of tasks, and that clustering improves label estimation. Experiments on synthetic and real datasets demonstrate that the proposed method outperforms type-agnostic baselines and other task-type–aware approaches.

**Audience:**

Yes

**Broader Impact Concerns:**

I have no specific broader impact concerns.

**Claims And Evidence:**

Yes

**Requested Changes:**

**1. Clarify Novelty and Contributions**

 The authors could clarify how their model and algorithm differ from prior work, particularly those that already address type-dependent worker reliability and spectral clustering (e.g., Kaito et al., 2024, and collaborative clustering literature). It would be helpful to highlight any specific theoretical challenges or insights that distinguish this work from standard spectral clustering analysis.

**2. Justify and Generalize Modeling Assumptions**

The paper could provide stronger justification for its simplifying assumptions, particularly the restriction to two task types and complete worker-task assignment. It would also improve the paper to include discussion on how these assumptions align with real-world crowdsourcing settings, and how the method might generalize to more realistic cases (e.g., multiple or continuous task types, sparse worker-task assignment). A toy experiment or illustrative extension to more than two task types could help support this discussion.

**3. Evaluate Robustness to Sparse Assignments**

The authors could include experiments where workers label only a subset of tasks to examine how performance changes when the complete assignment assumption is relaxed. This would help assess the method’s robustness and practical relevance in realistic settings.


**4. Improve Writing and Organization**

The authors could revise the paper for clarity, especially in the theoretical sections, to make the analysis and key ideas easier to follow. Reorganizing some sections would also help improve logical flow and readability.


**5. Expand and Update Related Work**

The related work section could be expanded to include recent and relevant literature on task heterogeneity, adaptive querying, and clustering in crowdsourcing. This would help position the contribution more clearly within the broader research landscape.

**Strengths And Weaknesses:**

# Strengths

 - **Theoretical analysis**: The paper provides a formal guarantee that perfect task-type clustering is achievable with high probability under a logarithmic worker-to-task ratio, using spectral methods, while a set of impractical assumptions are required.

 - **Empirical validation**: The method is evaluated on small-scale synthetic and real-world datasets, showing improved labeling accuracy compared to type-agnostic baselines.

# Weaknesses

 - **Unclear novelty**: The modeling assumptions (type-dependent worker accuracy), spectral clustering approach, and analysis closely follow existing literature, with unclear distinctions from prior work, e.g., [Kaito et al 24] Kaito et al. "Optimal Clustering from Noisy Binary Feedback", Machine Learning, 2024. Also, the spectral analysis follows known techniques, and the paper does not clearly identify new technical challenges or insights.

 - **Unrealistic assumptions**: The setting assumes only two task types and that all workers label all tasks, which is not practical in real-world crowdsourcing platforms. Indeed, the crowdsourcing systems are employed for constructing large-scale datasets. These assumptions need to be thoroughly justified. At least, regarding the complete assignments of all tasks to all workers, the authors could provide an empirical evaluation given spare assignments investigating the importance of the assumption. In addition, the method does not handle more general task heterogeneity (e.g., more than two types or continuous difficulty), unlike recent works such as [Kaito et al. 24].

 - **Poor writing and organization**: The paper is hard to follow in several places, and the related work section is incomplete and outdated.

---

> ### Author Response · Authors · 2025-05-13
> **Clarifying Novel Contributions, Assumptions, and Practicality of Our Clustering Approach**
>
> We thank the reviewer for their thoughtful feedback and valuable suggestions. We address each of the raised points below.
>
> **1. Novelty and Contribution**
> The primary novelty of our paper lies in establishing a perfect clustering guarantee in a multi-type crowdsourcing model, using only $log(d)$ workers, where d is the number of tasks.
> A key technical challenge in proving perfect clustering lies in controlling the $l_\infty$-norm perturbation of the principal eigenvector of the task-similarity matrix. Standard spectral analysis techniques—such as those based on the Davis-Kahan theorem—only provide bounds in the $l_2$-norm, which are insufficient for establishing exact recovery of all tasks. Our analysis, instead, leverages the low-rank plus perturbation structure of the expected task-similarity matrix and adapts techniques from Fan et al. [2018] to obtain tight $l_\infty$-norm control. This adaptation is non-trivial and, to our knowledge, has not been done before in the crowdsourcing literature.
> This contribution is clearly outlined in Contribution Point 2 and further discussed in Remark 1 of the paper.
>
> **2. Comparison with Kaito et al. (2024) [1]**
> We thank the reviewer for suggesting a comparison with Kaito et al. (2024). While both works consider different types of tasks, the models are significantly different.
>
>  (a) In Kaito et al., all workers are statistically identical, leading to a rank-1 expected response matrix. In contrast, our model assumes heterogeneous worker reliabilities for each task types, resulting in a rank-2 structure. This heterogeneity is central to crowdsourcing settings and necessitates weighted majority voting rather than simple majority voting.
>
>  (b) Kaito et al.'s clustering method requires the number of workers to exceed the number of tasks for reliable recovery. Our analysis, by contrast, shows that only log(d) workers are sufficient—making our approach significantly more practical for expert labeling scenarios.
>
>  We are happy to add this comparison in the related work section.
>
>
> **3. Modeling Assumptions**
> We appreciate the reviewer’s concerns regarding the assumptions of (i) two task types and (ii) full worker-task response matrices.
>
> (a) Two Task Types: Our algorithm can, in principle, be extended to more than two types by applying k-means to the dominant eigenvectors. However, accurate label estimation in such settings requires the reliability vectors of each type to be sufficiently distinct, and the data requirements grow with the number of types. While we have not yet explored the theoretical challenges associated with extending the model to handle more than two types of tasks, we recognize this as an interesting direction for future research.
>
> (b) Complete Assignments: Our primary goal is to theoretically establish a novel perfect clustering result in multi-type crowdsourcing settings under practically motivated assumptions on worker-task complexity. While extending the theory to handle sparsity is an important direction, we deliberately chose to focus on the fully connected setting to emphasize the core contribution.
>
> **4. Robustness to Sparse Assignments**
> Although our theoretical analysis assumes complete worker-task assignments, the real-world datasets used in our experiments—Dog, Temp, and RTE—are inherently sparse, with each worker labeling only a subset of tasks.
>
>
> **5. Writing and Organization**
> We appreciate the reviewer’s feedback on the presentation. We tried our best to structure the paper to balance technical rigor with accessibility. Nonetheless, in the revised version, we will include these for further improvements:
>
> (a) Add a summary table of all notations for quick reference.
>
> (b) Review and revise the structure of Section 3 for improved logical flow.
> We would also welcome any specific suggestions from the reviewer on sections they found unclear or difficult to follow.
>
> **6. Related Work**
>
>
> Thanks for the suggestion for including more recent literature to better position the scope of this paper. In view of this suggestion, we will include a broader range of literature on clustering in the revised version. We note that we focus solely on clustering tasks based on their type (e.g., easy or hard), motivated by extensions of the Dawid-Skene model where worker reliability varies across task types. In contrast, collaborative clustering typically aims to jointly cluster both tasks and workers under a block matrix or factor model. We compare our algorithm with Kim et al 2022 [2] which considers joint clustering of tasks and workers
>
> [1] Ariu, Kaito, et al. "Optimal clustering from noisy binary feedback." Machine Learning 113.5 (2024): 2733-2764.
>
>
> [2] Doyeon Kim, Jeonghwan Lee, and Hye Won Chung. A generalized worker-task specialization model for
> crowdsourcing: Optimal limits and algorithm. In 2022 ISIT.

---

> > ### Author Response · Authors · 2025-07-09
> > **Note on Revised Submission and Request for Further Review**
> >
> > Dear Reviewer Yfhp,
> >
> > We hope our previous responses to your comments were helpful and addressed your concerns satisfactorily. Following that exchange, we have submitted a revised version of the paper, with all changes highlighted in blue. A summary of the key updates is included in the revision, but we would also like to take this opportunity to highlight specific changes made in response to your suggestions.
> >
> > In particular, the related work section has been substantially revised to more clearly differentiate our contributions from prior literature. This includes works on heterogeneous task-based models, spectral clustering in crowdsourcing and beyond, as well as the papers you cited. The novelty and contribution of our work are now further reinforced through updates to both the contributions and related work sections. We clarify how our analysis differs from classical spectral clustering results in the stochastic block model (SBM) and Gaussian mixture model (GMM) settings, and why those techniques do not directly yield the type of worker complexity guarantee for perfect clustering that we establish in this work.  Furthermore, we highlight a gap in the literature: existing task-heterogeneous models do not provide any theoretical guarantee on task-type separation or clustering accuracy, which our work explicitly addresses.
> >
> > We also re-evaluated the presentation of the technical content and made several improvements. A table summarizing key notations has been added, and we now provide geometric interpretations of the main clustering theorem to enhance understanding. We have tried our best to make the technical sections more accessible, and we would be very grateful for any specific suggestions you might have on how to improve this further.
> >
> > We respectfully request that you evaluate our paper based on this revised version. Please do not hesitate to let us know if you have any additional comments or concerns—we would sincerely appreciate your feedback.
> >
> > Looking forward to hearing from you.

---

### Review · Reviewer_tuu7 · 2025-06-28

**Summary Of Contributions:**

The authors consider a clustered variant of the Dawid-Skene (DS) model, by assuming that there are two types of tasks and each worker's mistake probability is determined by the task's type. The paper proposes a two-stage approach, where in the first stage, the tasks are clustered from the (empirical) task-similarity matrix via computing the leading eigenvector and thresholding based on its entries. In the second tage, existing DS Algorithm is used per cluster, which consists of Triangular Estimation (TE) and Nitzan-Paroush (NP). The authors show that even with weighted majority voting scheme, the error exponent without clustering is strictly worse than the proposed clustering-based approach. The proposed approach is empirically compared against various baselines, showing its efficacy.

**Audience:**

Yes

**Claims And Evidence:**

Yes

**Requested Changes:**

**Minor**
- "proposition" "algorithm" "theorem" should be capitalized whenever applicable
- Showing the runtime would be helpful to see the overhead of two-stage approach, as well as runtime comparison of different algorithms.
- Also, maybe I missed it, but seeing the actual clustering accuracy would be nice.

**Critical**
- Some relevant discussions and references are missing. For instance, in clustering of GMMs, Lu & Zhou (2016) [2], they showed that for a general Dawid-Skene model including different task types with different reliability vectors, if I'm not mistaken. Also, [1] seems to also perform weighted majority voting of some sort. I can vaguely see that there are indeed some differences, but given the amount of literature in crowdsourcing theories, I feel that there is a strong need to clearly articulate the difference from the current submission and prior works, both model-wise and guarantee-wise, maybe even as a separate section/appendix.
- The result that a logarithmic number of tasks suffices for perfect clustering highly resembles the results for clustering in mixture models, such as GMMs [2,3]. Some discussions should be provided regarding this as well.
- In Proposition 1, the error exponents depend on $e^{w\_i}$'s. Then if $w_i = w$ for some $w > 0$, then we should recover the unweighted version. But then, $\hat{y}^{WMV}$ should be the same regardless of what $w > 0$ is, yet the guarantee seems to depend heavily on $w$, i.e., it is not "scale-invariant" in some sense. If my description is not mistaken, the authors should elaborate on this "gap."


**Questions**
- The idea of using the entries of leading eigenvector for thresholding seems similar to Fiedler vector entries that can be used to partition a graph via algebraic connectivity. Is this somewhat related to the approach taken by the authors?
- Is the Lloyd-type baseline of [1] also applicable here?


[1] https://ieeexplore.ieee.org/document/10252002
[2] https://arxiv.org/abs/1612.02099 (Sec 4.2)

**Strengths And Weaknesses:**

**Strengths**
- The use of $\ell\_\infty$-concentration to obtain tight guarantees for spectral clustering is clever. Also, interesting how spectral clustering alone can achieve perfect clustering without additional refinement steps as in SBMs or GMMs.
- Algorithm-specific lower bound for weighted majority voting schemes clearly shows the superiority of cluster-then-DS approach.
- Experiments with various baselines

**Weaknesses**
- Writing can be improved.
- Slightly lacking discussions related to prior work; see Requested Changes below.

---

> ### Author Response · Authors · 2025-07-03
> **Part 1 of the response: Addresses writing, notation, prior work, runtime performance, and a request for clustering accuracy.**
>
> Thank you for the constructive feedback. We have addressed your points in detail across three separate official comments. This is part 1.
>
> ### Comment
>
> Writing can be improved
>
> ### Response
>
> While we aimed to balance technical depth with clarity, we will make the following improvements in the revised version:
>
> (a) Include a summary table of all notations for easy reference.
>
> (b) Review and refine the structure of Section 3 to enhance the logical flow.
>
> We also welcome any specific suggestions.
>
> ### Comment
>
> Slightly lacking discussions related to prior work; see Requested Changes below.
>
> ### Response
>
> After carefully considering the reviewers’ comments, we plan to incorporate the cited papers to better position our work within the existing literature—both in terms of modeling assumptions and empirical performance. We will submit a revised version soon that includes a broader range of related work, aligned with this goal.
>
> ### Comment
>
> "proposition" "algorithm" "theorem" should be capitalized whenever applicable
>
> ### Response
>
> Thank you for pointing this out. We will upload a revised version of the manuscript in which “Proposition,” “Algorithm,” and “Theorem” are capitalized appropriately when referring to specific numbered items, in accordance with standard conventions.
>
> ### Comment
>
> Showing the runtime would be helpful to see the overhead of two-stage approach, as well as runtime comparison of different algorithms.
>
> ### Response
>
> We appreciate the comment of showing the runtime performance of different algorithms, particularly the overhead due to the clustering step. Our spectral clustering step is dominated by the computation of the principal eigenvector of the task-similarity matrix of size $d\times d$,  which can be done in $O(d^3)$ steps. The DS-based TE algorithm (the second step after clustering) that we used in our paper can be implemented in $O(n^2d)$ steps (dominated by the calculation of the worker-similarity matrix, see Appendix B of the paper).
>
> To illustrate the run-time of the DS-based and our two-step approaches, we provide here the average run time for the two algorithms : TA-TE (type-agnostic, DS-based) and TD-TE (clustering + TE+ NP-WMV) for the three practical datasets we considered in our paper : Bluebird, Dog, HC-TREC. We also added the runtime for the clustering step for different data. We see that the overhead is negligible for practical values of n and d. We use standard CPU in google Colab to report the runtime.
>
> **Runtime comparison (seconds)**
>
> | Dataset&nbsp;(workers, tasks) | Clustering | Type-agnostic TE | Type-dependent TE† |
> |-------------------------------|-----------:|-----------------:|-------------------:|
> | Bluebird (39 w, 108 t)       | **0.0998** | 0.0318 | 0.1280 |
> | Dog (78 w, 807 t)            | **0.2217** | 0.2064 | 0.5935 |
> | HC-TREC (10 w, 1000 t)       | **0.2927** | 0.0212 | 0.3207 |
>
> † *Type-dependent TE = clustering time + TE executed separately on each estimated type.*
>
>
> For comparing with type-dependent algorithms, we use the existing Matlab code provided by the author of the paper Kim et al 2023 [1]. The SDP algorithm (Kim et al) due to its complex parameter tuning procedure and using a semi-definite program solver, suffers from a substantially higher (few minutes to half an hour in a 16GB RAM laptop) runtime.
>
> ### Comment
>
> Also, maybe I missed it, but seeing the actual clustering accuracy would be nice
>
> ### Response
>
> Thank you for the suggestion. In real-world crowdsourcing datasets, ground-truth task types are not available—only the correct labels for each task are provided. Since our notion of task types is a modeling construct rather than part of the dataset, clustering accuracy cannot be evaluated on these datasets.
>
> However, if the reviewer finds it useful, we would be happy to include additional experiments on synthetic datasets where ground-truth types are known. These can illustrate how clustering performance varies with key problem parameters such as the angle and norm difference between reliability vectors and the number of tasks and workers.
>
> [1] Kim, Doyeon, Jeonghwan Lee, and Hye Won Chung. "A worker-task specialization model for crowdsourcing: Efficient inference and fundamental limits." IEEE Transactions on Information Theory 70.3 (2023): 2076-2117.

---

> ### Author Response · Authors · 2025-07-03
> **Second part of the response: including comparisons with GMM/DS models and related spectral methods.**
>
> ***Comment:***
>
> Some relevant discussions and references are missing. For instance, in clustering of GMMs, Lu & Zhou (2016) [2], they showed that for a general Dawid-Skene model including different task types with different reliability vectors, if I'm not mistaken. Also, [1] seems to also perform weighted majority voting of some sort. I can vaguely see that there are indeed some differences, but given the amount of literature in crowdsourcing theories, I feel that there is a strong need to clearly articulate the difference from the current submission and prior works, both model-wise and guarantee-wise, maybe even as a separate section/appendix.
>
> ***Response:***
>
> We appreciate the reviewer’s suggestion. After carefully considering the comments, we plan to incorporate the cited papers to better position our work within the existing literature—both in terms of modeling assumptions and empirical performance. We will submit a revised version that includes a broader range of related work, aligned with this goal.
>
> Specifically:
>
> (a) Lu & Zhou (2016) study a sub-Gaussian mixture model and, in the context of crowdsourcing, focus on the standard Dawid-Skene (DS) model where tasks are assumed to be probabilistically identical. While they allow multiple labels per task, the model does not account for heterogeneity across task types. Their Lloyd-type algorithm is closely related to Expectation-Maximization (EM) with spectral initialization. In contrast, we argue that in multi-type crowdsourcing settings, a type-dependent model is essential. Our paper demonstrates both empirically (via labeling performance and the eigen-spectrum of the task-correlation matrix) and theoretically why clustering tasks before applying a DS-style algorithm significantly improves performance in such settings.
>
> (b) Kim et al. (2023) consider a different notion of task-type dependency using a k-type specialization model. Their approach applies majority voting (MV) rather than weighted majority voting (WMV) in the second stage. In Section 4, we show that WMV-based inference consistently outperforms MV on real-world datasets. Furthermore, our modeling framework and spectral clustering approach differ substantially from their assumptions and methods.
>
> We will incorporate a more explicit comparison with Lu & Zhou (2016) in the revised version. A comparison with Kim et al. is already included in the paper.
>
> ***Comment:***
>
> The result that a logarithmic number of tasks suffices for perfect clustering highly resembles the results for clustering in mixture models, such as GMMs [2,3]. Some discussions should be provided regarding this as well.
>
> ***Response:***
>
> This is an excellent observation. Indeed, the condition that a logarithmic number of workers suffices for perfect clustering bears resemblance to results in other settings, such as the $O(\log(\text{number of nodes}))$ degree condition in exact recovery for stochastic block models (SBMs), and signal-to-noise ratio thresholds in sub-Gaussian mixture models.
>
> However, while the expected matrices may look similar, the underlying probabilistic models are fundamentally different. In particular, unlike SBM, entries in our task-task similarity matrix are not independent, as they aggregate responses from shared workers.
>
> Moreover, it is not straightforward to reduce our multi-type crowdsourcing model to a standard mixture model. The paper by Hu and Zhou (2016) considers only the single-type DS model in the crowdsourcing context. To connect our setting to a mixture model framework, we apply a nontrivial reduction by treating the four task groups—(easy, label +1), (easy, label –1), (hard, label +1), (hard, label –1)—as four mixture components over $n$-dimensional binary vectors (worker responses).
>
> This reduction allows us to invoke mixture model techniques. However, even then, the guarantees differ: Hu and Zhou show perfect clustering with probability at least $1 - e^{-\sqrt{n}}$ under $n = O(\log d)$, whereas we achieve a stronger bound of $1 - e^{-n}$ under the same condition.
>
> We will include a discussion of these connections and distinctions in the revised version to better contextualize our contributions.
>
>
> References
>
> [1] Kim, Doyeon, Jeonghwan Lee, and Hye Won Chung. A worker-task specialization model for crowdsourcing: Efficient inference and fundamental limits. IEEE Transactions on Information Theory 70.3 (2023): 2076–2117.
>
> [2] Lu, Yu, and Harrison H. Zhou. Statistical and computational guarantees of Lloyd's algorithm and its variants. arXiv preprint arXiv:1612.02099 (2016).
>
> [3] Gao, Chao, Yu Lu, and Dengyong Zhou. Exact exponent in optimal rates for crowdsourcing. In Proceedings of the 33rd International Conference on Machine Learning, PMLR, 48:603–611 (2016).

---

> ### Author Response · Authors · 2025-07-03
> **Third part of the response: theoretical clarification, including comparisons with GMM/DS models and related spectral methods.**
>
> ***Comment:***
>
> In Proposition 1, the error exponents depend on $e^{w_i}$. Then if $w_i = w > 0$, we should recover the unweighted version. But then $\hat{y}^{\text{WMV}}$ should be the same regardless of what $w$ is, yet the guarantee seems to depend heavily on $w$, i.e., it is not "scale-invariant" in some sense. If my description is not mistaken, the authors should elaborate on this "gap."
>
> ***Response:***
> This is an excellent question. The guarantee is indeed scale-invariant with respect to a positive scaling of $w$. Specifically, from Equation (17) of the expression $\varphi_n(w,r_k)$, we can see that:
> $\varphi_n(w,r_k) =  \varphi_n(aw,r_k)$
> for any $a > 0$. This is because the optimizing parameter $t$ in the exponent is multiplied by $w_i$, and scales accordingly.
>
> Consequently, when $w_i = w > 0$, the WMV rule reduces to unweighted majority voting, and the error exponent matches the one established in Theorem 5.1 of Gao et al. (2016) [3].
>
> ***Comment:***
>
> The idea of using the entries of the leading eigenvector for thresholding seems similar to Fiedler vector entries used to partition a graph via algebraic connectivity. Is this somewhat related?
>
> ***Response:***
>
> Thank you for raising this point. Although the Fiedler vector in stochastic block models (SBMs) and the principal eigenvector of our task-similarity matrix both encode a latent bipartition in expectation, they stem from fundamentally different probabilistic models.
>
> An SBM Laplacian is built from independent Bernoulli edges, with off-diagonal entries being 0/1, and communities recovered via the sign of the second eigenvector.
>
> By contrast, our crowdsourcing matrix is generated via a type-dependent worker-labeling process. Each entry aggregates the agreement of all $n$ workers on a task pair, takes integer values in $[0, n]$, and exhibits strong dependencies whenever two entries share a task. Differences in noise distribution and independence structure are reflections of this deeper modeling gap, which is why classical SBM/Fiedler guarantees cannot be directly transferred. This motivates our perturbation bounds and exact-clustering results, tailored to the multi-type crowdsourcing setting.
>
> ***Comment:***
>
> Is the Lloyd-type baseline of [1] also applicable here?
>
> ***Response:***
>
> We interpret the Lloyd-type baseline as referring to the majority voting baseline used by Kim et al. (2023) [1]. In Section 4 of our paper, we compare our algorithm against the baselines considered in their work, including their proposed algorithm, and demonstrate that our method outperforms them on the datasets we studied.
>
> References
>
> [1] Kim, Doyeon, Jeonghwan Lee, and Hye Won Chung. A worker-task specialization model for crowdsourcing: Efficient inference and fundamental limits. IEEE Transactions on Information Theory 70.3 (2023): 2076–2117.
>
> [2] Lu, Yu, and Harrison H. Zhou. Statistical and computational guarantees of Lloyd's algorithm and its variants. arXiv preprint arXiv:1612.02099 (2016).
>
> [3] Gao, Chao, Yu Lu, and Dengyong Zhou. Exact exponent in optimal rates for crowdsourcing. In Proceedings of the 33rd International Conference on Machine Learning, PMLR, 48:603–611 (2016).

---

> ### Author Response · Authors · 2025-07-09
> **Note on Revised Submission and Request for Further Review**
>
> Dear Reviewer tuu7,
>
> We hope our previous responses to your comments were helpful and addressed your concerns satisfactorily. Following that, we have submitted a revised version of our paper, with all changes highlighted in blue. A summary of the key updates is included in the revision, but we would also like to take this opportunity to highlight specific changes made in response to your suggestions.
>
> In particular, the related work section has been substantially revised to more clearly differentiate our contributions from existing literature, including both multi-type crowdsourcing models and classical results in Gaussian Mixture Models (GMMs) and Stochastic Block Models (SBMs). Based on your feedback, we have also included a runtime comparison of different algorithms, including the overhead introduced by the clustering step. Additionally, we added a synthetic experiment in the appendix illustrating how clustering error varies with problem parameters. We added a notation table, along with a geometric interpretation of the main clustering result to aid understanding.
>
> We respectfully request that you evaluate our paper based on this revised version. Please let us know if you have any further suggestions or concerns—we would greatly appreciate your feedback.
>
> Looking forward to hearing from you.

---

### Decision · Action_Editor_ULfe · 2025-08-25

**Recommendation:** Accept with minor revision

**Additional Comments:**

Adding more discussion on how the approach could extend to realistic settings and on the practical implications would help strengthen the contribution.

**Audience:**

Yes

**Audience Explanation:**

The paper could be of interest for the crowdsourcing community.

**Claims And Evidence:**

Yes

**Claims Explanation:**

This paper proposes a spectral clustering–based approach for multi-type crowdsourcing, supported by theoretical guarantees and empirical improvements over baselines. The method is well-analyzed and effective within its stated scope.

---

> ### Author Response · Authors · 2025-09-24
> **Camera Ready Revision Submitted**
>
> Respected action editor,
>
> The camera-ready revision is submitted.
>
> A discussion section has been added on how the approach could be extended to realistic settings and about the practical implications of our work. This is in response to the minor revision suggested.